# Encephalopathy-linked UFM1 variants impede neuronal protein translation, development, and function

Catarina Perdigão[1], Josefa Torres [ID][1], Helge M Magnussen [ID][2], Janina Koch[3], Elena Rudashevskaya[1], Frederieke Moschref[1], Maksims Fiosins[4], Fritz Benseler [ID][1], Sally Wenger[1], Tanja Nilsson[1], Sabine Beuermann[1], Stefan Bonn[4,5], Silvio O Rizzoli [ID][6], Yogesh Kulathu [ID][2], Olaf Jahn [ID][1,7], Benjamin H Cooper[1], Mateusz C Ambrozkiewicz[3], JeongSeop Rhee[1], Nils Brose [ID][1✉] & Marilyn Tirard [ID][1✉]

## Abstract

Genetic variants that hinder post-translational protein modifications by UFM1, UFMylation, cause encephalopathies. UFMylation regulates endoplasmic reticulum (ER) homeostasis, but how UFMylation deficiencies cause selective neurological defects is unknown. Using murine UFM1-deficient neurons, we investigated two types of UFMylation pathologies, UFM1 loss and expression of a pathogenic UFM1-R81C variant. We found that UFM1-deficiency confounds neuron development and synapse function. Mechanistically, UFM1 loss is associated with induction of ER stress, activation of the unfolded protein response (UPR) pathway, and reduced protein translation. These defects are rescued by wild-type UFM1, but only partially by UFM1-R81C. UFM1-deficient and UFM1-R81C-expressing neurons display distinct responses to ER stress, indicating that UFM1-R81C is not merely a loss-of-function variant. Exploring therapeutic options, we show that Trazodone, an inhibitor of the UPR, restores protein translation solely in UFM1-R81C-expressing neurons, and increases synapse numbers in both UFM1-KO and UFM1-R81C-expressing neurons. Our study unveils a pivotal role for UFMylation in neuronal development, provides a molecular understanding of the signaling mechanisms altered in UFM1-associated encephalopathies, and offers important insights into potential treatments for these disorders.

**Keywords** UFM1; Neuron; Synapse; Unfolded Protein Response; Encephalopathies
**Subject Categories** Genetics, Gene Therapy & Genetic Disease; Neuroscience

## Introduction

Post-translational protein modifications (PTMs) are integral to the regulation of protein function. They expand the functionality of the cellular proteome and are an essential determinant of the complexity and diversity of cellular phenotypes by regulating a plethora of cellular processes and signaling events. Accordingly, PTM aberrations are associated with numerous pathologies, particularly with neurological disorders (Xu et al, 2018). In the present study, we focused on PTMs involving the functionally still enigmatic Ubiquitin-Fold Modifier 1 (UFM1) (Komatsu et al, 2004).

As is the case with ubiquitin and other Ubiquitin-like modifiers (Ubl), the conjugation of UFM1 to lysine residues of target proteins, UFMylation (Ilic et al, 2022), requires the orchestrated action of an E1 enzyme (UBA5), an E2 enzyme (UFC1), and an E3 enzyme complex (UFL1 and UFBP1), while specific peptidases (UFSP1, UFSP2) remove UFM1 from substrates, thereby establishing a dynamic UFMylation cycle (Millrine et al, 2023; Peter et al, 2022). The peptidase UFSP1 is primarily responsible for the functional maturation of UFM1 by proteolytically processing immature UFM1 and exposing a C-terminal glycine through which it is covalently linked to target proteins (Millrine et al, 2022).

Strikingly, variants of *UFM1*, *UBA5*, *UFC1*, and *UFSP2* have all been associated with developmental and epileptic encephalopathies, cerebellar ataxia, or congenital neuropathies (Briere et al, 2021; Cabrera-Serrano et al, 2020; Colin et al, 2016; Daida et al, 2018; Di Rocco et al, 2018; Duan et al, 2016; Hamilton et al, 2017; Ivanov et al, 2022; Mattern et al, 2023; Muona et al, 2016; Nahorski et al, 2018; Ni et al, 2021; Szűcs et al, 2021). Apart from some distinct features, corresponding patients are characterized by multiple common disease manifestations, including microcephaly, global developmental delay, intellectual disability, epilepsy, failure to thrive, and delayed myelination (Briere et al, 2021; Cabrera-Serrano

[1]Max Planck Institute for Multidisciplinary Sciences, Department of Molecular Neurobiology, Göttingen, Germany. [2]MRC Protein Phosphorylation and Ubiquitylation Unit, School of Life Sciences, University of Dundee, Dundee, UK. [3]Institute of Cell Biology and Neurobiology, Charité-Universitätsmedizin Berlin, Berlin, Germany. [4]Institute of Medical Systems Bioinformatics, Centers for Biomedical AI (bAIome), Center for Molecular Neurobiology (ZMNH) and Translational Immunology (HCTI), University Medical Center Hamburg-Eppendorf, Hamburg, Germany. [5]German Center for Child and Adolescent Health (DZKJ), Partner Site Hamburg, University Medical Center Hamburg-Eppendorf, Hamburg, Germany. [6]Department of Neuro- and Sensory Physiology, University Medical Center Göttingen, Göttingen, Germany. [7]Department of Psychiatry and Psychotherapy, University Medical Center Göttingen, Göttingen, Germany. ✉E-mail: Brose@mpinat.mpg.de; Tirard@mpinat.mpg.de

et al, 2020; Colin et al, 2016; Daida et al, 2018; Di Rocco et al, 2018; Duan et al, 2016; Hamilton et al, 2017; Ivanov et al, 2022; Mattern et al, 2023; Muona et al, 2016; Nahorski et al, 2018; Ni et al, 2021; Szűcs et al, 2021). As regards UFM1, two disease-causing mutations have been described so far: (i) a 3-bp deletion in the *UFM1* promoter region (NM_001286704.1:c.-273_-271delTCA) that causes a reduction of UFM1 expression (Hamilton et al, 2017; Ivanov et al, 2022; Szűcs et al, 2021), and (ii) a missense mutation [NM_016617.3:c.241 C > T:p.(Arg81Cys)] (Nahorski et al, 2018) that leads to an amino acid substitution (R81C). Both exhibit an autosomal recessive inheritance pattern, indicating that expression of one wild-type (WT) allele is sufficient for normal brain development and function. While all known disease-related variants of *UFM1*, *UBA5*, and *UFC1* described to date cause decreased UFMylation, the known *UFSP2* mutations described so far cause an increase in UFMylation, indicating that a tight balance of UFMylation is essential for brain development and function (Briere et al, 2021; Cabrera-Serrano et al, 2020; Colin et al, 2016; Daida et al, 2018; Di Rocco et al, 2018; Duan et al, 2016; Hamilton et al, 2017; Ivanov et al, 2022; Mattern et al, 2023; Muona et al, 2016; Nahorski et al, 2018; Ni et al, 2021; Szűcs et al, 2021). Knock-out mouse models confirm this notion (Muona et al, 2016; Zhang et al, 2022), as deletion of *Ufl1* or *Ufbp1* in the mouse forebrain leads to microcephaly, epileptic seizures, and neuron loss (Zhang et al, 2022), and *Ufm1*-deletion in mouse neuronal precursor cells causes microcephaly and neuronal apoptosis (Muona et al, 2016). These phenotypes resemble the encephalopathic features of patients with aberrant UFMylation and demonstrate that neuronal development and function require UFMylation.

Beyond the pathological phenomena described above, very little is known about the molecular processes that are regulated by UFMylation in neurons, and the molecular defects associated with pathological aberrant neuronal UFMylation in neurons are unknown. Furthermore, only a few UFM1 substrates have been identified to date. All of these are associated with ER function, but it is not known whether or how these are related to the pathology observed in patients or which neuronal functionalities are affected.

With the present study, we attempted to resolve the key open issues regarding the nerve cell biology and pathobiology of UFM1. Using neurons from an Ufm1 conditional knock-out mouse line (UFM1-cKO) and a virus-based genetic complementation approach, we mimicked the pathological UFMylation scenario described in various encephalopathies, either by reducing UFM1 expression levels or by expressing mutant UFM1-R81C in neurons.

We show that UFM1-deficiency causes impaired neuronal development in vivo and in vitro, including a partial loss of synapses with consequent defects in synaptic signaling, as well as induction of ER stress, activation of the UPR, and reduced protein translation. Re-expression of WT UFM1 rescued these defects, while expression of UFM1-R81C only partially restored them and aggravated the ER-stress response. Mechanistically, in vitro UFMylation experiments showed a drastically reduced activation of UFM1-R81C by the E1 UBA5, indicating that UFMylation is impaired with UFM1-R81C. Finally, with regard to potential therapeutic strategies, we found that treatment of neurons with Trazodone, an antidepressant that can promote neuronal protein biosynthesis in certain settings and tamper with the excessive activation of the UPR, restores protein translation in UFM1-

R81C-expressing neurons and causes an increase in synapse number (Albert-Gasco et al, 2024).

Overall, our work reveals a key function of UFM1 in the regulation of neuronal development and function. Apart from some distinct features, pathologically relevant perturbations of UFMylation, here UFM1 loss and UFM1-R81C expression, cause a set of common cellular defects. The finding that some of these are ameliorated by the antidepressant Trazodone represents an interesting first step toward therapies of neurological disorders caused by aberrant UFMylation.

# Results

## UFM1-loss alters neuronal development and synapse numbers in vivo and in vitro

Most UFM1-associated encephalopathies are characterized by reduced UFMylation (Briere et al, 2021; Cabrera-Serrano et al, 2020; Colin et al, 2016; Daida et al, 2018; Duan et al, 2016; Hamilton et al, 2017; Muona et al, 2016; Nahorski et al, 2018; Szűcs et al, 2021). To study the effects of UFM1-loss in mouse cortex and circumvent the lethality of UFM1-KO, we combined CRISPR/Cas9 and in utero electroporation (IUE) to deplete UFM1 in E14.5 mouse cortical progenitors and analyzed their neural progeny at P23. The effectiveness of the CRISPR/Cas9 tools was confirmed in vitro using N2a cells (Appendix Fig. S1). Analysis of the complexity of neuronal dendrites of cortical pyramidal upper-layer neurons via Sholl analysis revealed a substantially reduced dendrite complexity of UFM1-KO neurons as compared to control cells (Fig. 1A–D). Furthermore, immunolabeling experiments demonstrated a significant decrease in the number of pre-synaptic (Synapsin1-positive) and post-synaptic (Shank2-positive) structures (Fig. 1E–H), indicating that UFM1-depletion in cortical lineages impairs neuronal development and synapse formation in vivo.

To assess the mechanisms underlying the defects caused by UFM1-deficiency, we made use of a conditional UFM1-KO mouse line (UFM1-cKO; Fig. 2A) and used an in vitro approach to study morphological and functional defects of UFM1-depleted neurons. In these and all following experiments, primary hippocampal UFM1-cKO neurons were infected after one day in vitro (DIV 1) with CRE-expressing virus (CRE) to induce UFM1 depletion, or with RFP-expressing virus as a control (RFP), and analyzed after either DIV 5 or DIV 10-13. Western blot analysis of DIV 10 neuron lysates revealed a 90% decrease in total UFM1 protein expression upon CRE-expression (Fig. 2B,C), validating our KO model. Interestingly, Western blot analyses showed that several UFM1-positive protein species of higher molecular weight were decreased upon UFM1-deletion, indicating the existence of several yet unknown UFM1 conjugates (Fig. 2B).

Regarding neuronal morphology, imaging analyses showed that UFM1-depleted neurons at DIV 13 are characterized by reduced soma size, dendrite complexity and branching, and reduced total dendrite length as compared to control neurons (Fig. 2D–I). Strikingly, this phenotype appears early during development, as UFM1-depleted neurons displayed these morphological defects already at DIV 5 (Appendix Fig. S2A–E). Furthermore, immuno-labeling analyses of UFM1-depleted neurons at DIV 13

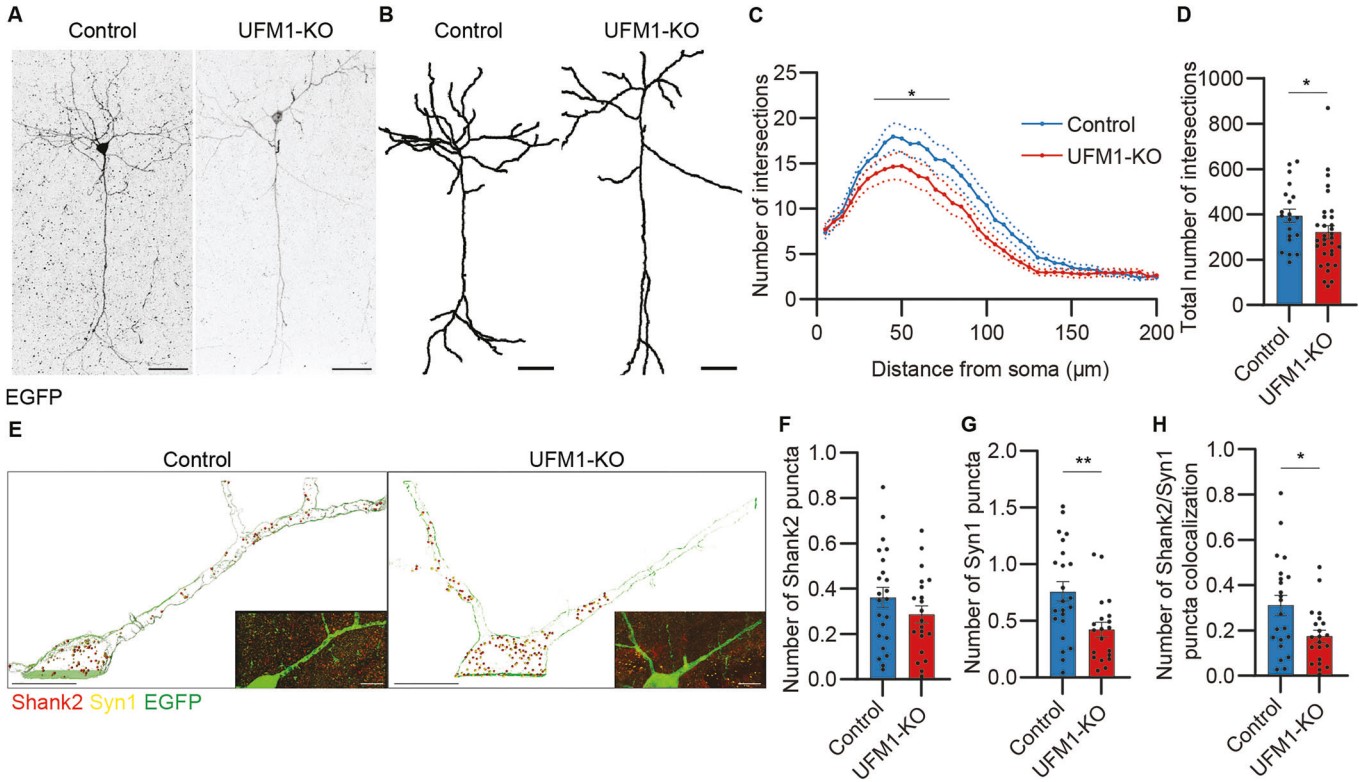

**Figure 1. UFM1 depletion reduces neuronal complexity and synapse number in the mouse brain.**

(A) Representative images of EGFP-immunolabelled (black) neurons in P23 sagittal cortical sections after IUE at E14.5 with plasmids encoding EGFP (control) or EGFP-sg-UFM1 (KO). Scale bar, 50 µm. (B) Three-dimensional reconstitution of a typical neuron based on its corresponding EGFP immunosignal, as shown in (A). Scale bar, 50 µm. (C) Sholl analysis line graph depicting the average number of intersections between dendrites and concentric circles radiating every 5 µm from the soma, using neuronal traces as shown in (B). Data were obtained from $N = 3$ different mouse brains and show the average of $n = 20$ control and $n = 32$ KO neurons. *$P = 0.0492$ using a D'Agostino–Pearson normality test and a Mann–Whitney test. Dotted lines show averages ± SEM. (D) Bar graph showing the total number of dendritic intersections, obtained from the Sholl analysis in (C). Data were obtained from $N = 3$ different mouse brains and show the average of $n = 20$ control and $n = 32$ KO neurons. *$P = 0.0492$ using a D'Agostino–Pearson normality test and a Mann–Whitney test. Dotted lines show averages ± SEM. (E) Representative 3D reconstructions of a typical control (WT) and UFM1-KO, EGFP-positive neuron (thin green line) immunostained for Shank2 (red) and Synapsin1 (yellow), as shown in the bottom right insets. IUE was performed on embryonic day E14.5 with plasmids encoding EGFP (WT) or EGFP-sg-UFM1 (UFM1-KO), and mouse brains were analyzed using anti-EGFP, Shank2, and Synapsin1 immunostaining at P23. Scale bar, 10 µm. (F–H) Bar graph depicting the total number of Shank2 (F), Synapsin1 (G), and colocalized Shank2/Synapsin1 (H) puncta colocalized with EGFP, normalized to the EGFP volume. Data were obtained from $N = 3$ different mouse brains and show the average of $n = 23$ control and $n = 30$ KO neurons. $P = 0.2215$ (F), **$P = 0.0047$ (G), *$P = 0.0137$ (H) using a D'Agostino–Pearson normality test and an unpaired $t$ test. Bar graphs show averages ± SEM. Source data are available online for this figure.

demonstrated a reduction in the number of pre- and postsynaptic puncta (Fig. 2J–L; Appendix Fig. S2F–H), which is reflected by a 70% decrease in the number of colocalized Synapsin1-PSD95 and Synaptotagmin1-PSD95 puncta as compared to control neurons (Fig. 2M; Appendix Figs. S1F and S2I). We next examined whether the remaining synapses in UFM1-depleted neurons show any obvious structural changes. Using super-resolution STED imaging, we analyzed the intensity profile of immunofluorescence signals for Synaptotagmin1, VGluT1, and PSD95, in order to quantify the levels of these proteins in synapses and to analyze their subsynaptic distribution (Appendix Fig. S3). We found that the intensities of VGluT1, Synaptotagmin1, and PSD95 signals were very similar between UFM1-KO and control cells (Fig. S3A–C), indicating that the synaptic targeting of these major pre- and postsynaptic proteins is not affected by UFM1-loss. Likewise, we did not detect changes in the subsynaptic distribution of VGluT1 or in the alignment between Synaptotagmin1- and PSD95-positive signals in UFM1-depleted neurons as compared to control cells (Appendix

Fig. S3B,C). These results indicate that the remaining synapses formed in the absence of UFM1 are comparable to those of control neurons. Altogether, the data described above show that UFM1 depletion strongly perturbs neuronal development and synapse formation in vivo and in vitro, indicating that UFM1 plays a key role in the underlying control processes.

## UFM1-loss perturbs excitatory synaptic transmission

We next conducted electrophysiological analyses of UFM1-deficient primary hippocampal neurons using whole-cell patch-clamp recordings of autaptic neurons in order to determine whether the morphological deficits seen upon UFM1-loss are paralleled by changes in synaptic signaling (Fig. 3). UFM1-deficient neurons showed reduced evoked excitatory postsynaptic current (EPSC) amplitudes (Fig. 3A,B), a reduced size of the pool of readily releasable vesicles, as assessed by the synaptic response triggered by hypertonic sucrose shock (RRP$_{sucrose}$; Fig. 3C), but no change of

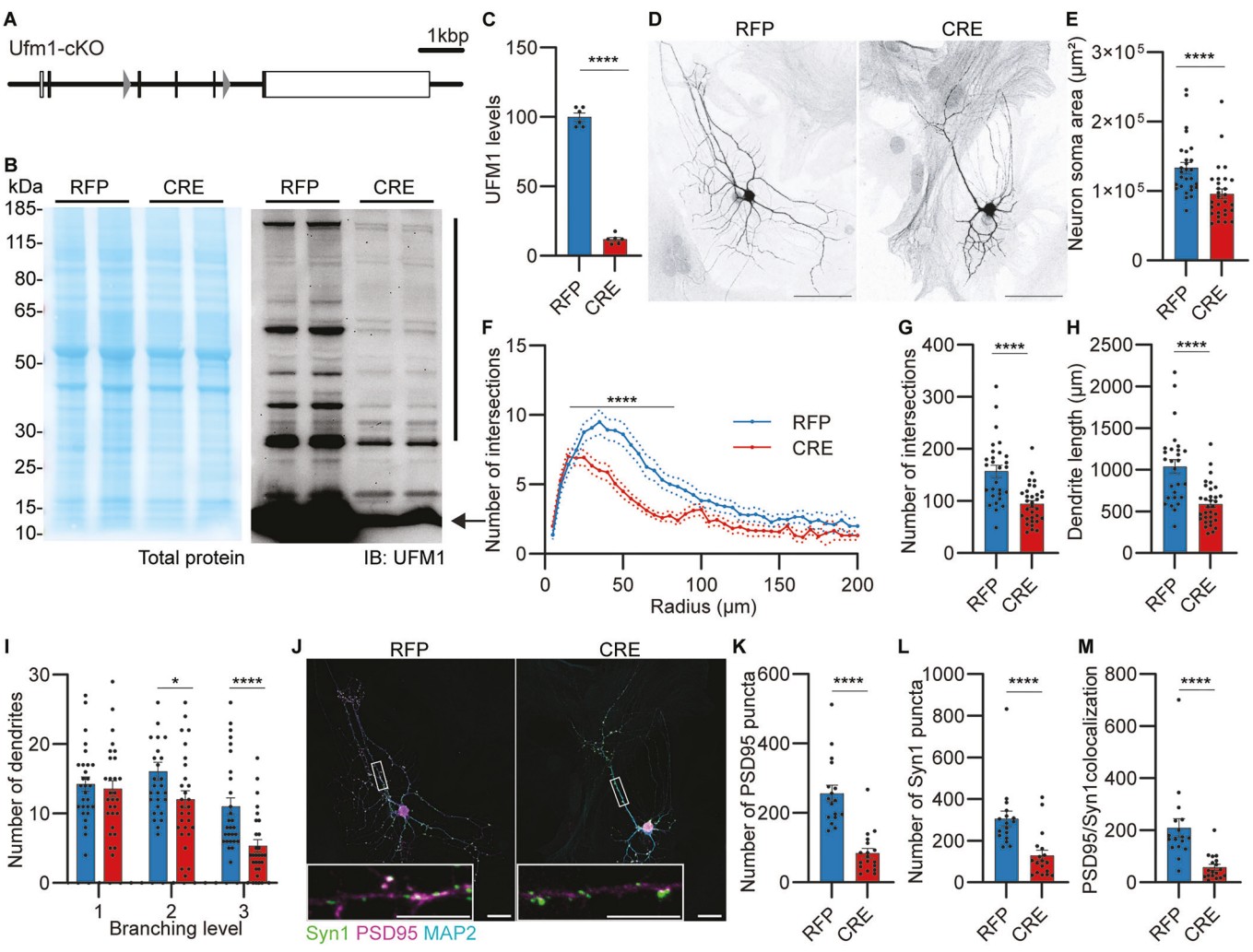

**Figure 2. UFM1 depletion reduces neuronal complexity and synapse number in vitro.**

(A) On scale schematic representation of the *Ufm1* conditional allele, indicating LoxP sites (gray triangles), coding exons (black boxes), and non-coding exons (white box). (B) Representative total protein stain (left panel) and anti-UFM1 immunoblot (IB, right panel) analysis of lysates from DIV 10 primary UFM1-cKO hippocampal neurons infected with RFP or CRE expressing viruses at DIV 1. Arrow, unconjugated UFM1; black line, UFM1 conjugates lost upon CRE-recombination. Molecular weights are indicated on the left (kDa). (C) Quantification of free and conjugated UFM1 levels, normalized to the total protein stain and expressed as a percentage of RFP. $N = 3$ experiments. ****$P = 0.0000000001$ using a D'Agostino–Pearson normality test and an unpaired *t* test. Bar graph show averages ± SEM. (D) Representative binary images of primary hippocampal UFM1-cKO neurons, infected at DIV 1 with RFP (left panel) or CRE (right panel) expressing viruses, fixed and immunolabelled for the dendritic marker MAP2 at DIV 13 (black). Scale bar, 50 μm. (E) Bar graph showing the area of neuronal somata (μm²) calculated from neurons as shown in (D). ****$P = 0.00009191$ using a D'Agostino–Pearson normality test and Mann–Whitney test. Bar graphs show averages ± SEM. (F) Sholl analysis line graph depicting the average number of intersections between dendrites and concentric circles radiating every 5 μm from the soma, using neuronal traces as shown in (D). ****$P = 0.0000093103$ using a D'Agostino–Pearson normality test and an unpaired *t* test. Dotted lines show averages ± SEM. (G) Bar graph showing the total number of dendritic intersections, obtained from the Sholl analysis in (F). ****$P = 0.0000093103$ using a D'Agostino–Pearson normality test and an unpaired *t* test. Bar graphs show averages ± SEM. (H) Bar graph showing the dendrite lengths, obtained from traces in (D). ****$P = 0.0000093221$ using a D'Agostino–Pearson normality test and an unpaired *t* test. Bar graphs show averages ± SEM. (I) Bar graph showing the number of primary, secondary and tertiary branching from dendrites (branching levels 1, 2, or 3, respectively) of neurons, as shown in (D). *$P = 0.0338$, ****$P = 0.00009247$ using a D'Agostino–Pearson normality test and Mann–Whitney test. Bar graphs show averages ± SEM. Data for (E–I) were obtained from $N = 5$ independent experiments, $n = 27$ RFP and $n = 32$ CRE neurons. (J) Representative images of primary hippocampal UFM1-cKO neurons, infected at DIV 1 with RFP (left panel) or CRE (right panel) expressing viruses, fixed and immunolabelled for Synapsin1 (green), PSD95 (magenta), and MAP2 (cyan) at DIV 13. White frame insets indicate the enlarged view below. Scale bars, 50 μm, 10 μm (insets). (K) Bar graph depicting the total number of PSD95 obtained from neurons as shown in (J). ****$P = 0.0000000833$ using a D'Agostino–Pearson normality test and a Mann–Whitney test. (L) Bar graph depicting the total number of Synapsin1 obtained from neurons as shown in (J). ****$P = 0.00004686$ using a D'Agostino–Pearson normality test and a Mann–Whitney test. (M) Bar graph depicting the total number of PSD95/Synapsin1 colocalized puncta obtained from neurons as shown in (J). ****$P = 0.000002006$ using a D'Agostino–Pearson normality test and a Mann–Whitney test. Data for (K–M) were obtained from $N = 3$ independent experiments, $n = 17$ RFP and $n = 18$ CRE cells. Bar graphs show averages ± SEM. Source data are available online for this figure.

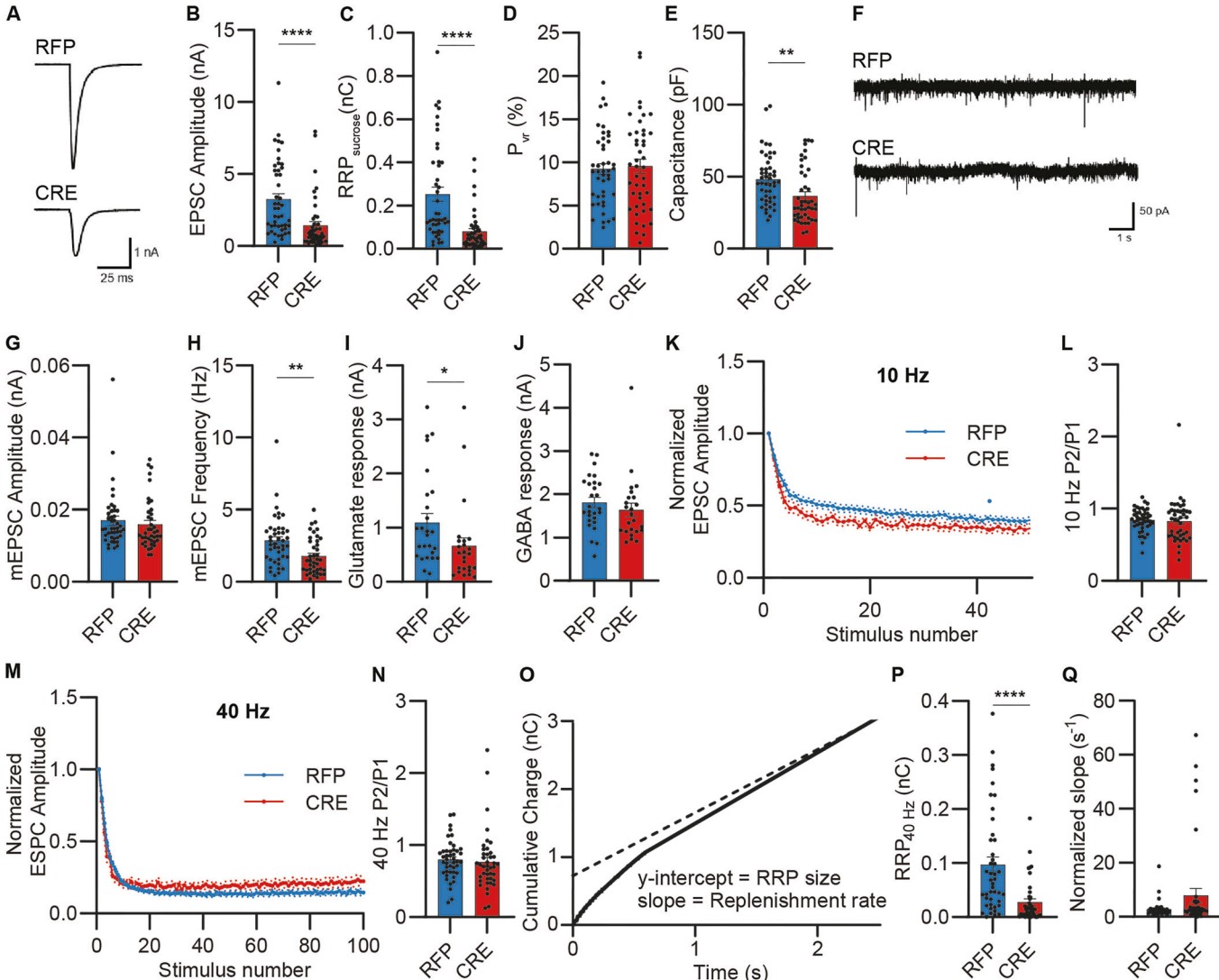

vesicular release probability ($P_{vr}$) (Fig. 3D). The membrane capacitance of cell somata was decreased in UFM1-KO neurons as compared to control cells (Fig. 3E), indicating a reduction in soma size, as seen in our morphological analyses (Fig. 2E). In addition, the frequency of spontaneous miniature EPSCs (mEPSCs) was reduced in KO neurons, while the mEPSC amplitude was not changed (Fig. 3F–H), which is in agreement with the reduced synapse number in UFM1-KO neurons and the unaltered $P_{vr}$ (Fig. 3D). Transmitter receptor density at the cell surface was assessed by fast application of glutamate and γ-aminobutyric acid (GABA). UFM1-KO neurons showed decreased responses to glutamate (Fig. 3I) but no such change upon GABA application (Fig. 3J). Altogether, these results are in line with the reduction in the number of excitatory synapses seen in glutamatergic autaptic UFM1-KO neurons, with consequent changes in synaptic signaling.

Next, we studied short-term plasticity upon high-frequency stimulation of neurons (Fig. 3K,M). UFM1-KO neurons did not show changes in short-term synaptic depression or paired pulse ratio upon 10 Hz and 40 Hz stimulation (Fig. 3K–N), which is in

accord with the unaltered $P_{vr}$ in these cells and indicates that short-term plasticity is not affected by UFM1-depletion. As seen with hypertonic sucrose stimulation ($RRP_{sucrose}$), the size of the RRP as measured by the integrated charge transfer during 40 Hz stimulation ($RRP_{40Hz}$, Fig. 3O) was strongly decreased in KO cells (Fig. 3P). The RRP refilling rate as assessed by the linear phase of the charge integral normalized to the $RRP_{40\ Hz}$ was not altered upon UFM1-loss (Fig. 3O,Q).

We also investigated the impact of UFM1 loss on synaptic function in striatal inhibitory GABAergic cells (Appendix Fig. S4). We found that the corresponding phenotype is qualitatively similar to the one seen in hippocampal excitatory neurons, but less pronounced. Inhibitory UFM1-KO neurons showed no apparent changes in the complexity of their dendritic tree, while the number of their synapses was decreased (Appendix Fig. S4A–F). Indeed, evoked inhibitory postsynaptic current (IPSC) amplitudes and $RRP_{sucrose}$ were reduced, leaving the $P_{vr}$ unaffected (Appendix Fig. S4G–J). The cell membrane capacitance was also decreased in UFM1-KO neurons as compared to control cells (Appendix Fig. S4K), and the frequency and amplitude of mIPSCs were

Figure 3.   UFM1 depletion alters excitatory synaptic transmission.

(A) Representative current traces of evoked excitatory postsynaptic currents (EPSC) of DIV 10-12 primary UFM1-cKO hippocampal autaptic neurons infected with RFP (up) and CRE (bottom) expressing viruses. (B) Bar graph depicting EPSC amplitude. ****$P = 0.000001860$ using a D'Agostino–Pearson normality test and a Mann–Whitney test. (C) Bar graph depicting the charge transferred by the release of the readily releasable pool of synaptic vesicles upon treatment with a hypertonic sucrose solution (RRP$_{sucrose}$). ****$P = 0.0000000938$ using a D'Agostino–Pearson normality test and a Mann–Whitney test. (D) Bar graph depicting the vesicular release probability (P$_{vr}$). $P = 0.746$ using a D'Agostino–Pearson normality test and an unpaired $t$ test. (E) Bar graph depicting the membrane capacitance. **$P = 0.0011$ using a D'Agostino–Pearson normality test and a Mann–Whitney test. Data for (B–E) were obtained from $N = 4$ independent experiments, $n = 46$ RFP cells and $n = 44$ CRE cells. Bar graphs show average ± SEM. (F) Representative traces of spontaneous miniature EPSC (mEPSC) of DIV 10-12 autaptic UFM1-cKO neurons infected with RFP (up) and CRE (bottom) expressing viruses. (G, H) Bar graphs depicting mEPSC amplitude (G) and frequency (H). Data were obtained from $N = 4$ independent experiments, $n = 43$ RFP cells and $n = 40$ CRE cells. **$P = 0.0011$ (H) using a D'Agostino–Pearson normality test and a Mann–Whitney test. Bar graphs show average ± SEM. (I) Bar graph depicting the amplitude of the peak current generated by the postsynaptic responses in DIV 10-12 autaptic UFM1-cKO neurons infected with RFP and CRE expressing viruses to the application of glutamate (100 µM). Data were obtained from $N = 4$ independent experiments, $n = 27$ RFP cells and $n = 25$ CRE cells. *$P = 0.0182$ using a D'Agostino–Pearson normality test and a Mann–Whitney test. Bar graphs show averages ± SEM. (J) Bar graph depicting the amplitude of the peak current generated by the postsynaptic responses in DIV 10-12 autaptic UFM1-cKO neurons infected with RFP and CRE expressing viruses to the application of GABA (3 µM). Data were obtained from $N = 4$ independent experiments, $n = 27$ RFP cells, and $n = 24$ CRE cells. Bar graphs show averages ± SEM. (K) Line graph depicting the average normalized EPSC amplitudes during 10 Hz stimulation trains. Data were normalized to the first response. Data were obtained from $N = 4$ independent experiments, $n = 44$ RFP cells, and $n = 42$ CRE cells. Dotted lines show averages ± SEM. (L) Bar graph showing paired-pulse ratios (P2/P1) obtained from (K). Data were obtained from $N = 4$ independent experiments, $n = 44$ RFP cells, and $n = 42$ CRE cells. Bar graphs show averages ± SEM. (M) Line graph depicting the average normalized EPSC amplitudes during 40 Hz stimulation trains. Data were obtained from $N = 4$ independent experiments, $n = 44$ RFP cells, and $n = 42$ CRE cells. Dotted lines show averages ± SEM. (N) Bar graph showing paired-pulse ratios (P2/P1) obtained from (M). Data were obtained from $N = 4$ independent experiments, $n = 44$ RFP cells, and $n = 42$ CRE cells. Bar graphs show averages ± SEM. (O) Illustration of the use of the cumulative charge transfer plot during 40 Hz train stimulation (control RFP neuron) to determine the RRP$_{40Hz}$ size ($y$ intercept of back-extrapolated linear part of plot, stippled line) and RRP replenishment rate (slope of linear part of plot). This approach was used for all assays of RRP$_{40Hz}$ and RRP replenishment rate using 10 or 40 Hz stimulation. (P) Bar graph showing the size of the readily releasable pool (RRP$_{40Hz}$), estimated by back-extrapolation of the cumulative EPSC after 40 Hz stimulus trains as shown in (O). Data were obtained from $N = 4$ independent experiments, $n = 44$ RFP cells, and $n = 41$ CRE cells. ****$P = 0.000001678$ using a D'Agostino–Pearson normality test and a Mann–Whitney test. Bar graphs show averages ± SEM. (Q) Bar graph depicting the RRP refilling rate (slope in O) normalized to the RRP$_{40\,Hz}$ (O, P). Data were obtained from $N = 4$ independent experiments, $n = 45$ RFP cells, and $n = 41$ CRE cells. Bar graphs show averages ± SEM. Source data are available online for this figure.

reduced in UFM1-KO neurons (Appendix Fig. S4L–N). Further, inhibitory UFM1-KO neurons showed decreased responses to exogenously applied glutamate but no such change upon GABA application (Appendix Fig. S4O,P), reflecting reduced transmitter receptor density at the cell surface. Finally, short-term plasticity and the RRP refilling rate were not altered in UFM1-KO cells (Appendix Fig. S4Q–V).

These results show that UFM1-depletion causes a severe neurodevelopmental defect that is accompanied by a severe reduction in the number of excitatory and inhibitory synapses, and a corresponding defect in synaptic signaling, while the remaining synapses in UFM1-KO cells appear to function normally. Complementing this, we conducted a 3D ultrastructural analysis of synapses in high-pressure frozen and freeze-substituted cultured primary hippocampal neurons using dual-axis Transmission Electron Microscopy (TEM) tomography to quantify the size and distribution of synaptic vesicle (SV) pools in pre-synaptic terminals (Appendix Fig. S5). We found no major morphological differences between RFP and CRE (UFM1-KO) synapses (Appendix Fig. S5A–D). Within 0–200 nm from the active zone (AZ), the cumulative frequency distribution of SVs, the total number of SVs, and their SV nearest-neighbor distances were similar in both UFM1-KO and control synapses (Appendix Fig. S5E–G). Upon restriction of the analysis to AZ-proximal (0–40 nm) SV pools, the number of SVs normalized to AZ area, the number of docked SVs representing the molecularly primed population comprising the RRP (Imig et al, 2014), and the SV diameter were not different between the conditions (Appendix Fig. S5H–J). These findings show that the remaining UFM1-deficient synapses have normal ultrastructural morphology, in agreement with our functional analysis (Fig. 3).

## UFM1 loss alters protein translation

UFMylation regulates protein translation and ER homeostasis, both of which are critical during brain development (Borisova et al, 2024; Martínez et al, 2018; Vásquez et al, 2022). For instance, aberrant protein translation can cause multiple neuromorphological defects, especially in dendritogenesis (Borisova et al, 2024; Harnett et al, 2022). To determine whether the neurodevelopmental and functional defects we observed upon UFM1-KO are related to changes in neuronal protein biosynthesis, we studied protein translation in UFM1-depleted neurons using Fluorescent Noncanonical Amino Acid Tagging (FUNCAT (Dieterich et al, 2010)). In this assay, the somatic fluorescence intensity of the methionine analog homopropargylglycine (HPG) coupled to Alexa-488 (HPG/Alexa-488) is proportional to its incorporation into newly synthesized proteins and serves as a proxy of the global protein translation rate (Fig. 4). Accordingly, neurons treated with the translation inhibitor cycloheximide (CHX) showed a ~50% reduction in HPG/Alexa-488 signal intensity, validating the approach (Fig. 4A,B). UFM1-KO neurons showed a similar reduction in HPG/Alexa-488 intensity when compared to control cells, and this effect was not further aggravated upon additional CHX treatment (Fig. 4A,B), indicating a severe reduction in protein translation in UFM1-deficient neurons. We confirmed this finding using an alternative approach and pulsed neurons with puromycin (Arguello et al, 2018), which incorporates into nascent polypeptides during translation as an aminonucleoside Tyr-tRNA mimetic. Anti-puromycin Western blot analysis of total neuron lysates revealed a ~30% reduction in the levels of incorporated puromycin in UFM1-KO neurons as compared to control cells (Fig. 4C,D). Altogether, these data show that UFMylation safeguards protein translation homeostasis in neurons.

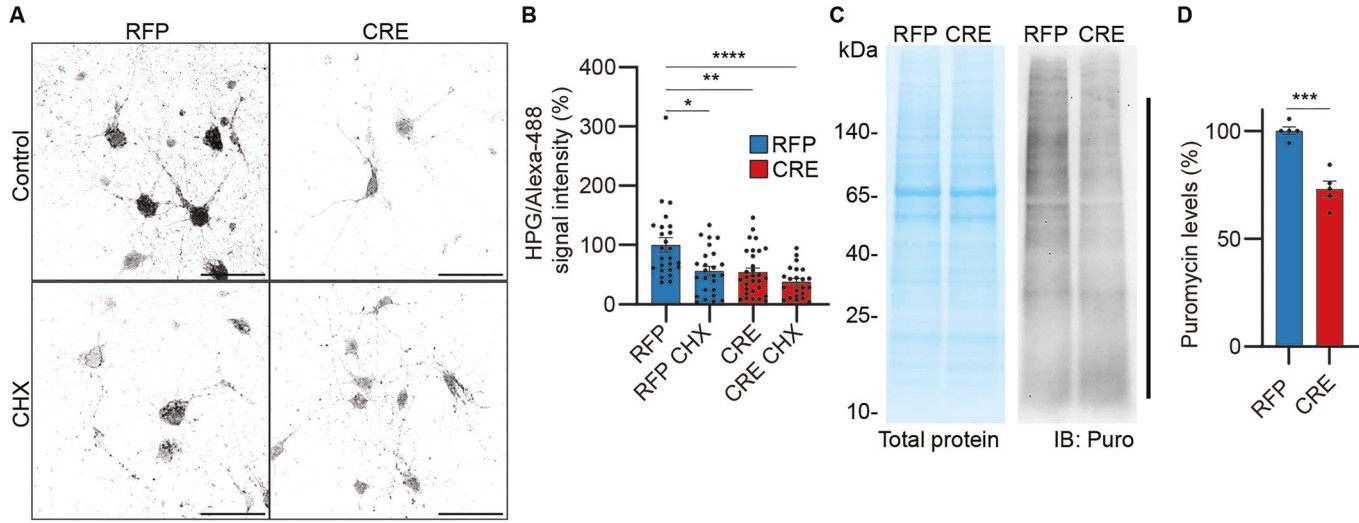

**Figure 4. UFM1 depletion impairs protein translation.**

(A) Representative images of the HPG/Alexa-488 signal (black) in DIV 10 primary UFM1-cKO hippocampal neurons infected at DIV 1 with RFP (left panels) or CRE (right panels) expressing viruses and processed for FUNCAT analysis at DIV 10. CHX cycloheximide. Scale bar, 50 μm. (B) Bar graph depicting the somatic HPG/Alexa-488 signal intensity in (A). Intensity values were normalized to RFP and expressed as a percentage. Data were obtained from $N = 3$ independent experiments, $n = 25$ RFP control, $n = 25$ RFP CHX, $n = 30$ CRE control, and $n = 22$ CRE CHX cells. *$P = 0.0176$, **$P = 0.0040$, ****$P = 0.00006463$ using a D'Agostino–Pearson normality test and Kruskal–Wallis test. Bar graphs show averages ± SEM. (C) Total protein stain (left panel) and representative anti-puromycin immunoblot (IB, right panel) analysis of lysates from DIV 10 primary UFM1-cKO hippocampal neurons infected at DIV 1 with RFP or CRE expressing viruses, incubated with puromycin (20 μg/ml) for 10 min. Black line, puromycilated proteins. (D) Quantification of anti-puromycin signal in (C, black line), normalized to the total protein levels (C, left panel) and expressed as a percentage of RFP. Data were obtained from $N = 4$ independent experiments, $n = 4$. ***$P = 0.0002$ using a D'Agostino–Pearson normality test and unpaired $t$ test. Bar graphs show averages ± SEM. Source data are available online for this figure.

To obtain higher-level insight into the molecular pathways that are indirectly affected by the loss of UFM1 and may contribute to the cellular phenotypes we observed in UFM1-deficient neurons, we performed RNA-sequencing and differential expression analysis of UFM1-KO (CRE) and control (RFP) neurons (Dataset EV1; Appendix Figs. S6–8). Upon filtering, the expression of 539 genes was significantly altered, with 355 genes being upregulated, and 184 genes showed downregulation. Interestingly, many of the down-regulated genes operate in synaptic processes such as synapse structure determination and SV cycling (Appendix Fig. S7, synaptic ontology and molecular function terms), the upregulated genes are most strongly associated with protein translation and RNA processing (Appendix Fig. S8), possibly representing compensatory changes in response to the protein translation defects described above.

## Expression of UFM1 and encephalopathy-linked UFM1-R81C in UFM1-depleted neurons

Beyond the link between reduced UFM1 levels and epileptic encephalopathy, a homozygous mutation in *UFM1* that leads to an R81C substitution (UFM1-R81C) was described in patients with early encephalopathy (Nahorski et al, 2018). In view of this, we decided to use UFM1-KO neurons in culture in combination with viral rescue experiments to examine how the UFM1-R81C variant affects neuronal morphology, synapse numbers, and synaptic signaling, using UFM1-KO neurons re-expressing wild-type UFM1 (UFM1-WT) as controls. We used UFM1-WT and

UFM1-R81C constructs containing the human cDNA sequence, but human and mouse UFM1 proteins are identical.

Neuron lysates analyzed by anti-UFM1 immunoblotting (Appendix Fig. S9A–D) confirmed the reduction in UFM1 levels upon CRE expression and the reversal of this reduction upon re-expression of UFM1-WT. Strikingly, no UFM1 signal was observed upon expression of UFM1-R81C (Appendix Fig. S9A,C). As the antibody we used targets a C-terminal region of UFM1 that encompasses residue 81, we were concerned that the R81C mutation prevents corresponding immunoblot detection. With an antibody against the N-terminal region of UFM1, we specifically, but very weakly, detected UFM1, whose levels were depleted upon CRE expression and rescued upon re-expression of either UFM1-WT or UFM1-R81C (Appendix Fig. S9B,D). However, it was not possible to detect UFM1 conjugates with this antibody, likely due to its insufficient sensitivity (Appendix Fig. S9B,D).

To resolve any limitations in detecting UFM1-R81C and to determine whether UFM1-WT and UFM1-R81C mRNAs can be re-expressed at comparable levels, we performed RT-qPCR and mass spectrometry. RT-qPCR confirmed the depletion of endogenous *Ufm1* mRNA upon CRE expression (Appendix Fig. S9E), showed that viral re-expression led to similar levels of mRNAs encoding UFM1-WT and UFM1-R81C (Appendix Fig. S9F), and demonstrated that the virus titers used were comparable as assessed by qPCR of the WPRE region of the viral construct (Appendix Fig. S9E). These data show that *UFM1-WT* and *UFM1-R81C* transcripts are expressed at comparable levels in our experimental preparation.

To validate this result at the protein level and to confirm the R81C amino acid substitution, we employed proteomic analyses of HEK cells (Appendix Fig. S10) and cultured primary neurons (Appendix Fig. S11) expressing the exogenous UFM1 variants. In HEK cells, we overexpressed the HA-tagged UFM1 variants (Appendix Fig. S10A). As expected, anti-UFM1 immunoblotting with the C-terminus-targeting antibody confirmed the expression of HA-UFM1-WT, but not of HA-UFM1-R81C, while anti-HA Western blot analysis shows that both HA-UFM1 variants are expressed at comparable levels (Appendix Fig. S10A). Proteasome inhibition by MG132 did not substantially alter these expression levels, indicating similar protein turnover rates (Appendix Fig. S10A). Based on the Western blot results, gel areas containing unconjugated UFM1 were excised from a Coomassie-stained gel, digested with the endoprotease Lys-C, and analyzed by LC-MS/MS. Both HA-UFM1 variants were identified with 100% sequence coverage in the respective samples, allowing to unambiguously distinguish between wild-type and mutant via mass spectrometric sequencing of their C-terminal peptides (Appendix Fig. S10B–F). Label-free protein quantification (LFQ) based on the MS data showed that the relative abundance of the two HA-UFM1 variants was comparable (Appendix Fig. S10D), corroborating corresponding mRNA analyses and the anti-HA Western blotting analyses.

Next, we performed a similar set of experiments in neurons (Appendix Fig. S11), with the major difference that the UFM1 variants were not fused to HA in order to avoid perturbations in UFMylation efficiency by the HA tag, and to mimic the disease state as closely as possible (Arguello et al, 2018). The UFM1 variants were quantified by the mass spectrometric approach described above, as UFM1-R81C is not detectable with the C-terminus-targeting antibody we used. Although the lower total protein levels in neurons (compare Coomassie-stained gels in Appendix Figs. S10A and S11A) required optimization of the LC-MS/MS method, LFQ of the UFM1 variants and sequencing of their C-terminal peptides were also successful (Appendix Fig. S11B–F). In sum, these results confirm the UFM1-loss upon CRE expression in UFM1-KO neurons and show that exogenous UFM1-WT and UFM1-R81C are expressed at similar levels in neurons.

## Re-expression of UFM1-R81C only partially rescues the morphological and functional defects associated with UFM1-loss

After having confirmed comparable expression levels of UFM1-WT and UFM1-R81C, we next evaluated their potential to rescue the defects in neuronal morphology, synapse numbers, synaptic signaling, protein translation, and ER homeostasis that we observed in UFM1-KO neurons. Sholl analysis showed that UFM1-loss decreased the overall dendrite complexity and length, and that these morphological defects were fully rescued by expression of UFM1-WT after both DIV 5 and DIV 13 (Fig. 5A–E; Appendix Fig. S12). In contrast, UFM1-R81C re-expression achieved only a partial rescue of the dendrite complexity defects (Fig. 5A–E). As regards the reduction in total synapse numbers seen upon UFM1-KO, only re-expression of UFM1-WT, but not of UFM1-R81C, fully restored this phenotypic defect (Fig. 5F–H), with UFM1-R81C only partially rescuing this defect.

Next, we evaluated synaptic signaling upon expression of UFM1-WT or UFM1-R81C in UFM1-depleted neurons (Fig. 6). While expression of UFM1-WT almost completely rescued the EPSC amplitudes and the $RRP_{sucrose}$ defects, expression of UFM1-R81C did so only partially, with no effect on $P_{vr}$ (Fig. 6A–D). As regards spontaneous synaptic signaling, we detected a slight reduction in mEPSC amplitudes upon UFM1-depletion (Fig. 6E,F), which was not observed in earlier experiments (Fig. 3F–H). This difference likely originates from the different maturation states of the neurons, as it appeared that the differences between UFM1-KO and control neurons increased as cultures matured (Appendix Fig. S13). In any case, the expression of UFM1-WT or UFM1-R81C almost fully restored mEPSC amplitudes and frequencies to WT levels (Fig. 6E–G). Synaptic plasticity was similar in all four conditions tested, as revealed by measuring synaptic responses to high-frequency stimulation (Fig. 6H) or paired-pulse ratios (Fig. 6I). Similar to the $RRP_{sucrose}$, the defects in $RRP_{40Hz}$ (Fig. 6J, see also Fig. 3O) were fully rescued by expression of UFM1-WT, while UFM1-R81C only partially rescued this parameter. As before (Fig. 3Q), the RRP refilling rate remained unaltered (Fig. 6K). Overall, these results show that re-expression of UFM1-WT can fully restore the morphological and functional neuronal defects caused by UFM1-ablation, whereas UFM1-R81C has only partial rescue potential.

## Expression of UFM1-R81C only partially rescues protein translation and aggravates the ER stress response

We next performed UFM1-KO-rescue experiments to assess whether the UFM1-R81C variant affects protein translation (Fig. 7). While expression of UFM1-WT fully restored protein translation, expression of UFM1-R81C restored translation to only 76% of the control level (Fig. 7A,B). As UFMylation has so far mainly been implicated in ER homeostasis, we next investigated several key ER-associated features, including activation of the unfolded-protein-response (UPR), ER membrane fluidity, ER accumulation of KDEL-containing proteins, and ER response to stress induced by thapsigargin (Tg).

The UPR reprograms cellular gene expression and translation upon proteotoxic stress. We first quantified the expression levels of the three main sentinels of the UPR, ATF6, IRE1α, and PERK, by Western blotting (Fig. 7C–F). The levels of cleaved and activated ATF6 and of IRE1α were not significantly changed by UFM1-depletion or re-expression of UFM1-WT or UFM1-R81C (Fig. 7C–E). However, PERK levels were increased by 67% in UFM1-depleted cells as compared to controls (Fig. 7C,F), indicating an increase in ER stress and PERK-UPR pathway activation. Interestingly, re-expression of UFM1-WT reversed this phenotype, while UFM1-R81C had only a partial effect, leading to intermediate PERK levels (Fig. 7F). Although not significant, the levels of phosphorylated eIF2α (p-eIF2α)—a downstream target of PERK—showed trends of changes akin to the changes seen in PERK expression levels (Fig. S14A,B).

Beyond this, treatment with thapsigargin, an inhibitor of the ER Ca2+ ATPase (SERCA) and an ER stressor (Kettel and Karagoz, 2024; Munro and Pelham, 1987; Yamamoto et al, 2003), led to an increase in PERK levels in all four conditions (CRE, RFP, WT and R81C), while ATF6 and IRE1α levels were unaltered (Appendix Fig. S14C–L). Of note, in our experimental settings, these changes

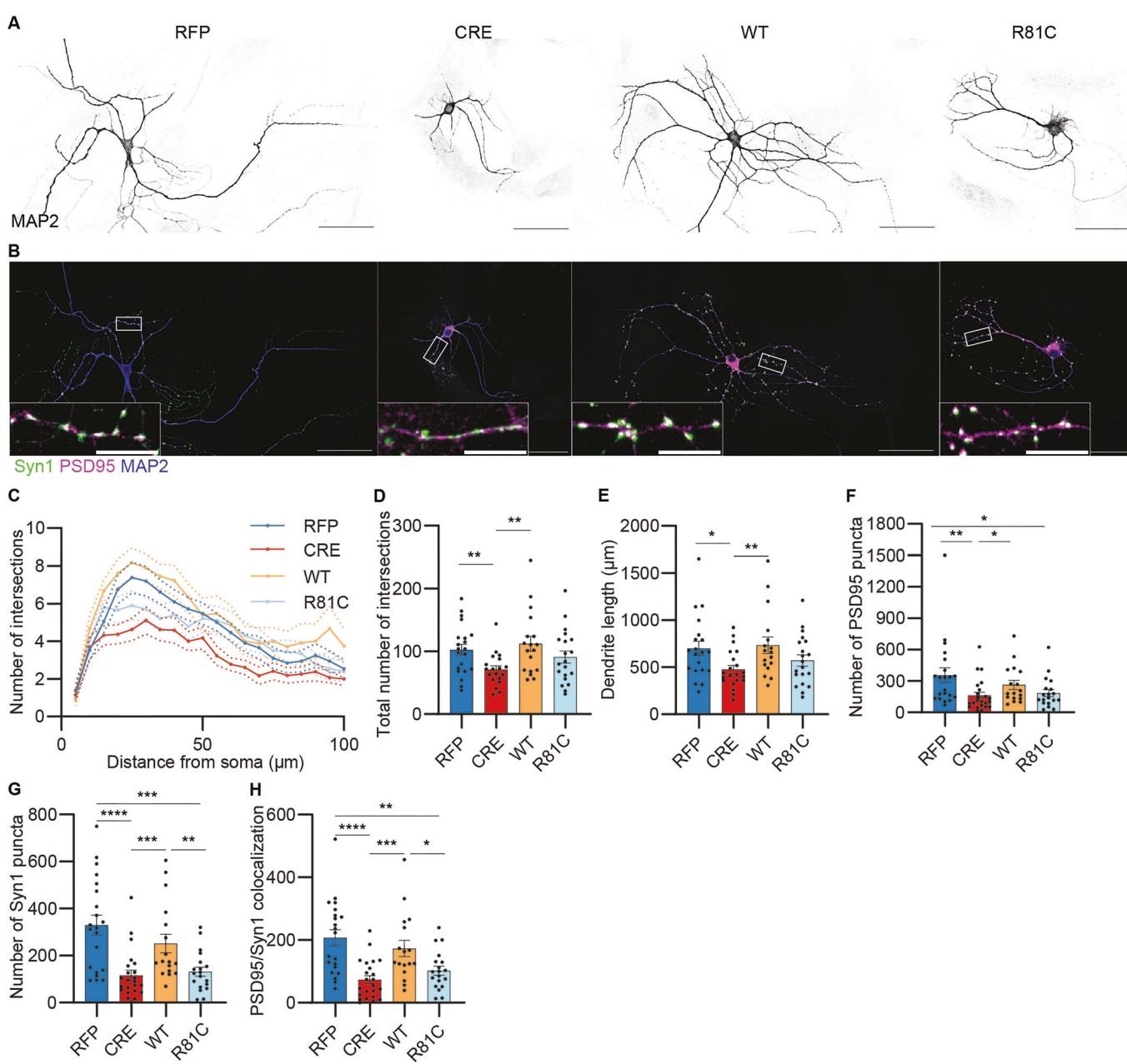

**Figure 5.** **Neuronal expression of UFM1-R81C only partially rescues UFM1-depletion-induced defects in neuronal complexity and synapse number.**

(A, B) Representative images of DIV 13 primary UFM1-cKO hippocampal neurons, infected at DIV 1 with viruses expressing either RFP, CRE, CRE combined with UFM1-WT (WT), or UFM1-R81C (R81C), fixed and immunolabelled for MAP2 (blue), Synapsin 1 (Syn1, green), and PSD95 (magenta). Representative images of MAP2 immunostaining (black) are presented in (A), while corresponding triple immunostained neurons are depicted in (B). (B) White frame insets indicate the enlarged view below. Scale bars, 50 μm, 10 μm (insets). (C) Sholl analysis lines graph depicting the average number of intersections between dendrites and concentric circles radiating every 5 μm from the soma, using traces as shown in (A). Dotted lines indicate averages ± SEM. (D) Bar graph showing the total number of dendritic intersections, obtained from the Sholl analysis in (C). **$p_{RFP-CRE}$ = 0.0065, **$p_{CRE-WT}$ = 0.0056 using a D'Agostino–Pearson normality test and Mann–Whitney test. (E) Bar graph showing the total dendrite lengths, obtained from traces in (A). **$p_{CRE-WT}$ = 0.0095, *$p_{RFP-CRE}$ = 0.0158, using a D'Agostino–Pearson normality test and Mann–Whitney test. (F) Bar graph depicting the total number of PSD95 in neurons as depicted in (B). **$p_{RFP-CRE}$ = 0.0019 *$p_{RFP-RC}$ = 0.0197, *$p_{CRE-WT}$ = 0.0184 using a D'Agostino–Pearson normality test and Mann–Whitney test. (G) Bar graph depicting the total number of Synapsin1 in neurons as depicted in (B). ****$p_{RFP-CRE}$ = 0.00001664, ***$p_{RFP-RC}$ = 0.0003 ***$p_{CRE-WT}$ = 0.0005, **$p_{WT-RC}$ = 0.0093 using a D'Agostino–Pearson normality test and Mann–Whitney test. (H) Bar graph depicting the total number of PSD95/Synapsin1 colocalized puncta in neurons as depicted in (B). ****$p_{RFP-CRE}$ = 0.00002026, ***$p_{CRE-WT}$ = 0.0009, **$p_{RFP-RC}$ = 0.0025, *$p_{WT-RC}$ = 0.0318 using a D'Agostino–Pearson normality test and Mann–Whitney test. For (C–H), data were obtained from $N$ = 4 independent experiments, $n$ = 21 RFP, 23 CRE, 17 UFM1-WT (WT), and 19 UFM1-R81C (R81C) neurons. Bar graphs show averages ± SEM. Source data are available online for this figure.

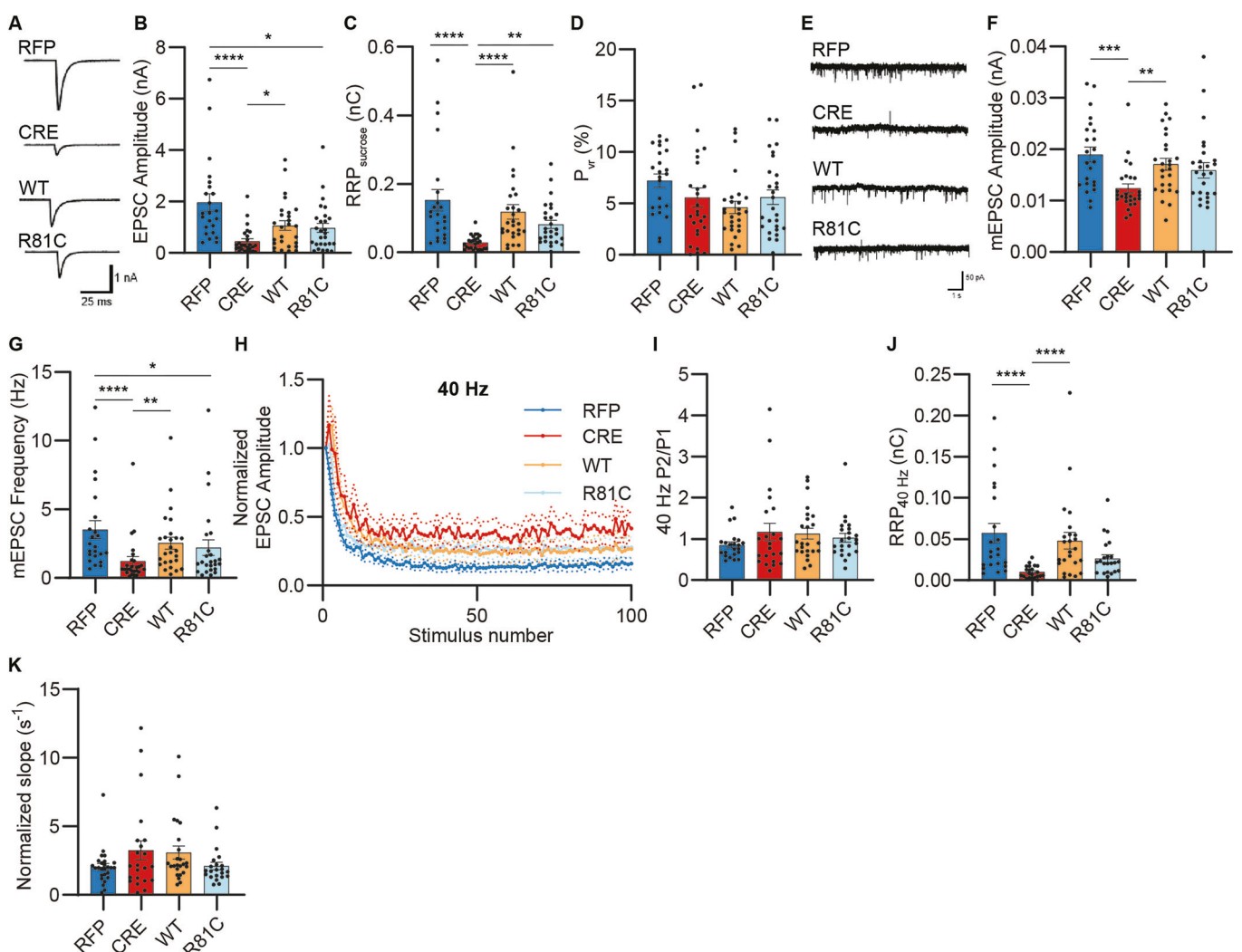

**Figure 6. Neuronal expression of UFM1-R81C only partially rescues UFM1-depletion-induced defects in synaptic transmission.**

(A) Representative current traces of evoked excitatory postsynaptic currents (EPSC) of DIV 10–12 primary UFM1-cKO hippocampal autaptic neurons infected with RFP, CRE, and CRE combined with UFM1-WT (WT) or UFM1-R81C (R81C). (B–D) Bar graphs depicting EPSC amplitude (B), the charge transferred by the release of the readily releasable pool of synaptic vesicles upon treatment of hypertonic sucrose (RRP$_{sucrose}$) (C), and the vesicular release probability (P$_{vr}$) (D), obtained from traces in (A). Data were obtained from $N = 4$ experiments, $n = 32$ RFP, 35 CRE, 38 WT, and 34 R81C cells. Normality was tested using a D'Agostino–Pearson normality test. For (B), data were compared using a Kruskal–Wallis test where **** p$_{RFP-CRE}$ = 0.000001297, * p$_{RFP-RC}$ = 0.0163, * p$_{CRE-WT}$ = 0.0030. In (C), data were compared using a Kruskal–Wallis test where ****p$_{RFP-CRE}$ = 0.0000003008, ****p$_{CRE-WT}$=0.000001113, **p$_{CRE-RC}$ = 0.0145. In (D), data were compared using a Kruskal–Wallis test. Bar graphs show averages ± SEM. (E) Representative traces of spontaneous miniature EPSC (mEPSC) of DIV 10–12 autaptic neurons infected with RFP, CRE, CRE combined with UFM1-WT (WT) or UFM1-R81C (R81C) expressing viruses. (F, G) Bar graphs depicting mEPSC amplitude (F) and frequency (G). Data were obtained from $N = 4$ experiments, $n = 30$ RFP, 34 CRE, 37 WT, and 32 R81C cells. Normality was tested using a D'Agostino–Pearson normality test. For (F), data were compared using a Kruskal–Wallis test, where ***p$_{RFP-CRE}$ = 0.0013, ** p$_{CRE-WT}$ = 0.0013. In (G), data were compared using a Kruskal–Wallis test where **** p$_{RFP-CRE}$ = 0.00002308, * p$_{RFP-RC}$ = 0.0292, ** p$_{CRE-WT}$ = 0.0008. Bar graphs show averages ± SEM. (H) Line graph depicting the average normalized EPSC amplitudes during 40 Hz stimulation trains of. Data were obtained from $N = 4$ experiments, $n = 31$ RFP, 31 CRE, 35 WT, and 30 R81C cells. Dotted lines indicate averages ± SEM. (I) Bar graph showing paired-pulse ratios (P2/P1) obtained from (H). Data were obtained from $N = 4$ experiments, $n = 31$ RFP, 31 CRE, 35 WT, and 30 R81C cells. Data were compared using a D'Agostino–Pearson normality test and a Kruskal–Wallis test. Bar graphs show averages ± SEM. (J) Bar graph showing the size of the readily releasable pool (RRP$_{40\ Hz}$), estimated by back extrapolation of the cumulative EPSC curve of the 40 Hz stimulus trains. Data were obtained from $N = 4$ experiments, $n = 31$ RFP, 31 CRE, 35 WT, and 30 R81C cells. Data were compared using a D'Agostino–Pearson normality test and a Kruskal–Wallis test where **** p$_{RFP-CRE}$=0.00002805, **** p$_{CRE-WT}$=0.000008545. Bar graphs show averages ± SEM. (K) Bar graph depicting the RRP refilling rate (slope in H) normalized to the RRP$_{40\ Hz}$. Data were obtained from $N = 4$ experiments, $n = 26$ RFP, 22 CRE, 24 WT, and 22 R81C cells. Data were compared using a D'Agostino–Pearson normality test and a Kruskal–Wallis test. Bar graphs show averages ± SEM. Source data are available online for this figure.

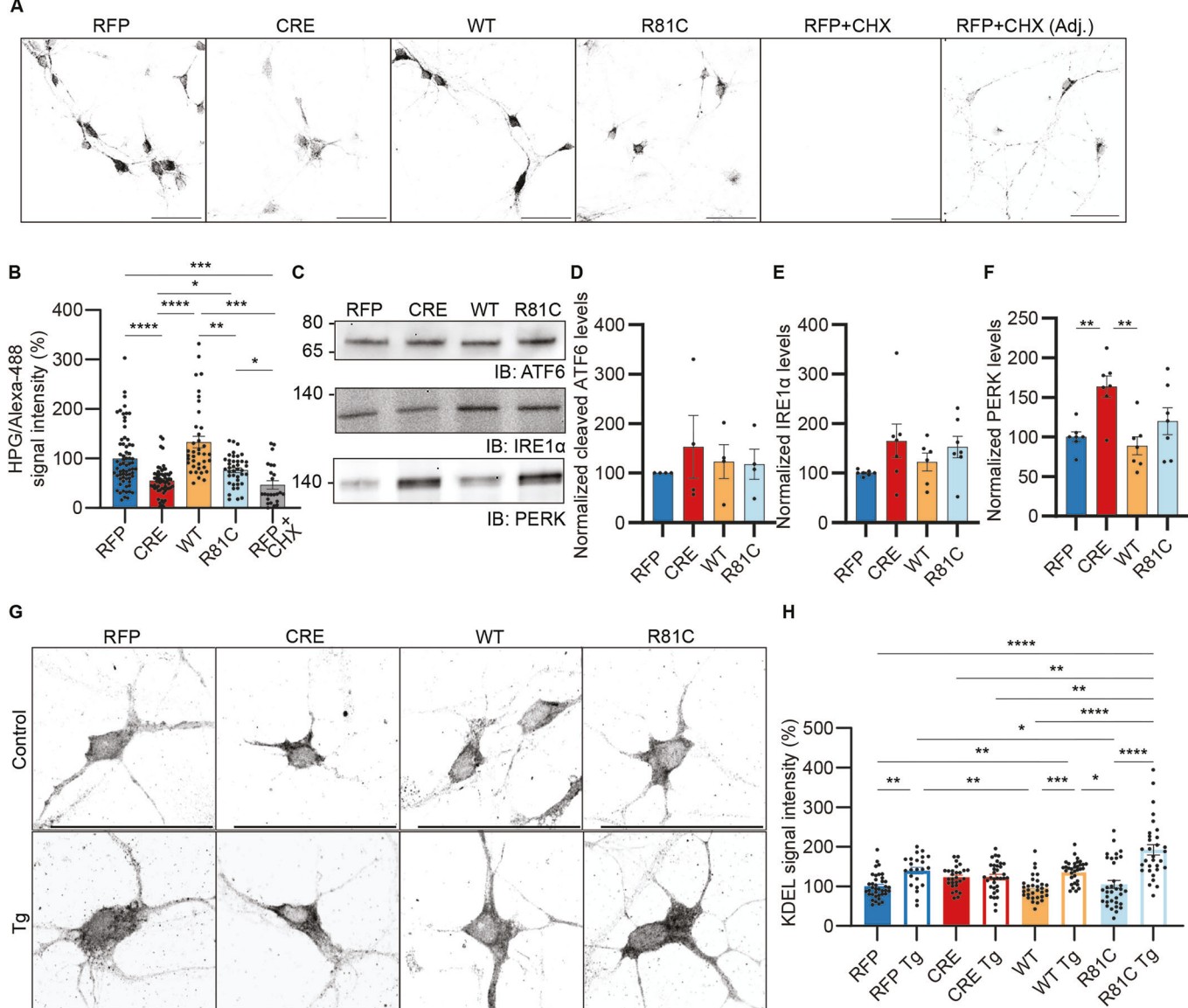

**Figure 7. Neuronal expression of UFM1-R81C only partially restores UFM1-depletion-induced defects in protein translation and UPR-PERK activation.**

(A) Representative images of HPG/Alexa-488 labeled primary hippocampal UFM1-cKO neurons (black) infected at DIV 1 with viruses expressing either RFP, CRE, CRE combined with UFM1-WT (WT) or UFM1-R81C (R81C) and processed for FUNCAT analysis at DIV 10. CHX, Cycloheximide; Adj., adjusted displayed intensity. Scale bar, 50 μm. (B) Bar graph depicting the quantification of the HPG/Alexa-488 signal intensity as shown in black in (A), expressed as a percentage of the HPG/Alexa-488 signal intensity of RFP control neurons. Data were obtained from $N = 3$ experiments, $n = 67$ RFP, 60 CRE, 38 WT, 40 R81C, and 24 RFP + CHX neurons. ****$p_{RFP-CRE} = 0.000008327$, ****$p_{CRE-WT} = 0.0000000001$, ****$p_{WT-RFP+CHX} = 0.0000000131$, ***$p_{RFP-RFP+CHX} = 0.0001$, *$p_{CRE-RC} = 0.0246$, **$p_{WT-RC} = 0.0046$, *$p_{RC-RFP+CHX} = 0.0228$ using a D'Agostino–Pearson normality test and Kruskal–Wallis test. Bar graphs show averages ± SEM. (C) Anti-ATF6, PERK, and IRE1α analysis of lysates from DIV 13 UFM1-cKO primary hippocampal neurons infected at DIV 1 with viruses expressing either RFP, CRE, CRE combined with UFM1-WT (WT) or UFM1-R81C (R81C). Molecular weights are indicated on the left (kDa). (D) Bar graph depicting the quantification of cleaved ATF6 IB signal as shown in (C). $N = 4$ experiments with $n = 3$. Data were analyzed using a D'Agostino–Pearson normality test and Kruskal–Wallis test. Bar graphs show averages ± SEM. (E) Bar graph depicting the quantification of IRE1α IB signal as shown in (C). $N = 4$ experiments, $n = 6$. Data were analyzed using a D'Agostino–Pearson normality test and Kruskal–Wallis test. Bar graphs show averages ± SEM. (F) Bar graph depicting the quantification of PERK IB signal as shown in (C). $N = 4$ experiments, $n = 6$. **$p_{RFP-CRE} = 0.0079$, **$p_{CRE-WT} = 0.0017$ using a D'Agostino–Pearson normality test and Ordinary one-way ANOVA. Bar graphs show averages ± SEM. (G) Representative images of DIV 5 primary hippocampal UFM1-cKO neurons infected at DIV 1 with viruses expressing either RFP, CRE, CRE combined with UFM1-WT (WT) or UFM1-R81C (R81C) and treated with thapsigargin (Tg, 3 μM, 15 min, bottom panels) or DMSO (top panels) as control. Neurons were fixed at DIV 5 and immunolabelled with anti-KDEL antibody (black). Scale bar, 50 μm. Data were obtained from N = 3 experiments, n = 38 RFP, 28 CRE, 32 UFM1-WT (WT), 34 UFM1-R81C (R81C), 23 RFP Tg, 34 CRE Tg, 33 UFM1-WT Tg (WT Tg), and 31 UFM1-R81C Tg (R81C Tg) neurons. (H) Bar graph depicting the quantification of the KDEL signal intensity in control and Tg-treated neurons, as shown in (G), normalized against the anti-KDEL signal intensity of RFP and expressed as a percentage. Data were obtained from $N = 3$ experiments, $n = 38$ RFP, 28 CRE, 32 UFM1-WT (WT), 34 UFM1-R81C (R81C), 23 RFP Tg, 34 CRE Tg, 33 UFM1-WT Tg (WT Tg), and 31 UFM1-R81C Tg (R81C Tg) neurons. ****$p_{RFP-RC\ Tg} = 0.0000000041$; ****$p_{WT-RC\ Tg} = 0.0000000016$; ****$p_{RC-RC\ Tg} = 0.0000002162$; ***$p_{WT-WT\ Tg} = 0.0004$; **$p_{RFP-RFP\ Tg} = 0.0071$; **$p_{RFP-WT\ Tg} = 0.0011$; **$p_{CRE-RC\ Tg} = 0.0046$; **$p_{WT-RFP\ Tg} = 0.0027$; **$p_{CRE\ Tg-RC\ Tg} = 0.0044$; *$p_{RC-RFP\ Tg} = 0.0464$; *$p_{RC-WT\ Tg} = 0.0117$ using a D'Agostino–Pearson normality test and Kruskal–Wallis test. Bar graphs show averages ± SEM. Source data are available online for this figure.

were only visible at the protein levels, as mRNA levels of the three sentinels remained unchanged, even upon thapsigargin treatment (Appendix Fig. S14G–J). Importantly, thapsigargin treatment led to the efficient stress-induced cleavage of *Xbp1* in all four conditions, validating our ER stress approach (Appendix Fig. S14J–L). In sum, our data indicate that UFM1-loss causes ER stress and the activation of the PERK-UPR pathway, and that this effect is fully rescued by UFM1-WT but not by UFM1-R81C.

The UPR is activated by the accumulation of misfolded proteins inside the ER, which impacts ER lipid composition and membrane fluidity (Kettel and Karagoz, 2024). To explore corresponding effects of UFM1-loss and re-expression, we used Fluorescence Recovery After Photobleaching (FRAP) of the ER-tracker BODIPY-Glibenclamid to assess ER-membrane dynamics (Appendix Fig. S15). We did not detect any differences in BODIPY-Glibenclamid FRAP between the four conditions tested (Appendix Fig. S15A), but did observe a minor but significant decrease in the total intensity levels after 5 min of bleaching in UFM1-depleted neurons, which was rescued by re-expression of either UFM1-WT or UFM1-R81C (Appendix Fig. S15B, C). These results indicate that ER membrane features are not dramatically affected by UFM1-depletion or expression of UFM1-R81C and the consequent UPR induction.

We then investigated the effects of UFM1-loss and UFM1-R81C expression on ER-homeostasis and ER stress via an alternative and indirect approach, measuring the ER accumulation of KDEL-containing proteins. Many ER proteins, including key chaperones involved in ER homeostasis and UPR activation, contain a KDEL sequence whose binding to KDEL receptors in the Golgi apparatus activates the retrograde transport of KDEL-containing proteins back to the ER, contributing to quality control in the ER and maintains ER homeostasis (Majoul et al, 2001; Yamamoto et al, 2003). In view of this, we tested whether the perturbations in ER homeostasis caused by UFM1-loss or the UFM1-R81C variant are paralleled by changes in the subcellular distribution of KDEL-containing proteins. Furthermore, we investigated the ER stress response to thapsigargin and monitored somatic KDEL signal intensity. Control cells showed an enhancement of somatic KDEL levels in response to thapsigargin, indicating the induction of ER stress. In contrast, naive UFM1-KO neurons already showed elevated KDEL levels as compared to controls, and no further effect of thapsigargin, indicating that UFM1-loss already caused ER stress (Fig. 7G,H). The rescue expression of either UFM1-WT or UFM1-R81C restored KDEL signal intensity to that of control cells, but the response to thapsigargin in rescued cells was much higher upon UFM1-R81C expression than with UFM1-WT (Fig. 7G,H), indicating that UFM1-R81C exacerbates the ER stress response. Overall, these data indicate that UFM1-loss and the expression of the UFM1-R81C variant lead to different pathophysiological responses to ER-stress.

## The UFM1-R81C variant affects UFM1 activation

To assess the effect of the R81C mutation on the UFMylation cascade, we performed in vitro UFMylation assays (Fig. 8A). We observed a moderate retardation in UFSP1-mediated UFM1 maturation, as measured by cleavage of UFM1-GFP fusion construct (Millrine et al, 2022), with 50% of UFM1-WT-GFP being cleaved after 6 min, as compared to 15 min for UFM1-R81C-GFP (Fig. 8B,C). Further, we observed drastically reduced UBA5-mediated activation of UFM1-R81C as compared to UFM1-WT (Fig. 8D,E). Indeed, while 50% of UBA5 formed a thioester bond with UFM1 (UBA5 ~ UFM1) after 1 min, a comparable level of UBA5 ~ UFM1-R81C was not even reached after 30 min. In contrast, the efficacy of transfer of UFM1-R81C from UBA5 to UFC1 was similar to that of UFM1-WT within the time frame of our assay (Fig. 8F,G), indicating that once UFM1-R81C is activated by the E1 enzyme, UFMylation can progress normally. Importantly, size-exclusion chromatography yielded similar elution profiles for UFM1-WT and UFM1-R81C, indicating that the R81C mutant is soluble and does not affect the oligomeric state of UFM1 (Fig. 8H). Altogether, these results indicate that the UFM1-R81C variant causes aberrant UFMylation dynamics.

## Trazodone treatment rescues protein translation in neurons expressing UFM1-R81C

Our analysis of UFM1-deficient neurons identified a characteristic set of morphological and functional phenotypic changes, including defects in dendrite complexity and synapse numbers, impaired protein translation, and aberrant cell stress responses, involving excessive activation of the PERK-UPR pathway (Figs. 1–4 and 7). Comparative analyses of UFM1-WT and the encephalopathy-linked UFM1-R81C variant showed that several of these phenotypic changes are not or only partially rescued by the UFM1-R81C variant (Figs. 5–7). Interestingly, Trazodone, a clinically used antidepressant, was shown previously to improve protein translation and synaptogenesis in neurons (Albert-Gasco et al, 2024; Goncalo and Vieira-Coelho, 2021; Halliday et al, 2017b), and to prevent PERK-UPR pathway activation in the brain (Halliday et al, 2017a). This pharmacological profile prompted us to test whether Trazodone can alleviate the defects caused by UFM1-loss or UFM1-R81C expression. To this end, we treated cultured neurons at DIV 3, DIV 6, and DIV 9 with Trazodone (Fig. 9A), and analyzed protein translation, activation of the PERK-UPR pathway, and neuronal morphology.

Trazodone alleviated the UFM1-KO-induced increase in p-eIF2α levels, validating our methodology (Fig. 9B,C). Of note, p-eIF2α levels were not changed in control neurons, indicating that Trazodone does not influence the PERK-UPR pathway under resting conditions (Fig. 9B,C). Strikingly, Trazodone treatment fully restored translation to control levels in UFM1-R81C-expressing cells, but did so only partially in UFM1-KO cells (Fig. 9D,E). Finally, we examined whether the Trazodone-mediated rescue of protein translation is also sufficient to mitigate the morphological defects seen upon UFM1 depletion or expression of UFM1-R81C. Sholl analyses of dendrite complexity showed that Trazodone treatment does not improve the defects in dendrite complexity seen upon UFM1 depletion or expression of UFM1-R81C (Fig. 10A,B; Appendix Fig. S16). As regards synapse numbers, Trazodone treatment did not affect the number of synaptic puncta in control neurons. Although the values remained below those of control neurons, Trazodone treatment doubled the number of synapses in UFM1-R81C-expressing neurons and led to a robust increase in synapse numbers in UFM1-KO neurons (Fig. 10C–E).

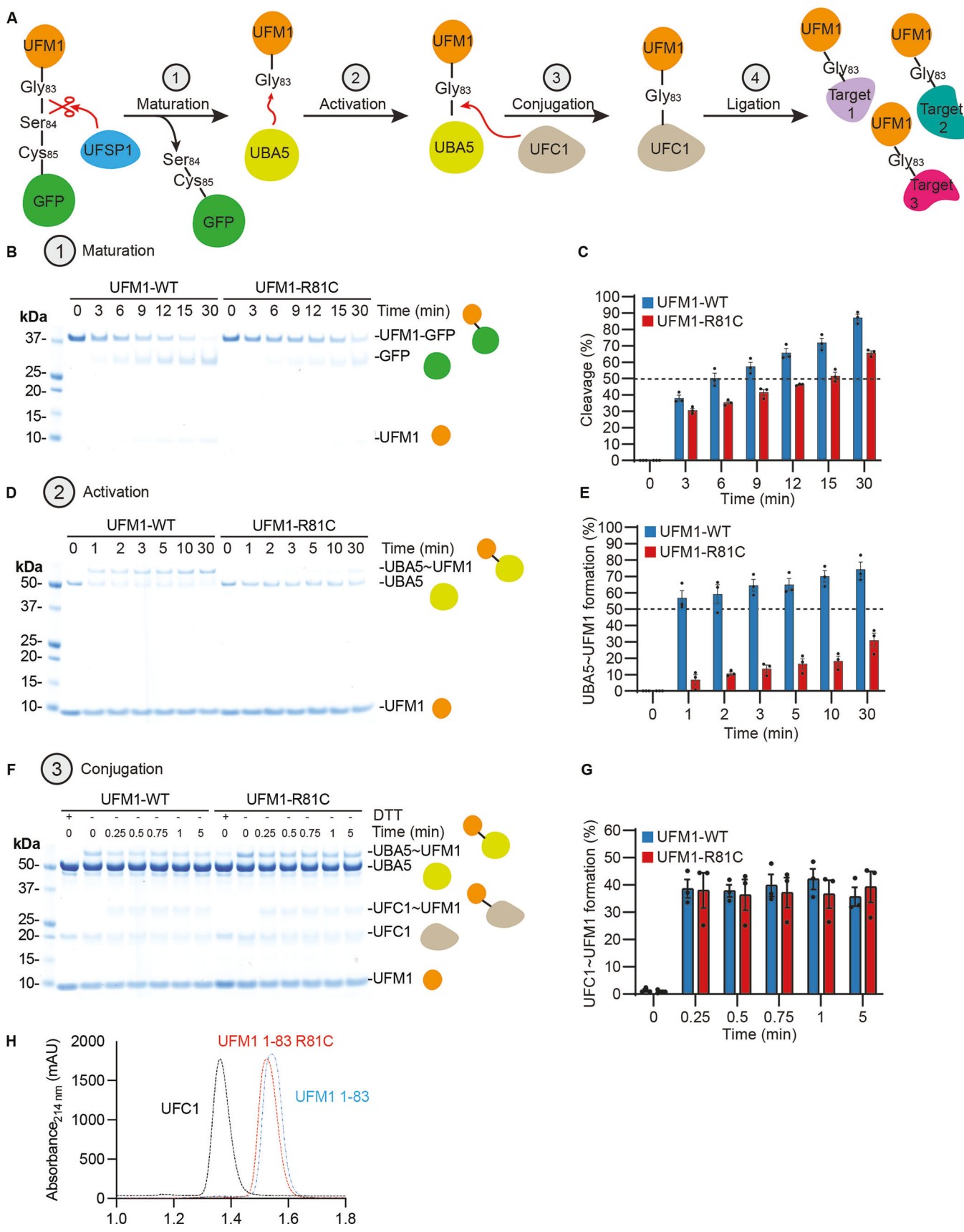

◀ **Figure 8. In vitro UFMylation assay assessing the maturation, E1 charging, and E2 transfer of UFM1-WT and UFM1-R81C.**

(A) Scheme of the UFMylation process. Numbers in gray circles indicate the UFMylation steps. A GFP-UFM1 construct was used to facilitate the detection of the cleavage by UFSP1, reflecting the maturation of UFM1 (1). Mature UFM1 is activated by the E1 enzyme UBA5 (2), and conjugated to the E2 enzyme UFC1 (3), before it is ligated to various target proteins (4). (B) Coomassie-stained gel depicting time-dependent maturation of UFM1-WT-GFP (left) or UFM1-R81C-GFP (right). The UFM1-WT/R81C-GFP (37 kDa, top band) constructs were cleaved by UFSP1, generating a mature UFM1-WT/R81C band (10 kDa, lower band) and releasing GFP (30 kDa, middle band). The reactions were stopped at the indicated time by the addition of loading dye and run on a 4–12% Bis-Tris gel under reducing conditions. (C) Bar graph showing the quantification of the disappearance of the UFM1-WT-GFP (blue) or UFM1-R81C-GFP (red) bands upon cleavage by UFSP1 over time by determination of the UFM1-WT/R81C-GFP bands signal intensity, subtracted from the corresponding intensity at time point 0 and expressed as %. Data were obtained from $N = 3$ independent experiments. The bar graph shows average ± SD. (D) Coomassie-stained gel depicting time-dependent charging of UBA5 with UFM1-WT (left) or UFM1-R81C (right). UBA5 (50 kDa, middle top band) charged with UFM1-WT or UFM1-R81C (10 kDa, bottom band) results in the formation of a thioester-linked UBA5 ~ UFM1 conjugate (top band). The reactions were stopped at the indicated time by the addition of loading dye and run on a 4–12% Bis-Tris gel under non-reducing conditions. (E) Bar graph depicting the quantification of UBA5 charging with UFM1-WT (blue) or UFM-R81C (red) over time determined by ratiometric measurement between the UBA5 ~ UFM1-WT/R81C bands signal intensity over the UBA5 band signal intensity at time point 0 and expressed as %. Data were obtained from $N = 3$ independent experiments. The bar graph shows average ± SD. (F) Coomassie-stained gel depicting time-dependent transfer of UFM1-WT (left) or UFM1-R81C (right) from UBA5 to UFC1. UFM1 (10 kDa, bottom band) is transferred from UBA5 (50 kDa, second band from top) to UFC1 (20 kDa, second band from bottom), resulting in the formation of a thioester-linked UFC1 ~ UFM1 conjugate (30 kDa, middle band). The reactions were stopped at the indicated time above the gel image by the addition of loading dye and run on a 4–12% Bis-Tris gel under non-reducing conditions. (G) Bar graph showing the quantification of UFC1 charged with WT (blue) or UFM1-R81C (red) over time determined by ratiometric between UFC1 ~ UFM1 over (UFC1 + UFC1 ~ UFM1) (E). Data were obtained from $N = 3$ independent experiments. The bar graph shows average ± SD. (H) Size-exclusion chromatography elution profile of mature UFM1-WT, UFM1-R81C, and UFC1. 0.1 mg of protein was run on a Superdex™ 75 increase 3.2/300 column. Source data are available online for this figure.

# Discussion

## UFM1 in neuron development and synaptic signaling

UFM1 and its conjugation machinery are ubiquitously expressed, and downregulation of the UFMylation pathway in vivo results in multiple neurodevelopmental defects (Briere et al, 2021; Cabrera-Serrano et al, 2020; Colin et al, 2016; Daida et al, 2018; Duan et al, 2016; Hamilton et al, 2017; Ivanov et al, 2022; Muona et al, 2016; Nahorski et al, 2018; Pan et al, 2023; Szűcs et al, 2021; Zhang et al, 2022), indicating that UFMylation has a fundamental role in neuronal development and function. We found that UFM1-depletion in neurons causes strong morphological defects both in vivo and in vitro, including a strong reduction in synapse number that results in severe defects in synaptic signaling (Figs. 1–3). At the cellular level, UFM1-loss causes decreased protein translation, excessive activation of the PERK-UPR pathway, and protein accumulation in the ER (Figs. 4 and 7), ultimately reflecting perturbed ER homeostasis. Importantly, rescue expression of UFM1-WT reverses all these defects, establishing UFM1 as a key regulatory factor in neuronal development, morphogenesis, and function (Figs. 5, 6, and 7).

The decrease in protein translation and the perturbed ER homeostasis are likely the major causes for the defects in neuronal development, synapse formation, and synapse function seen upon UFM1 loss as all these processes require very substantial protein biosynthesis activity (Borisova et al, 2024; Vásquez et al, 2022; Wang et al, 2012). Indeed, most UFM1 targets described so far are associated with the ER, including the ribosomal protein RPL26 as the best characterized UFM1 target (DaRosa et al, 2024; Makhlouf et al, 2024; Scavone et al, 2023; Walczak et al, 2019). UFMylation of RPL26 regulates ribosome recycling after translation termination and ER-RQC, so that UFM1 depletion is expected to result in the dysregulation of this pathway and in resolving stalled ribosomes at the ER, ultimately perturbing protein translation and consequently impacting neuronal development and synapse formation (DaRosa et al, 2024; Endo et al, 2023; Makhlouf et al, 2024; Scavone et al, 2023; Walczak et al, 2019; Wang et al, 2020).

In addition, a number of non-ER-associated UFM1 targets have been described in non-neuronal cells. These are involved in DNA-repair mechanisms, programmed cell death, and the regulation of mitochondrial function (Liu et al, 2020; Qin et al, 2019; Wang et al, 2019; Yoo et al, 2014; Zhu et al, 2022). Alterations in these pathways in neurons could also contribute to the phenotype we observed in UFM1-depleted neurons.

Our comparative Western blot analysis of WT and UFM1-KO neuronal lysates revealed numerous UFM1-specific bands (Fig. 2), indicating that, as yet unidentified, neuronal proteins involved in the regulation of neuronal cell morphology and function might be UFM1 targets. For a first test of this notion, we employed anti-UFM1 immuno-affinity purification and mass spectrometric analysis of purified proteins to identify UFM1 target candidates in adult mouse brain (Dataset EV2; Appendix Fig. S17). This approach revealed around ~300 candidates for putative novel neuronal UFM1 targets (Dataset EV2), including UBA5 and UFC1, validating our methodology. Strikingly, several non-ER proteins were identified, including several guanyl-nucleotide exchange factors (GEFs). In alignment with this finding, our RNA-seq analysis revealed that genes involved in guanyl ribonucleotide binding were strongly downregulated in UFM1-depleted neurons (Molecular function terms, Appendix Fig. S7). The most prominent protein candidate target, and a particularly interesting one, is DOCK4 (Appendix Fig. S17C), whose deletion is associated with neurodevelopmental delay and microcephaly, i.e., phenotypes that are also observed in patients with mutations linked to impaired UFMylation (Herbst et al, 2024). While the UFMylation pattern of these candidates remains to be validated, our findings provide an informative basis for future investigation of UFMylation in neurons.

## Pathogenic mechanisms of UFMylopathies

Genetic studies have been of key importance in identifying genetic causes of encephalopathies (Hamdan et al, 2017; Morrison-Levy et al, 2021; Niemi et al, 2018), thereby paving the way to the development of disease-specific therapies based on the affected

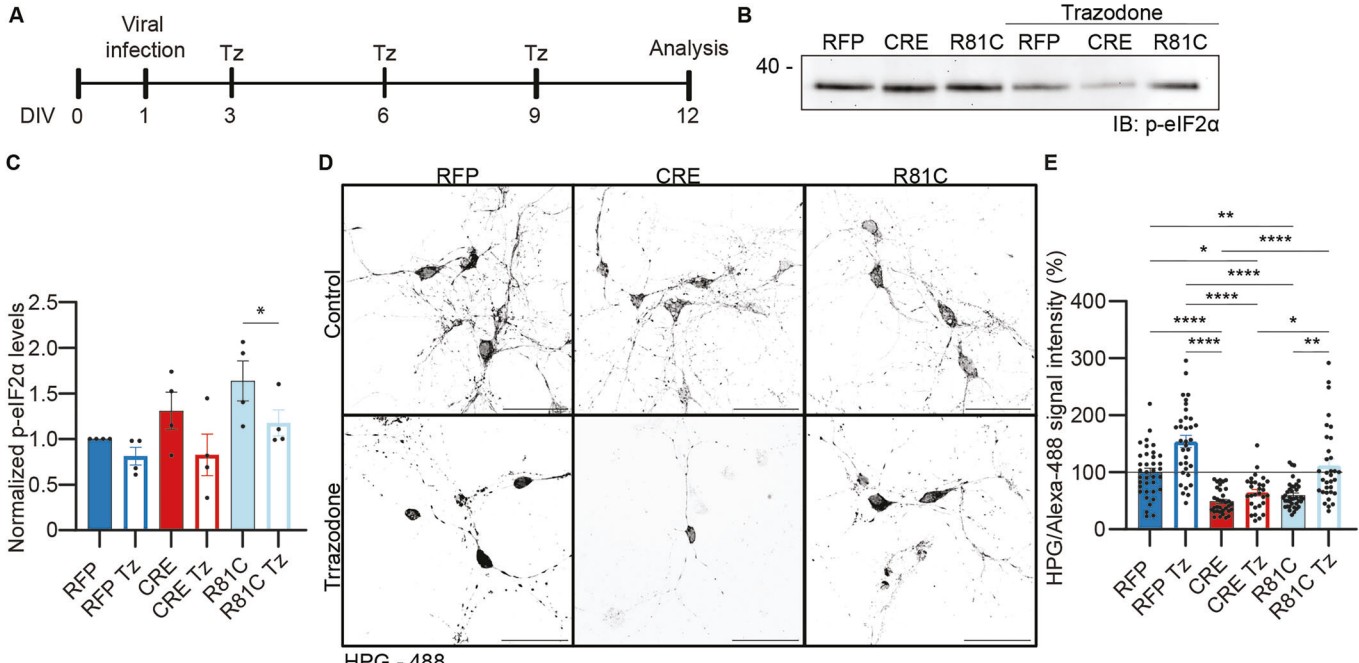

**Figure 9. Trazodone treatment rescues protein translation in UFM1-R81C expressing neurons.**

(A) Schematic of the experimental approach for Trazodone treatment of primary hippocampal neurons. DIV 0 indicates the day of dissection. DIV, days in vitro, Tz, Trazodone. (B) Anti-p-eIF2α immunoblot (IB) analysis of DIV 12 primary UFM1-cKO hippocampal neuron lysates. Neurons were infected on DIV 1 with RFP, CRE, or CRE combined with UFM1-R81C expressing viruses, treated with Trazodone (Tz, 20 μM) or DMSO as a control every 3 days. Molecular weights are indicated on the left (kDa). (C) Bar graph depicting the quantification of p-eIF2α immunosignal in (B), normalized to the total protein stain and expressed as a percentage of RFP. $N = 4$ experiments, $n = 4$, *$P = 0.0466$ using a D'Agostino–Pearson normality test and Paired $t$ test. Bar graphs show averages ± SEM. (D) Representative images of HPG/Alexa-488 labeled primary hippocampal UFM1-cKO neurons infected at DIV 1 with RFP, CRE, or CRE combined with UFM1-R81C expressing viruses, treated with Trazodone (Tz, 20 μM) or DMSO as a control every 3 days, and processed for FUNCAT analysis after DIV 12. Scale bar, 50 μm. (E) Bar graph depicting the quantification of the HPG/Alexa-488 signal intensity in (D). $N = 3$ independent experiments, $n = 37$ RFP control, 34 CRE control, 38 UFM1-R81C (R81C) control, 34 RFP Tz, 29 CRE Tz, and 33 UFM1-R81C Tz (R81C Tz) neurons. ****$p_{RFP-CRE} = 0.000003363$, ****$p_{CRE-RC\ Tz} = 0.000004554$, ****$p_{RC-RFP\ Tz} = 0.0000000012$, ****$p_{RFP\ Tz-CRE\ Tz} = 0.0000006168$, ****$p_{CRE-RFP\ Tz} = 0.0000000001$, **$p_{RFP-RC} = 0.0011$, **$p_{RC-RC\ Tz} = 0.001331$, *$p_{RFP-CRE} = 0.0316$, *$p_{CRE\ Tz-RC\ Tz} = 0.03113$ using a D'Agostino–Pearson normality test and Kruskal–Wallis test. Bar graphs show averages ± SEM. Source data are available online for this figure.

genes and related processes. Likewise, studying the molecular processes affected by defective UFMylation will ultimately help with the development of specific therapies to treat UFMylation-linked encephalopathies, i.e., UFMylopathies. Several variants identified in *UFM1* itself or in genes encoding enzymes of the UFMylation machinery are linked to encephalopathies with overlapping features, including neurodevelopmental delay, microcephaly, and seizures. Most of these variants result in decreased levels of UFM1 and UFMylated targets (Briere et al, 2021; Cabrera-Serrano et al, 2020; Colin et al, 2016; Daida et al, 2018; Duan et al, 2016; Hamilton et al, 2017; Ivanov et al, 2022; Muona et al, 2016; Nahorski et al, 2018; Pan et al, 2023; Szűcs et al, 2021). For instance, a 3-bp deletion detected in the *UFM1* promoter of human patients with global developmental delay, epileptic encephalopathy, and hypomyelination (Hamilton et al, 2017; Szűcs et al, 2021), causes an 80% reduction in UFM1 promoter activity when tested in a neuroblastoma cell line, indicating that the corresponding patients have substantially decreased UFM1 levels (Hamilton et al, 2017; Ivanov et al, 2022). We used a conditional KO approach to mimic this genetic scenario in mouse hippocampal neurons (Fig. 2B,C), and found that UFM1-loss causes aberrant neuronal morphology and synaptic signaling (Figs. 1–3)—phenotypes that can explain the neurological defects seen in the affected

patients. Importantly, we identified potential pathogenic mechanisms at the cell biological level, including aberrations in the levels of UPR activators and in ER-stress (Figs. 4 and 7), which have been implicated in several neurodevelopmental and neurodegenerative disorders (Borisova et al, 2024; Godin et al, 2016; Lindholm et al, 2017; Passemard et al, 2019; Vásquez et al, 2022).

A second disease-linked variant of *UFM1*, which causes an R81C amino acid substitution and symptoms similar to those caused by UFM1 deficiency (Nahorski et al, 2018), appears to operate by a somewhat different pathogenic mechanism than UFM1 deficiency. We found that expression of UFM1-R81C only partially rescues the defects observed upon UFM1-KO (Figs. 5–7), indicating that UFM1-R81C has hypomorphic features. On the other hand, UFM1-KO and UFM1-R81C-expressing neurons displayed strikingly distinct ER-stress responses. Under resting conditions, UFM1-depleted neurons showed signs of ER-stress, while UFM1-R81C-expressing neurons did not, and UFM1-R81C-expressing neurons showed an aggravated ER-stress response to thapsigargin as compared to UFM1-KO and control cells (Fig. 7H).

In principle, the UFM1-R81C variant could affect neuronal UFMylation in multiple ways. Our data indicate that the R81C substitution retards UFM1 maturation and severely impairs E1-mediated UFM1 activation, without affecting UFM1 transfer to the

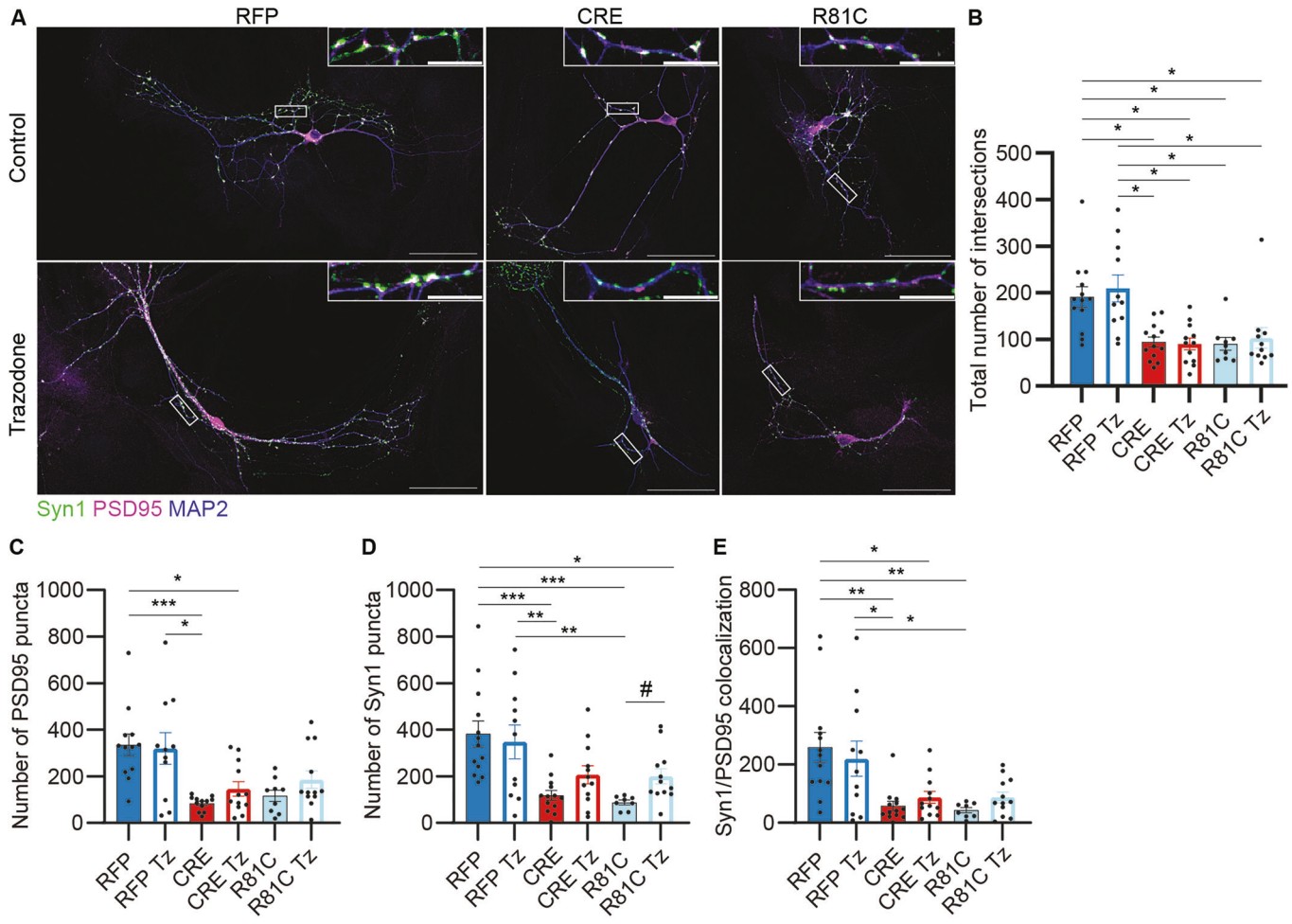

**Figure 10. Trazodone treatment leads to a partial rescue in the number of synaptic puncta in UFM1-R81C expressing neurons.**

(A) Representative images of DIV 12 UFM1-cKO primary hippocampal neurons infected at DIV 1 with RFP, CRE or CRE combined with UFM1-R81C expressing viruses, treated with Trazodone (Tz, 20 μM, bottom row) or DMSO (top row) as a control every 3 days, fixed and immunolabelled after DIV 12 for Synapsin1 (green), PSD95 (magenta) and MAP2 (blue). White frame insets indicate the region enlarged above. Scale bars, 50 μm, 10 μm (insets). (B) Sholl analysis bar graph depicting the average number of intersections between dendrites and concentric circles radiating every 5 μm from the soma. $*p_{RFP-CRE} = 0.0815$, $*p_{RFP-RC} = 0.0248$, $*p_{RFP-CRE\ Tz} = 0.0128$, $*p_{RFP-RC\ Tz} = 0.0248$, $*p_{CRE-RFP\ Tz} = 0.0227$, $*p_{RC-RFP\ Tz} = 0.0283$, $*p_{RFP\ Tz-CRE\ Tz} = 0.0158$, $*p_{RFP\ Tz-RC\ Tz} = 0.0292$ using a D'Agostino–Pearson normality test and Kruskal–Wallis test. (C) Bar graph depicting the quantification of the number of PSD95 in neurons as depicted in (A). $***p_{RFP-CRE} = 0.0009$, $*p_{RFP-CRE\ Tz} = 0.0361$, $*p_{CRE-RFP\ Tz} = 0.0271$ using a D'Agostino–Pearson normality test and Kruskal–Wallis test. (D) Bar graph depicting the quantification of the number of Synapsin1 puncta in neurons as depicted in (A). $***p_{RFP-CRE} = 0.0006$, $***p_{RFP-RC} = 0.0009$, $**p_{CRE-RFP\ Tz} = 0.0063$, $**p_{RC-RFP\ Tz} = 0.0063$, $*p_{RFP-RC\ Tz} = 0.0442$ using a D'Agostino–Pearson normality test and a Holm–Šídák's multiple comparisons test. $\#p_{RC-RC\ Tz} = 0.0137$ using an unpaired $t$ test. (E) Bar graph depicting the total number of PSD95/Synapsin1 colocalized puncta in neurons as depicted in (A). $**p_{RFP-CRE} = 0.0014$, $**p_{RFP-RC} = 0.0031$, $*p_{RFP-CRE\ Tz;\ RFP-RC\ Tz} = 0.0111$, $*p_{CRE-RFP\ Tz} = 0.0219$, $*p_{RC-RFP\ Tz} = 0.0285$ using a D'Agostino–Pearson normality test and a Holm–Šídák's multiple comparisons test. For (B–E), data were obtained from $N = 3$ independent experiments, $n = 12$ RFP control, 13 CRE control, 9 UFM1-R81C (R81C) control, 11 RFP Tz, 12 CRE Tz, and 12 UFM1-R81C Tz (R81C Tz) cells. Bar graphs show averages ± SEM. Source data are available online for this figure.

E2 (Fig. 8). These defects are likely to cause an imbalance in the cellular UFMylation dynamics rather than a simple reduction in UFMylation mimicking UFM1 loss. It appears that under resting conditions, UFMylation with UFM1-R81C is possible but not sufficient to maintain normal neuronal development (Fig. 7), whereas under ribosome stalling conditions, where UFM1 must be rapidly conjugated to substrates, the consequent imbalance in UFMylation dynamics caused by the R81C mutation exacerbates the ER stress response (Fig. 7G,H). Thus, while having hypomorphic features under physiological conditions, mutant UFM1-R81C may act as a dominant-negative or gain-of-function (toxic) mutant when the ER is challenged, a critical aspect to consider in

the context of UFM1-associated neurodevelopmental disorders and corresponding therapeutic approaches. The ultimate analysis of the pathogenic mechanisms by which UFM1-loss and the UFM1-R81C variant affect neurons will require knowledge of the full repertoire of UFM1 targets and of their alterations under the different pathogenic conditions.

## Towards therapies of UFMylopathies

As stated above, multiple genetic studies have linked *UFM1* variants or variants of the genes encoding components of the UFMylation cascade to encephalopathies (Briere et al, 2021;

Cabrera-Serrano et al, 2020; Colin et al, 2016; Daida et al, 2018; Duan et al, 2016; Hamilton et al, 2017; Ivanov et al, 2022; Muona et al, 2016; Nahorski et al, 2018; Pan et al, 2023; Szűcs et al, 2021). We explored two aspects of UFMylopathies, reduced UFMylation and neuronal expression of the UFM1-R81C variant, in order to identify molecular processes involved in the UFMylation-linked encephalopathies and to explore therapeutic approaches. We found that both UFM1-depletion and UFM1-R81C expression activate the PERK-UPR pathway, which led us to test for possible ameliorating effects of Trazodone, an antidepressant that can downregulate the PERK-UPR pathway (Albert-Gasco et al, 2024; Halliday et al, 2017b) (Fig. 9), on UFMylopathy-related phenotypes in mouse neurons. Indeed, Trazodone treatment mitigated the excessive PERK-UPR activation (Fig. 9). Beyond this, Trazodone treatment fully rescued protein translation in neurons expressing UFM1-R81C, but not in UFM1-depleted neurons, leading to an increase in synapse numbers (Figs. 9 and 10).

Trazodone was reported to increase the translation of proteins involved in synaptogenesis in vivo (Albert-Gasco et al, 2024; Halliday et al, 2017b). Therefore, it currently remains unclear why the phenotypes of UFM1-deficient and UFM1-R81C-expressing neurons are differentially affected by Trazodone. The possible residual UFMylation activity of the UFM1-R81C mutant may explain this differential response by maintaining ER homeostasis and thus taming the UPR. As we do not understand yet where Trazodone treatment and protein UFMylation intersect, alternative scenarios, yet to be determined, may have to be considered.

This uncertainty notwithstanding, we propose that therapies targeting protein translation and ER stress are relevant treatment principles for UFMylopathies. Several known drugs target ER-protein folding or UPR activators. The best studied ones in the disease context are chemical chaperones that prevent ER-stress by facilitating the folding of misfolded ER proteins, and small molecules that modulate UPR pathways (Marciniak et al, 2022; Rabouw et al, 2019). Future studies should focus on their potential to mitigate the defects of UFM1-depleted or UFM1-R81C-expressing neurons.

## Outlook—UFM1 beyond protein homeostasis

While most UFM1 targets identified so far are associated with protein translation and the ER, a growing number of non-ER-associated targets have been identified more recently (Liu et al, 2020; Qin et al, 2019; Wang et al, 2019; Yoo et al, 2014; Zhu et al, 2022). Moreover, our own Western blot analysis of WT and UFM1-KO brain samples indicates the presence of as yet unidentified, neuronal UFMylated proteins, and our pilot experiment combining anti-UFM1 affinity purification and mass spectrometric analysis identified several new candidate UFM1 targets in mouse brain. In view of this, it will not only be important to study the UFM1-dependent translatome under normal and disease-related conditions but also to obtain comprehensive knowledge of all relevant targets of UFM1-WT and UFM1-R81C in neurons. On the basis on this type of information, it will be possible to determine the relevant cellular and molecular processes regulated by UFMylation, to explore their alteration in UFMylopathies, and to design novel therapeutic approaches.

## Methods

**Reagents and tools table**

| Reagent/resource | Reference or source | Identifier or catalog number |
|---|---|---|
| **Experimental models** | | |
| UFM1cKO | GemPharmatech | T008257 |
| **Recombinant DNA** | | |
| pRK5-UFM1 WT | Addgene | #133865 |
| pRK5-HA-UFM1 | Addgene | #134639 |
| pRK5-HA-UFM1-R81C | This paper | |
| pf(syn)-NLS-iCRE-RFP | Produced in the department | |
| pf(syn)-NLS -RFP | Produced in the department | |
| pf(syn)-NLS-iCRE-RFP-P2A-UFM1-WT | This paper | |
| pf(syn)-NLS-iCRE-RFP-P2A-UFM1-R81C | This paper | |
| pVSVG packaging | Produced in the department | |
| pCMV delta R8.9 envelope | Produced in the department | |
| **Software** | | |
| PRISM | | |
| Fiji | | |
| **Other** | | |
| **Antibody** | **Manufacturer** | **Catalog number** |
| UFM1 (1:1000 WB) | Abcam | Ab109305 |
| UFM1 (1:1000 WB) | Novus Biologicals | NBP2-94235 |
| MAP2 (1:1000 ICC) | Novus Biologicals | NB300-213 |
| Shank2 (1:500 IHC) | Synaptic Systems | 162 204 |
| Synapsin 1 (1:1000 ICC) | Synaptic Systems | 106 103 |
| Synapsin 1 (1:500 IHC) | Synaptic Systems | 106 011 |
| PSD95 (1:500 ICC) | NeuroMab | 75-028 |
| RFP (1:1000 ICC) | Synaptic Systems | 390 004 |
| PERK (1:1000 WB) | Cell Signaling | 3192 |
| ATF6 (1:1000 WB) | Novus Biologicals | NBP1-40256 |
| IRE1α (1:1000 WB) | Cell Signaling | 3294 |
| p-eIF2α (Ser51) (1:1000 WB) | Cell Signaling | 3398 |
| GFP (1:500 IHC) | MBL | MBL-598 |
| Puromycin (1:1000 WB) | Merck | MABE343 |
| HA (1:1000 WB) | Biolegend | 901515 |
| KDEL (1:200 ICC) | Enzo | ADI-SPA-827-D |
| FluoTag-X2 anti-PSD95 (1:100 ICC) | NanoTag | N3702 |
| FluoTag-X2 anti-VGluT1 (1:100 ICC) | NanoTag | N1602 |

| Reagent/resource | Reference or source | Identifier or catalog number |
|---|---|---|
| FluoTag-X2 anti-Synaptotagmin 1 (1:100 ICC) | NanoTag | N2302 |
| Anti-Chicken 405 (1:1000 ICC) | Abcam | Ab175674 |
| Anti-Chicken 647 (1:1000 ICC) | ThermoFisher | A-32933 |
| Anti-Guinea pig 555 (1:1200 IHC) | ThermoFisher | A-21435 |
| Anti-Rabbit 488 (1:1200 IHC; 1:1000 ICC) | ThermoFisher | A-11008 |
| Anti-Mouse 647 (1:1200 IHC; 1:1000 ICC) | ThermoFisher | A-31571 |
| Anti-Guinea pig 555 (1:1000 ICC) | ThermoFisher | A-21435 |
| Anti-Mouse-HRP (1:5000 WB) | Jackson ImmunoResearch | 115-035-146 |
| Anti-Rabbit-HRP (1:5000 WB) | Jackson ImmunoResearch | 111-035-144 |
| **DNA sequences/oligonucleotides** | | |
| Gene | Forward | Reverse |
| Gapdh | 5'-ATCTCCACTTT GCCACTGC -3' | 5'-AGGTCGGTGTGAA CGGATTTG -3' |
| UbcC | 5'-CCACACAAAGCCC CTCAAT -3' | 5'-CAAAGATCTGCAT CGTCTCTCTC -3' |
| Endogenous Ufm1 #1 | 5'-CCTGCACAGACTG CTGGGAA-3' | 5'-GCAGCTTCCAACT CGGTCT-3' |
| Endogenous Ufm1 #2 | 5'-GCGGTTCCACCAT GTCGAA -3 | 5'-GCGTACTTTCAGG AACACTGAGA-3' |
| Human UFM1 | 5'-GTGGAACAGTACG AGCGCG -3' | 5'-CCGACGTCAGCGT GATCTTA -3' |
| WRPE | 5'-GGCTGTTGGGCAC TGACAAT-3' | 5'-CCGAAGGGACGTA GCAGAAG-3' |
| ATF6 | 5'-GACAGCTCTTCGC TTTGGAC-3' | 5'-GGACGAGGTGGT GTCAGAG-3', |
| Xbp1 | 5'-CCGTGAGTTTTCTC CCGTAA-3' | 5'-AGCAAGTGGTGGA TTTGGAA-3' |
| Ern1 | 5'-GTCATCTCTGAGG GCAACCA-3' | 5'-ATTCACGAGCAAT GACGTCC-3 |
| Eif2ak3 | 5'-CGATGACGACGT GGAACTGC-3' | 5'-ACCCCACGTCCA AATCCCA-3' |
| Ufm1 for cloning sg-UFM1 | 5'-CGCTAGCCTGCA GGTCGACGGTGTGT-CAGGCGGTTCCAC-3' | 5'-GGGTCAGCTTGCC GATATCGACTGGCT TTCAGCAGAAATGC-3' |
| **Chemicals** | | |
| Puromycin | SIGMA | |
| Trazodone | SIGMA | |
| Thapsigargin | SIGMA | |
| Cycloheximide | SIGMA | |

## Animals

Mouse lines were maintained in the animal facilities of the Max-Planck Institute for Multidisciplinary Sciences, Göttingen, or in the animal facility of the Charité University Hospital. The *Ufm1*-cKO mouse line was obtained from GemPharmatech and carries LoxP sites before exons 3 and after exon 5 of the *Ufm1* gene (Fig. 2A). The line was kept in a homozygous state on a C57BL6/N background. Animals homozygous for *Ufm1*-cKO are viable, fertile, and exhibit no overt phenotypic changes in the cage environment. Mouse breeding was performed with permission of the Niedersächsisches Landesamt für Verbraucherschutz und Lebensmittelsicherheit (LAVES), according to European Union Directive 2010/63/EU and the ARRIVE guideline. Mice were housed in individually ventilated cages with bedding and nesting material, under specific pathogen-free conditions at 21 °C and 55% relative humidity, and under a 12 h/12 h light/dark cycle. Mice received food and water *ad libitum*. Health monitoring was done quarterly following FELASA recommendations with either NMRI sentinel mice or animals from the colony. All mice were sacrificed by administering a lethal dose of pentobarbital or decapitation (P0).

## Sex, age, and developmental stage of animals for in vivo experiments

Mice were used at embryonic (E), or postnatal (P) stages, as reported for each experiment. Each sample included both sexes within litters without distinctions. Littermates of both sexes were randomly assigned to experimental groups during experimental procedures or the collection of brain tissue.

## *In utero* electroporation (IUE)

*In utero* electroporation (IUE) experiments were performed in compliance with the guidelines for the welfare of experimental animals approved by the State Office for Health and Social Affairs, Council in Berlin, Landesamt für Gesundheit und Soziales (LaGeSo), permission G0184/20, Experiment 2. IUE was performed as previously described (Ambrozkiewicz et al, 2018; Borisova et al, 2024), using gRNA targeting *Ufm1* (target sequence: CTTTAAAATCACGTTGACGT) in bicistronic pX330/Cas9 (Wang et al, 2020). Briefly, equimolar pX330 and pCAG-EGFP (total 300 ng/uL) were co-injected into the lateral embryonic ventricle. Subsequent electroporation was performed using platinum electrodes, applying 6 pulses of 37 V. Brains of electroporated animals were fixed for immunohistochemistry by intracardiac perfusion with 4% paraformaldehyde/PBS solution at P23.

## Hippocampal and striatal neuron culture

Mouse autaptic and primary hippocampal and striatal neuron cultures were prepared as previously described (Burgalossi et al, 2012; Ripamonti et al, 2020). Briefly, P0 mice brains were dissected, and hippocampi or striata were isolated and digested for 1 h (hippocampi) or 75 min (striata) at 37 °C in Dulbecco's modified Eagle's medium (DMEM) containing 2.5 U/ml papain (Worthington Biomedical Corporation), 0.2 mg/ml L-cysteine, 1 mM $CaCl_2$, and 0.5 mM EDTA. Papain digestion was stopped by incubation with DMEM supplemented with 10% (v/v) heat-inactivated fetal bovine serum (FBS), 2.5 mg/ml albumin, and 2.5 mg/ml trypsin inhibitor. Tissues were triturated in Neurobasal medium (Neurobasal medium-A) containing 2% (v/v) B27, 100 U/ml penicillin, and 100 µg/ml streptomycin. Neurons were seeded in

Poly-L-Lysine (PLL)-coated plates or on 12 mm glass coverslips with the following densities: 50,000 cells/well of a 24-well plate for immunolabelling assays, 2 hippocampi (~150,000 cells/10 cm plate) for protein and RNA extraction or 25,000 cells/well in 8-well IBIDIs for live-imaging assays. For autaptic neuron culture (Burgalossi et al, 2012), neurons were plated at 3000 cells/plate (see below). Neurons were maintained at 37 °C and 5% $CO_2$ in Neurobasal medium supplemented with 2% (v/v) B27, 100 U/ml penicillin, and 100 µg/ml streptomycin. The day of dissection was counted as day in vitro 0 (DIV 0). At DIV 1, primary hippocampal neurons were infected with viruses as indicated in the figure legends. All reagents were from Gibco or Sigma-Aldrich, unless stated otherwise. Chemical treatments: Trazodone (Sigma, 20 µM) treatment was done every 3 days starting DIV 3. Cycloheximide (Sigma, 10 µg/ml) was applied for 2.5 h for FUNCAT experiments. Thapsigargin (Sigma, 3 µM) was applied for 15 min. Control neurons were treated with corresponding vehicles.

## HEK cell culture and lentivirus production

HEK293FT cells (ThermoFisher R70007) and N2a (Neuro-2A, DSMZ) cells were maintained at 37 °C, and 5% $CO_2$ in DMEM (ThermoFisher catalog number 10566016) supplemented with 10% (v/v) FBS, 100 U/ml penicillin, and 100 µg/ml streptomycin. For over-expression experiments and lentivirus production, cells were transiently transfected using Lipofectamine 2000 (ThermoFisher), following the manufacturer's instructions. The HEK293FT cell line was not authenticated or tested for mycoplasma. The N2a cells were routinely tested for mycoplasma, but not re-authenticated.

Lentivirus production was performed as previously described (Ripamonti et al, 2020). Briefly, HEK293FT cells were maintained in DMEM (ThermoFisher catalog number 10566016) supplemented with 10% (v/v) FBS, 100 U/ml penicillin, 100 µg/ml streptomycin, and 500 µg/ml gentamycin. Cells were transiently transfected with lentiviral constructs, combined with helper and packaging constructs in OPTIMEM medium (Gibco) containing 2% (v/v) FBS, 100 U/ml penicillin, 100 µg/ml streptomycin, and 10 mM sodium butyrate. Viral particles were collected after 48 h and concentrated via a series of centrifugation steps using concentration filters (Amicon Ultra-15 centrifugal filters, 10 kDa cut-off; Millipore). High-titer lentiviral aliquots were snap-frozen and stored at −80 °C.

## Molecular cloning

Molecular cloning of all vectors was performed using restriction enzymes from New England BioLabs, according to the manufacturer's protocols. Site-directed mutagenesis was performed using cloned Pfu DNA polymerase (Agilent Technologies), according to the manufacturer's instructions. Vector sources can be found in the vector Reagents and Tools Table. pRK5-HA-UFM1-R81C was generated via site-directed mutagenesis. Lentiviral constructs iCre-P2A-UFM1-WT and iCre-P2A-UFM1-R81C were generated by subcloning UFM1-WT or UFM1-R81C from pRK5 to the p(f)Syn-NLS-iCRE-RFP-P2A backbone. For the pCAG-EGxxFP-Ufm1 vector, the genomic sequence of the murine *Ufm1* locus was PCR-amplified using PrimeSTAR GXL polymerase (Takara) and primers listed in the primer list, and inserted into the pCAG-EGxxFP (Addgene, #50716) using EcoRI digestion and NEBuilder system (Mashiko et al, 2013).

## Western blotting

Cells were washed with PBS, lysed in RIPA buffer (150 mM NaCl, 1% (v/v) Triton X-100, 0,1% (v/v) SDS, 20 mM Tris pH 7.4–7.7) with protease inhibitors (1 µg/ml aprotinin, 0.5 µg/ml leupeptine, 17.4 µg/ml PMSF) and 20 mM N-ethylmaleimide (NEM). Protein concentration was determined using the Bradford method (Biorad). Proteins were then heated in the sample buffer at 65 °C for 20 min. SDS-PAGE was performed with standard discontinuous gels (Laemmli, 1970) or with commercially available 4%–12% Bis-Tris gradient gels (Invitrogen). Blot transfers were done using nitrocellulose membranes (Amersham Protran), according to standard procedures. Reversible MemCode protein staining (ThermoFisher) was performed to determine the total protein amount loaded. Blots were blocked with blocking buffer [PBS containing 5% (w/v) milk and 1% (v/v) Tween, or TBS containing 3% (w/v) BSA and 1% (v/v) Tween], and probed using primary and secondary antibodies diluted in blocking buffer. Blots were developed using enhanced chemiluminescence (GE Healthcare). Chemoluminescent signals were detected using an INTAS ECL Chemostar Imager. MemCode protein stain and blot images were analyzed using Fiji. Primary and secondary antibodies are listed in the Reagents and Tools Table.

## Puromycilation assay

Puromycin (20 µg/ml, Sigma) was added to the medium of DIV 10 neurons for 10 min at 37 °C, and 5% $CO_2$, as described (Borisova et al, 2024; Schmidt et al, 2009). The cells were washed with PBS, lysed for immunoblotting, and analyzed via Western blotting (see above), using an anti-puromycin antibody. The puromycin signal intensity was quantified using Fiji and normalized to the total protein loading as assessed using MemCode protein stain.

## Statistical analysis

Statistical tests were performed using PRISM 10. Test type was picked based on the number of comparisons made and the normality of the sample. Normality distribution was determined using D´Agostino–Pearson or Shapiro–Wilk normality tests. Test type, levels of statistical significance, and number of replicates are indicated in the figure legends. Statistical analysis was performed using a Kruskal–Wallis and Dunn´s multiple comparison test, unpaired $t$ test, Welch's test, or Mann–Whitney tests. Data are shown as mean ± SEM unless stated otherwise. For all figures, statistically significant differences are denoted in graphs with asterisks representing p-value ranges (*$P < 0.05$, **$P < 0.01$, ***$P < 0.001$, ****$P < 0.0001$). $N$ refers to the number of animals, and $n$ corresponds to the number of cells or biological replicates. $N$ and $n$ are indicated in the legends. No randomization, sample size estimation, or blinding procedures were applied. Outliers were excluded after performing ROUT.

## Immunocytochemistry (ICC)

Cells were fixed using 4% paraformaldehyde (PFA) diluted in 1× PBS. After fixation, coverslips were washed with PBS and blocked for 20 min in PBS containing 0.1% (v/v) Triton X-100 and 2.5% (v/v) fish skin gelatine (Sigma). Neurons were incubated with primary

antibodies for 1 h at room temperature (RT) or overnight at 4 °C. After three consecutive washes with PBS, coverslips were incubated with secondary antibodies for 45 min at RT, washed with PBS several times, and mounted using AquaPolymount (Polysciences). Primary and secondary antibodies were diluted in PBS containing 2.5% fish skin gelatin. For staining with nanobodies, cells were blocked in PBS supplemented with 2.5% (v/v) BSA and 0.1% (v/v) Triton X-100 for 20 min. Cells were incubated with nanobodies diluted in the same solution for 2 h at RT, washed, and mounted with Mowiol (VWR). Primary and secondary antibodies are listed in the Reagents and Tools Table.

### Immunohistochemistry (IHC)

Brains of electroporated animals were fixed for IHC by intracardiac perfusion with 4% paraformaldehyde (PFA)/PBS solution at P23. Mice were injected with a lethal dose of pentobarbital. Brains were post-fixed overnight (O/N) in 4% PFA and cryoprotected in 30% (w/v) sucrose diluted in 0.9% NaCl. 150 μm coronal sections were prepared with a cryostat and collected free-floating in PBS. Individual sections were blocked with PBS buffer supplemented with 10% (v/v) horse serum and 0.05% (v/v) Triton X-100 for 1 h at room temperature (RT). Slices were then incubated with primary antibodies for 2 days, extensively washed, and incubated with secondary antibodies O/N at 4 °C, with gentle shaking. Antibodies were diluted in PBS buffer supplemented with 10% (v/v) horse serum and 0.05% (v/v) Triton X-100. After washing with blocking buffer, slices were mounted with AquaPolymount (Polysciences). Primary and secondary antibodies are listed in the Reagents and Tools Table.

### Fluorescent noncanonical amino acid tagging (FUNCAT)

FUNCAT was performed using Click-iT™ Alexa Fluor™ 488 (Thermo). Briefly, DIV 10 or 12 neurons were starved in pre-warmed HBSS for 1 h at 37 °C, and 5% $CO_2$ and subsequently incubated with 1 mM HPG (L-homopropargylglycine) still in HBSS, for 90 min at 37 °C, and 5% $CO_2$. Neurons were washed with PBS and fixed with 4% PFA for 15 min. In the condition with cycloheximide (CHX) treatment, 10 μg/ml CHX was added to HBSS during the first incubation and kept until fixation. Neurons were then fluorescently labeled, according to the manufacturer's instructions via clicking of the Alexa-488 to the HPG (HPG/Alexa-488) and further immunolabeled for MAP2 before the coverslips were mounted on glass slides with AquaPolymount (Polysciences).

### Image acquisition and image analysis

All imaging was performed using Leica SP8 Confocal Microscope, Nixon Spinning Disc Microscope, or Abberior StedyCon. All microscope settings (laser power, detector settings) were kept the same between samples in each imaging session. The samples in each experiment were acquired in the same imaging session.

For morphology and synapse imaging of autaptic neurons, SP8 with a 63x objective (numerical aperture, NA, 1.4) was used. Neurons were selected for imaging based on immunolabelling for RFP, thereby validating the viral infection efficiency. MAP2 immunoreactivity was used to define regions of interest for the acquisition of the neuronal

processes. Z-stacks (step size 0.33 μm/slice) were acquired in order to image the entire neuron.

Super-resolution imaging was performed using StedyCon with a 100x objective (NA 1.45), and neurites of RFP-positive neurons were imaged.

Neuronal morphology was determined by Sholl analysis, using filament tracing (Imaris) of MAP2-positive neurites, and synapse puncta number and colocalization were determined using spots detector (Imaris). Only spots separated by less than 1 μm were counted as colocalizing.

The analysis of the HPG/Alexa-488 signal intensity (FUNCAT assay), was performed using SP8 with a 63x objective (NA 1.4). HPG/Alexa-488 signal intensity within the MAP2-positive cell body was quantified by determining the sum of the HPG/Alexa-488 signal intensity observed in all stacks (Fiji). Background values were subtracted. Intensity values were normalized to RFP control neurons.

For morphological analysis of EGFP-positive neurons in the mouse brain, SP8 with 20x objective (NA 0.75) was used. Z-stacks were acquired to image the complete neuronal cell body and processes. Dendrite complexity was examined using the Neuroanatomy plug-in (Fiji) by manually tracing neurite branches. Blinding during analysis was achieved by image name coding. For the analysis of synaptic puncta, SP8 with 63x objective (NA 1.4) was used. Acquisition was done in order to image a part of the cell body and at least 30 μm length of the main neurite. Images were deconvolved using Huygens, and neuronal volume was determined based on the EGFP signal that delineates the cell surface. Synaptic puncta number and colocalization were quantified using spots detector (Imaris). Only spots in contact with EGFP surface were considered. Only spots separated by less than 1 μm to each other were counted as colocalizing.

Live imaging and FRAP were performed using a Nixon Spinning Disc Microscope with a 100x objective (NA 1.49). DIV 5 neurons were washed 3 times with PBS and incubated with the ER-tracker Bodipy-Glibenclamid (Thermo) in pre-warmed Tyrode buffer (119 mM NaCl, 5 mM KCl, 25 mM HEPES, 2 mM $CaCl_2$, 2 mM $MgCl_2$, 6 g/l glucose, pH 7.4) for 30 min at 37 °C and 5% $CO_2$. After 3 washes with PBS, neurons were imaged in Tyrode buffer, in a constant 37 °C and 5% $CO_2$ chamber. Single confocal planes were acquired using a 488 nm laser. Neurons were selected for imaging based on the RFP signal. Photobleaching was performed using 100% laser intensity on the two main neurites of each neuron. Signal recovery was monitored every second. FRAP analysis was performed using FRAP profiler v2 plug-in (Fiji), which normalizes the overall bleaching to the background bleaching over time (https://worms.zoology.wisc.edu/research/4d/4d.html#frap).

### Anti-UFM1 brain affinity purification

Adult, 8–12 weeks old wild-type C57BL6N mice were killed via cervical decapitation and brain was quickly removed and flash frozen in liquid $N_2$. Frozen brain was pulverized in a porcelain mortar into powder and resuspended in 1% SDS, 150 mM NaCl, and 20 mM Tris pH 7.6. After sonication, lysate was diluted in 1% Triton, 150 mM NaCl, 20 mM Tris pH 7.6 in the presence of protease inhibitors and 20 mM NEM. Lyzate was cleared via ultracentrifugation at 100.000 g for 30 min and pre-adsorbed to protein A beads (Thermo) for 1 h.

After removal of the beads, anti-UFM1 antibody (40 µL) or corresponding rabbit IgG was incubated for 4 h, followed by a 2 h incubation with beads. Beads were then washed three times in lysis buffer before being elution with bound proteins in LDS sample buffer. Eluted samples were heated for 20 min at 65 °C before loading on a precast Bis-Tris gel (Invitrogen).

## LC-MS/MS analysis

### Gel electrophoresis and in-gel digestion

HEK293 cell lysates, cultured neuron lysates, or affinity-purified mouse brain proteins were separated on precast NuPage Bis-Tris 4–12% gradient gels (Invitrogen) using MES running buffer. Gels were run in parallel in the same gel electrophoresis chamber, and proteins were either visualized by colloidal Coomassie staining (gel 1) or transferred onto nitrocellulose membranes for immunodetection (gel 2) as described above. Coomassie signals were detected with a conventional transmitted light scanner, while a CCD camera system (Intas) was used to detect the ECL signal of immunoreactive bands on blotting membranes. The respective images were overlaid in Photoshop (Adobe) and false-colored in Fiji/ImageJ 1.52 to identify gel regions of interest. Gel bands were excised manually and subjected to in-gel digestion with Lys-C (Roche, for the investigation of the UFM1 C-terminal peptides), and with trypsin (Serva, for the identification of potential UFM1 targets) under standard conditions with prior reduction (DTT) and alkylation (iodacetamid, IAA) of free (i.e., non NEM-modified) cysteine sulfhydryl groups. Extracted proteolytic peptides were dried in vacuum centrifuge, desalted on AttractSPE C18 tips (Affinisep), and reconstituted in 0.1% TFA for LC-MS/MS analysis.

### Liquid chromatography

Nanoscale liquid chromatography (nanoLC) of proteolytic peptides was performed on an Ultimate 3000 RSLCnano LC system (NCS-3500RS Nano ProFlow; Thermo Fisher Scientific). Mobile phase A was 0.1% formic acid (FA) (v/v); mobile phase B was 0.1% FA in 80% acetonitrile (ACN) (v/v). Peptides were loaded to the trap column with 0.05% (v/v) TFA, 2% ACN (v/v) at a flow rate of 10 µl/min. Separation was performed at 40 °C column temperature with a gradient of solvent B: 5–25% for 37.5 min followed by 25–40% for 7.5 min (45 min total separation time) at a flow rate of 0.3 µl/min. HEK293 cell samples were analyzed on a column setup consisting of an Acclaim PepMap 100 nanoViper C18 trap column (100 µm × 2 cm, 5 µm, 100 Å) and an Acclaim PepMap RSLC nanoViper C18 separation column (75 µm × 25 cm, 2 µm, 100 Å); neuron samples were analyzed on a column setup consisting of a PepMap NEO C18 trap column cartridge (300 µm × 5 mm, 5 µm, 100 Å) and a DNV PepMap Neo double nanoViper C18 separation column (75 µm × 15 cm, 2 µm, 100 Å; all columns from Thermo Fisher Scientific). Samples for the identification of potential UFM1 targets were analyzed with a trap cartridge of the same specification as used for neuron samples, and Acclaim PepMap RSLC, nanoViper (75 µm × 15 cm, 2 µm 100 Å) separation column.

### Mass spectrometry

After separation by LC, peptides were introduced into an Orbitrap Exploris 480 mass spectrometer via a Nanospray Flex ion Source equipped with a stainless steel emitter (Thermo Fisher Scientific).

The mass spectrometer was operated in a data-dependent acquisition (DDA) and positive polarity mode. Spray voltage was set at 2.1 kV, the funnel RF level at 40, and the heated ion transfer tube temperature at 275 °C. The mass range was set to $m/z$ 375–1500, and a lock mass (445.12003 $m/z$) was used during data acquisition. All data were acquired in positive polarity using profile mode for MS1 and centroid mode for MS2. The individual acquisition settings were as follows:

Method 1 (HEK293 cell and neuron samples)—MS1, resolution 60,000 at $m/z$ 200, automated gain control (AGC) target 300%, maximum injection time (IT) 25 ms, intensity threshold 5e3, included charge states 2+ to 8+, exclusion duration 23 s, data-dependent mode-cycle time of 2 s between master scans; MS2, isolation window 1.6 $m/z$, normalized collision energy (NCE) 28%, resolution 15,000 at $m/z$ 200, AGC target 75%, maximum IT 40 ms.

Method 2 HEK (improved fragment ion spectra quality for HEK293 cell samples)—MS1, resolution 120,000 at $m/z$ 200, AGC target "standard", maximum IT "auto", intensity threshold 5e3, included charge states 2+ to 6+, exclusion duration 18 s, data-dependent mode-cycle time of 2 s; MS2, isolation window 1.6 m/z, fixed NCE 28% for 2+ and stepped NCE 30/35/40% for 3+ to 6+ precursor ions, resolution 30,000 at $m/z$ 200, AGC target "standard", maximum IT "auto", inclusion list (mass and z range) containing precursors corresponding to all UFM1 peptides identified with Method 1.

Method 2 Neuron (improved fragment ion spectra quality for neuron samples) – instrument settings as for Method 2 HEK, exclusion range 15 s, inclusion list (mass and z range) containing precursors corresponding to the individual C-terminal peptide of both UFM1 variants and the four shared peptides used for label-free quantification (LFQ).

In the experiment for the identification of potential UFM1 targets the following settings were used: MS1, resolution 60,000 at $m/z$ 200, AGC target 300%, maximum IT 25 ms, intensity threshold 5e3, included charge states 2+ to 5+, exclusion duration 22 s, data-dependent mode number of scans, TOP30; MS2, isolation window 1.6 $m/z$, NCE 28%, resolution 15,000 at $m/z$ 200, AGC target 50%, maximum IT 40 ms.

### Data analysis

Data analysis was performed with Proteome Discoverer (PD) 3.1.0.638 software (Thermo Fisher Scientific) using the Sequest HT search engine. MS data was searched against 2 FASTA protein sequence databases: one containing either the UniProtKB/Swiss-Prot human proteome (Taxonomy ID 9606, canonical sequences downloaded on 2023-10-24) supplemented with the sequence of mature HA-UFM1-WT/HA-UFM1-R81C; or mouse proteome (Taxonomy ID 10090, canonical sequences downloaded on 2024-06-14) supplemented with the sequences of mature UFM1-WT/ UFM1-R81C for HEK293 cell samples or neuron samples, respectively. Another FASTA database was PD-contaminants_2015_5, containing possible contaminant proteins. Search parameters were as follows: digestion enzyme LysC(Full), maximum missed cleavage sites 2, minimum peptide length 6, maximum peptide length 144. Precursor ion mass tolerance was 10 ppm for raw files acquired with MS1 resolution of 60,000 or

5 ppm for raw files acquired with MS1 resolution of 120,000, while the fragment ion mass tolerance was 0.02 Da in all cases. Oxidation (Met), Carbamidomethyl (Cys), and N-ethylmaleimide (Cys) were taken into account as dynamic peptide modifications (max. equal modifications per peptide 3, max. dynamic modifications per peptide 4). Acetylation, loss of Met, and loss of Met with subsequent acetylation were allowed as dynamic modifications at the protein N-terminus. Target decoy PSM validator (HEK293 cell samples) or Percolator (neuron samples) was used for validation, and the final result was filtered for identifications with false discovery rate (FDR) of 1%. For label-free quantification (LFQ) according to the TopN method, the precursor abundance was quantified based on the area of the peak, and protein abundance was calculated as the average of the four most abundant peptides shared by both UFM variants.

For identification of potential UFM1 targets, samples were analyzed against 3 FASTA protein sequence databases: (i) UniprotKB/Swiss-Prot + TrEMBL mouse proteome (Taxonomy ID 10090, proteome ID UP000000589, canonical and isoform sequences, downloaded on 2025-03-07), (ii) PD-contaminants_2015_database and (iii) fasta file with Rabbit Immunoglobulins (UniProtKB, ID 9986, filtered for containing "Immunoglobilin"/"Ig").

Data analysis settings that differ from those described above: digestion enzyme trypsin (Full), precursor mass tolerance 10 ppm, fragment mass tolerance 0.02 Da. Percolator was used for validation, results were filtered for FDR 1%. Raw files corresponding to the gel bands of the same sample were uploaded to the analysis as fractions of the sample.

## In vitro UFMylation

### Protein expression and purification
Expression and purification of UBA5, UFSP1, UFM1-GFP, UFM1 1-83, and UFC1 were performed as previously described (Makhlouf et al, 2024; Millrine et al, 2022). All proteins were stored in 25 mM HEPES pH 7.5, 200 mM NaCl, 1 mM DTT.

### Size-exclusion chromatography
Proteins (0.1 mg) were loaded on a Superdex™ 75 increase 3.2/300 column (Cytiva), pre-equilibrated in 25 mM HEPES pH 7.5, 200 mM NaCl, 0.5 mM TCEP.

### Biochemical assays
All biochemical reactions were stopped with loading dye at the indicated time-points and separated on 4–12% Bis-Tris SDS-PAGE under non-reducing conditions unless indicated otherwise, followed by staining with Instant Blue$^R$ Coomassie Protein Stain (Abcam).

For the UFM1-GFP cleavage assay, 2 µM UFM1-GFP and 0.01 µM UFSP1, pre-incubated in 25 mM HEPES pH 7.5, 200 mM NaCl, 10 mM DTT for 10 min, were mixed in 25 mM HEPES pH 7.5, 200 mM NaCl at 25 °C. For the UBA5 ~ UFM1 formation assay, 1 µM UBA5 and 7 µM UFM1 1-83 were mixed in 25 mM HEPES pH 7.5, 200 mM NaCl, 5 mM MgCl2, 5 mM ATP at 30 °C. For the UFC1 ~ UFM1 formation assay, 4 µM UBA5 was pre-charged with 14 µM UFM1 1-83 for 2 s or UFM1 1-83 R81C for 30 min. UBA5-charging was stopped by the addition of 50 mM EDTA, and the E2-charging reaction was initiated by the addition of 2 µM UFC1 at 20 °C.

### Data analysis
Protein levels were measured by quantifying bands using Fiji. For UFM1 maturation, GFP-UFM1 level at each timepoint was subtracted from GFP-UFM1 level at timepoint 0 (GFP-UFM1$_{t0}$), and then normalized against GFP-UFM1$_{t0}$. For the UFM1-UBA5 formation assay, the UFM1-UBA5 level at each timepoint was normalized against UBA5 level at timepoint 0. For UFM1-UFC1 interaction, UFM1-UFC1 level at each timepoint was normalized against the sum of UFC1-UFM1 and UFC1 at each timepoint.

## Electrophysiology

Whole-cell patch-clamp recordings were performed in DIV 10-12 autaptic excitatory hippocampal or inhibitory striatal neurons. Medium was changed for the extracellular solution (4 mM MgCl$_2$, 4 mM CaCl$_2$, 140 mM NaCl, 2.4 mM KCl, 10 mM Glucose, 10 mM HEPES, pH 7.6), and cell soma was injected with intracellular solution (136 mM KCl, 1 mM EGTA, 17.8 mM HEPES, 4.6 mM MgCl$_2$, 0.3 mM Na$_2$GTP, 4 mM NaATP, 5 U/ml phosphocreatine kinase, and 15 mM creatine phosphate, osmolarity 315-320 mOsmol/L, pH 7.4). All recordings were done using an EPC-9USB amplifier (HEKA electronics) controlled by PatchMaster software (Version 2×90.3, HEKA electronics). The recording rate was 10 kHz, Rseries up to 10 MΩ using 50% compensation. Voltage-clamp configuration was used, and action potentials were induced by depolarizing steps from −70 to 0 mV. Microelectrodes were fabricated from borosilicate glass pipettes using a micropipette puller (P-27, Sutter Instrument). Pipette resistances were used at ~2–4 MΩ. Excitatory post-synaptic currents (EPSC) were recorded using frequencies of 0.2 Hz. The release of vesicles from the readily releasable pool (RRP) was triggered by applying 0.5 M sucrose solution. Synaptic plasticity studies were performed by stimulating neurons with 10 Hz and 40 Hz stimulus trains. Recording of miniature EPSC (mEPSC) was done under the application of 300 mM tetrodotoxin (Tocris). Post-synaptic responses were induced by the application of 100 µM Glutamate (Sigma) or 3 µM γ-aminobutyric acid (GABA, Sigma). Neurons were selected based on their nuclear RFP signal using an inverted microscope (Zeiss). Data analysis was performed using AxoGraph software (Version 1.5.4, Axograph Scientific).

## RNA isolation, RNA-sequencing, and analysis

DIV 13 primary hippocampal neurons were subjected to TRIzol-RNA extraction following the manufacturer's protocol (Zymo). RNA quality was assessed by measuring the RNA integrity number (RIN) using a Fragment Analyzer HS Total RNA Kit (Advanced Analytical Technologies, Inc.). Library preparation for RNA-Seq was performed in the STAR Hamilton NGS automation using the Illumina Stranded mRNA Prep, Ligation (Cat. N°20040534 and the Illumina RNA UD Indexes Set A, Ligation, 96 Indexes, 96 Samples Cat. N°20091655), starting from 300 ng of total RNA. The size range of the final cDNA libraries was determined by applying the SS NGS Fragment 1- to 6000-bp Kit on the Fragment Analyzer (average 340 bp). Accurate quantification of cDNA libraries was performed by using the

QuantiFluor™ dsDNA System (Promega). cDNA libraries were amplified and sequenced by using an S4 flow cell NovaSeq6000; 300 cycles, 25 Mio reads/sample from Illumina.

### Raw read and quality check

Sequence images were transformed with BaseCaller Illumina Software to BCL files and demultiplexed to fast files with bcl2fastq v2.20.0.422. The sequencing quality was asserted using FastQC v.0.11.5 software (http://www.bioinformatics.babraham.ac.uk/projects/fastqc/) (Wingett and Andrews, 2018).

### Mapping and normalization

Sequences were aligned to the reference genome Mus musculus (GRCm39 version 110, https://www.ensembl.org/Mus_musculus/Info/Index) using the RNA-Seq alignment tool (version 2.7.8) (Dobin et al, 2013), allowing for 2 mismatches within 50 bases. Subsequently, read counting was performed using featureCounts (Liao et al, 2014).

### PCA and other quality checks

The total gene count box plot displays gene expression levels in each of the samples analyzed (Fig. S6A). The horizontal line inside each box represents the median gene expression. The box itself encompasses the interquartile range, covering from the 25th to the 75th percentile. The whiskers extend to the 10th percentile on the lower end and the 90th percentile on the higher end. Any points outside the whiskers, particularly those above the 90th percentile, are shown as individual circles. The plot uses color to differentiate between conditions, with red indicating the CRE condition and blue representing the RFP condition. Principal component analysis (PCA) was performed using the sklearn Python library (version 1.3.1) (Fig. S6B).

### Differential expression and pathway enrichment analysis

Differential expression analysis was performed using PyDESeq2, a Python implementation of DESeq2 method (Love et al, 2014; Muzellec et al, 2023) (version 0.5.1). Pathway enrichment was done with GSEA method (Subramanian et al, 2005) using blitzgsea library (version 1.3.47) (Lachmann et al, 2022) as well as the EnrichR method using the gseapy library (version 1.1.3). Gene sets "KEGG 2019 Mouse", "GO Molecular Function 2023", "GO Biological Process 2023" and "GO Cellular Component 2023" were used. For SynGO gene annotation (Volcano plot, enrichement), SynGO version 20231201 (https://www.syngoportal.org, (Koopmans et al, 2019)) was used. The calculation of the score in pathway analysis was done as previously described (Hendriks et al, 2018).

## RNA extraction and real-time quantitative PCR (RT-qPCR)

Neurons were homogenized in 300 μl of DNA/RNA Shield (Zymo Research) and stored at -80 °C until RNA extraction and purification using the Direct-zol RNA microprep Kit (QIAGEN) and according to the manufacturer's protocol.

Reverse transcription of purified RNA was performed with Super-Script III First-Strand Synthesis System (Invitrogen) following the manufacturer's instructions, with random 9-mer oligonucleotides and oligodT. For each reaction, 0.5 μg of total RNA was used. The cDNA samples were stored at −20 °C until further processing.

RT-qPCR was performed using Mic qPCR (BioMolecular Systems). Each gene was analyzed in duplicates. All genes were analyzed using Power SYBR™ Green PCR Master Mix (Applied Biosystems), where 0.2 pmol/μl of each RT-primer (see Reagents and Tools Table) and 1 μL of cDNA were used in a final reaction volume of 20 μl.

All gene Ct values were standardized to the Ct values of *Gapdh* and *Ubcc*. For mRNA fold change analysis, Ct values were also normalized to the levels of the RFP condition. Oligonucleotide sequences are listed in the Reagents and Tools Table.

## Cleavage analysis of *Xbp1*

qPCR was loaded on a 2% agarose gel containing GelRed nucleic acid stain (Biotum/Avantor) and 100 bp Plus DNA Ladder as size indicator (Thermo). Xbp-1 fragment analysis was performed as previously described (Andreae et al, 2025).

## Primary hippocampal neuron culture for electron microscopy

For electron microscopy experiments, neuronal monolayer cultures from P0 UFM1-cKO mice were grown directly on carbon- and PDL-coated 6 mm sapphire disc freezing substrates (Wohlwend). Sapphire discs were mounted on 18 mm glass coverslips using Matrigel and placed in a 12-well plate for UV sterilization for 1 h at RT. Hippocampi were prepared as described above in this section. Cultured neurons were vitrified by high-pressure freezing at DIV 12 and further processed for 3D ultrastructural analysis.

## High-pressure freezing, freeze-substitution, and ultramicrotomy

Sapphire discs with neurons were infected at DIV 1 with lentiviral constructs designed to express CRE-recombinase and RFP, or RFP alone (negative control). At DIV 12, infected neurons underwent three washing steps in pre-warmed (37 °C) washing solution (4 mM MgCl₂, 4 mM CaCl₂, 140 mM NaCl, 2.4 mM KCl, 10 mM Glucose, 10 mM HEPES, pH 7.6) before rapid cryofixation using an EM ICE high-pressure freezer (Leica) and subsequent storage in liquid nitrogen. Semi-automated freeze-substitution was performed using an AFS2 device (Leica) equipped with customized aluminum sapphire disc revolvers, and samples were subsequently embedded in epoxy resin according to previously published protocols (Imig et al, 2020; Rostaing et al, 2006). Sapphire discs were separated from polymerized samples with a razor blade, and resin block-faces were trimmed with an EM TRIM high-speed milling device (Leica) in preparation for ultramicrotomy. A 35° Ultra-Semi diamond knife (Diatome) mounted on a UC7 ultramicrotome (Leica) was used to cut 250 nm-thick sections onto formvar-coated copper parallel line grids (Gilder; G100P-Cu). Mounted sections were coated on both surfaces with Protein A-conjugated 10-nm gold fiducial particles (Cell Microscopy Core Products, University Medical Center Utrecht, The Netherlands) in preparation for 3D ultrastructural analysis using transmission electron tomography.

## Electron microscopy

Synapses were identified and selected for 3D ultrastructural analysis in tiled overviews acquired at ×5,000 magnification using a 200 kV

**The paper explained**

**Problem**

UFMylation, the post-translational modification of proteins with the ubiquitin-like modifier UFM1, has been implicated in the regulation of protein translation. Strikingly, mutations in genes encoding components of the UFMylation pathway and the consequent perturbation of UFMylation cause developmental and epileptic encephalopathies, but the role of UFMylation in nerve cells is unclear. The present study examined how UFMylation regulates neuronal development and function and how pathologically altered UFMylation perturbs these processes.

**Results**

Based on a conditional UFM1 knock-out mouse model, we mimicked two pathological scenarios, UFM1 loss and the expression of mutant UFM1-R81C in neurons, and characterized their development and function. We found that UFM1 loss and UFM1-R81C expression cause, to varying degrees, a set of common cellular defects, including impaired neuronal development, defects in synaptic signaling, and reduced protein translation. Interestingly, treatment of neurons with the antidepressant Trazodone restores protein translation in UFM1-R81C-expressing neurons and increases synapse number in UFM1-KO and UFM1-R81C-expressing neurons.

**Impact**

Our study provides insights into the role of UFMylation in neuronal development, into the molecular basis of UFM1-associated neurodevelopmental diseases, and into potential treatments for these disorders.

Talos F200C G2 scanning/transmission electron microscope equipped with a 16 MP Ceta CMOS camera (Thermo Scientific) and SerialEM software (*IMOD* Software Package (Mastronarde, 2005)). Tilt series (±60° tilt range; 1° tilt increments) were acquired at 36,000x (unbinned pixel size = 0.4 nm) magnification from orthogonal axes using a Model 2040 dual-axis specimen holder (Fischione). Tomograms were generated from tilt-series using Gaussian filtering and a weighted back-projection algorithm implemented using *Etomo* software (IMOD software package (Kremer et al, 1996)). Pre-synaptic vesicles were initially segmented as perfect spheres using SynapseNet (Muth et al, 2025) and then proof-read using 3dmod software (*IMOD Software PackMastrona-deage (*Kremer et al, 1996*)*). Pre-synaptic membranes and active zones were segmented manually on every 10th or 3rd tomographic slice, respectively. Segmented data were analyzed using *mtk* and *imodinfo* programs in conjunction with a customized Python script to quantify the relative spatial distribution of pre-synaptic vesicles (i.e., closest approach to active zone release sites), their size (i.e., SV diameter), and clustering characteristics (i.e., intervesicular nearest-neighbor distances), respectively.

## Data availability

The datasets produced in this study are available in the following databases: (i) the RNA-seq data (Dataset EV1: RNA-seq analysis of UFM1 deficient neurons) is accessible under the identifier GSE299278 at Gene Expression Omnibus (https://www.ncbi.nlm.nih.gov/gds/); (ii) the mass spectrometry proteomics data (Dataset EV2: Identification of UFM1 candidate targets from

mouse brain) have been deposited to the ProteomeXchange Consortium via the PRIDE partner repository (https://www.ebi.ac.uk/pride/) with the dataset identifier PXD071267 (Perez-Riverol et al, 2025).

The source data of this paper are collected in the following database record: biostudies:S-SCDT-10_1038-S44321-026-00389-6.

## Peer review information

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

## Acknowledgements

We thank D. Warnecke, C. Harenberg, and M. Wzietek from the MPI-NAT DNA Core Facility for oligonucleotides synthesis and animal genotyping, A. Guenther for helping with neuron cultures, T. Liepold and M. Uecker for sample preparation for mass spectrometry, and R. Dannenberg and K. Cuthill for the support with perfusions of P23 animals. We thank the MPI-NAT and Charite Animal Facilities for the maintenance of mouse colonies, and the MPI-NAT Light Microscopy Facility for the help with microscope settings. We thank Theres Schaub for the design and cloning of the pEGxxFP-sg-UFM1. MCA is a Scholar of the FENS-Kavli Network of Excellence, and work in his lab was supported by the Fritz Thyssen Foundation (10.23.2.003MN), the German Research Foundation (DFG; project IDs 515247130 and 536563141). Work by NB, MT, JT, FM, and BHC was supported by the German Research Foundation (Deutsche Forschungsgemeinschaft, DFG; SFB1286-A01/A03/A09).

## Author contributions

**Catarina Perdigão**: Conceptualization; Resources; Data curation; Formal analysis; Supervision; Validation; Investigation; Visualization; Methodology; Writing—original draft; Project administration; Writing—review and editing.
**Josefa Torres**: Conceptualization; Data curation; Formal analysis; Supervision; Validation; Investigation; Methodology. **Helge M Magnussen**:

Conceptualization; Data curation; Formal analysis; Validation; Investigation; Methodology; Writing—original draft; Writing—review and editing. **Janina Koch**: Data curation; Formal analysis; Investigation; Methodology; Writing—original draft. **Elena Rudashevskaya**: Data curation; Formal analysis; Validation; Methodology; Writing—original draft; Writing—review and editing. **Frederieke Moschref**: Data curation; Formal analysis; Funding acquisition; Validation; Visualization; Methodology; Writing—original draft; Writing—review and editing. **Maksims Fiosins**: Conceptualization; Data curation; Formal analysis; Writing—review and editing. **Fritz Benseler**: Resources; Data curation; Software; Formal analysis; Validation; Investigation; Visualization; Methodology; Writing—original draft; Writing—review and editing. **Sally Wenger**: Formal analysis; Investigation; Methodology. **Tanja Nilsson**: Formal analysis; Validation; Investigation. **Sabine Beuermann**: Investigation. **Stefan Bonn**: Data curation; Formal analysis; Supervision; Funding acquisition; Writing—original draft; Writing—review and editing. **Silvio O Rizzoli**: Conceptualization; Resources; Data curation; Formal analysis; Supervision; Methodology; Writing—original draft; Writing—review and editing. **Yogesh Kulathu**: Resources; Supervision; Validation; Writing—original draft; Writing—review and editing. **Olaf Jahn**: Conceptualization; Resources; Data curation; Software; Formal analysis; Supervision; Investigation; Methodology; Writing—original draft; Writing—review and editing. **Benjamin H Cooper**: Data curation; Formal analysis; Funding acquisition; Investigation; Methodology; Writing—original draft; Writing—review and editing. **Mateusz C Ambrozkiewicz**: Conceptualization; Data curation; Formal analysis; Supervision; Funding acquisition; Validation; Investigation; Writing—original draft; Writing—review and editing. **JeongSeop Rhee**: Conceptualization; Supervision; Validation; Investigation; Visualization; Methodology; Writing—original draft. **Nils Brose**: Conceptualization; Funding acquisition; Writing—original draft; Project administration; Writing—review and editing. **Marilyn Tirard**: Conceptualization; Data curation; Formal analysis; Supervision; Funding acquisition; Validation; Investigation; Visualization; Methodology; Writing—original draft; Project administration; Writing—review and editing.

Source data underlying figure panels in this paper may have individual authorship assigned. Where available, figure panel/source data authorship is listed in the following database record: biostudies:S-SCDT-10_1038-S44321-026-00389-6.

## Funding

## Disclosure and competing interests statement
The authors declare no competing interests.

