## [Peer Review File · EMBO Molecular Medicine]

Encephalopathy-linked UFM1 variants impede neuronal protein translation, development, and function.

Catarina Perdigao, Josefa Torres, Helge Magnussen, Janina Koch, Elena Rudashevskaya, Frederieke Moschref, Maksims Fiosins, Fritz Benseler, Sally Wenger, Tanja Nilsson, Sabine Beuermann, Stefan Bonn, Silvio Rizzoli, Yogesh Kulathu, Olaf Jahn, Benjamin Cooper, Mateusz Ambrozkiwicz, JeongSeop Rhee, Nils Brose, and Marilyn Tirard

Corresponding authors: Marilyn Tirard (tirard@mpinat.mpg.de) , Nils Brose (brose@mpinat.mpg.de)

Review Timeline:

Submission Date:	4th Mar 25
Editorial Decision:	15th Apr 25
Revision Received:	9th Dec 25
Editorial Decision:	18th Dec 25
Revision Received:	27th Jan 26
Accepted:	28th Jan 26

Editor: Zeljko Durdevic

Transaction Report:

15th Apr 2025

Dear Dr. Tirard,

Thank you for the submission of your manuscript to EMBO Molecular Medicine, and please accept my apologies for the delay in getting back to you, which is due to the fact that one referee needed more time to complete his/her review. We have now received feedback from the three reviewers who agreed to evaluate your manuscript.

As you will see from the reports pasted below, while the referees #1 and #2 are overall supportive raising important concerns, referee #3 recognizes the interest of the study but also raises serious concerns particular regarding the inconclusiveness of the study. After the cross-commenting discussion it became clear that the focus of the major revision should be on providing more mechanistic insight and performing additional experiments that further strengthen the main conclusions of the study. Considering that the revision will require extensive experimentation we think six months rather than three months would be more appropriate to provide the complete revision. If you would like to discuss further the points raised by the referees, I am available to do so via email or video. Let me know if you are interested in this option.

We would welcome the submission of a revised version within six months for further consideration. Please let us know if you require longer to complete the revision.

I look forward to receiving your revised manuscript.

Yours sincerely,

Zeljko Durdevic

Zeljko Durdevic
Senior Editor
EMBO Molecular Medicine

We require:

- 1) A .docx formatted version of the manuscript text (including legends for main figures, EV figures and tables). Please make sure that the changes are highlighted to be clearly visible.
- 2) Individual production quality figure files as .eps, .tif, .jpg (one file per figure). For guidance, download the 'Figure Guide PDF': (<https://www.embopress.org/page/journal/17574684/authorguide#figureformat>).
- 3) A .docx formatted letter INCLUDING the reviewers' reports and your detailed point-by-point responses to their comments. As part of the EMBO Press transparent editorial process, the point-by-point response is part of the Review Process File (RPF), which will be published alongside your paper.
- 4) A complete author checklist, which you can download from our author guidelines (<https://www.embopress.org/page/journal/17574684/authorguide#submissionofrevisions>). Please insert information in the

checklist that is also reflected in the manuscript. The completed author checklist will also be part of the RPF.

6) It is mandatory to include a 'Data Availability' section after the Materials and Methods. Before submitting your revision, primary datasets produced in this study need to be deposited in an appropriate public database, and the accession numbers and database listed under 'Data Availability'. Please remember to provide a reviewer password if the datasets are not yet public (see <https://www.embopress.org/page/journal/17574684/authorguide#dataavailability>).

12) Author contributions: You will be asked to provide CRediT (Contributor Role Taxonomy) terms in the submission system. These replace a narrative author contribution section in the manuscript.

13) A Conflict of Interest statement should be provided in the main text.

14) Every published paper now includes a 'Synopsis' to further enhance discoverability. Synopses are displayed on the journal webpage and are freely accessible to all readers. They include a short stand first (maximum of 300 characters, including space) as well as 2-5 one-sentences bullet points that summarizes the paper. Please write the bullet points to summarize the key NEW findings. They should be designed to be complementary to the abstract - i.e. not repeat the same text. We encourage inclusion of key acronyms and quantitative information (maximum of 30 words / bullet point). Please use the passive voice. Please attach these in a separate file or send them by email, we will incorporate them accordingly.

15) Include a Reagents and Tools Table as part of the Methods section, which can be downloaded from our author guidelines (<https://www.embopress.org/page/journal/17574684/authorguide#structuredmethods>)

***** Reviewer's comments *****

Referee #1 (Comments on Novelty/Model System for Author):

The authors have used a very interesting approach to address potential cellular mechanisms of encephalopathies caused by mutations in the UFM1 gene by deletion of the gene in murine neurons and replacing it by expression of constructs carrying either wildtype sequence or the sequence of a pathological human mutation. The work is technically sound and includes a state-of-art strategies. The medical significance is twofold: firstly, a potential disease mechanism in humans is identified, and secondly, a drug approved for the treatment of humans is used to correct the cell physiological deficits of neurons expressing the mutant human protein. Altogether I consider the mouse model as very reasonable for the addressed topic.

Referee #1 (Remarks for Author):

Using suitable mouse models, the authors studied potential molecular and cellular pathomechanisms underlying encephalopathies caused by absence or pathological mutations of the UFM1 gene and consequently the disturbance of the process of UFMylation. In a first in vivo experiment, the Ufm1 gene was knocked out in the murine cortex after day 14.5 in utero and analyzed at postnatal day P23. This resulted in simplification of neuronal architecture and significant reduction in excitatory synapse number indicating clear deficits in neuronal development. All further experiments were performed in primary cultures of hippocampal neurons from cKO mice with a floxed Ufm1 gene upon infection with viruses {plus minus}Cre recombinase expression. For electrophysiological experiments, specifically autaptic micro-island cultures were used. These experiments basically confirm the in vivo findings. Individual synapses, however, that are formed seemed relatively normal. As major consequence of impaired UFMylation, a strong reduction of protein translation was detected in UFM1-KO primary neurons. In a clever set of experiments, re-expression of either wildtype UFM1 or UFM1 carrying a pathological human mutation (UFM1-R81C) was used to assess for rescue of observed phenotypes. Thereby expression levels of UFM1-WT and UFM1-R81C were carefully controlled. Under these conditions wildtype proteins favorably compensated UFM1 deficiency whereas UFM1-R81C only partially rescued the defects. Main deficits, as compared to wildtype rescue, were observed in response to ER stress. The data suggest that the human mutation in UFM1 enhances the unfolded protein stress response via the Protein kinase R-like ER kinase (PERK) pathway. Interestingly, trazodone, an inhibitor of the PERK pathway that is already in use as an antidepressant, can counteract UFM1-R81C deficits in normalizing unfolded protein response and therefore is a candidate to be repurposed for patients carrying the UFM1 gene mutation.

This is a thoroughly designed and well-performed study that contributes significantly to the understanding of cellular mechanisms underlying UFMylation-related neurological disorders and identifies a candidate drug to be assessed for future treatment. I have only a couple of minor comments and questions.

1) General: The authors convey the impression that UFM1 dysfunction leading to encephalopathy only affects excitatory neurons and glutamatergic synapses. The only experiment concerning inhibitory neurotransmission documents that GABAergic input on primary excitatory neurons seems normal. I hardly can believe that UFMylation does not affect interneurons and suggest that this point should be addressed in the discussion.

2) For clarity and understandability for non-specialists, all essential abbreviations should be defined also in the abstract (e.g. PERK-UPR). In general, a list of abbreviations may help to increase the readability.

3) Cartoon in synopsis: If I understand correctly, the phenotype mainly affects neuronal architecture, dendritic branching and reduced number of synapses, while available synapses are relatively normal. The cartoon gives the impression that individual synapses are severely impaired in their function. Is this what is meant?

4) page 4: "... we combined CRISPR/Cas9 and in utero electroporation (IUE) to deplete UFM1 in E14.5 mouse cortical progenitors and their neural progeny at P23." This statement needs clarification. I assume it means that UFM1 is deleted from E14.5 onwards in cortical progenitors and their neural progeny and that P23 was the day analysis. Please, clarify.

5) Fig. 1 (legend): "Data from B and C were obtained...", I assume that this must read "C and "D"

6) Fig. 5: Legends for panels A and B are basically identical. However, panel A does not show any immunostaining. Please, adapt the text to what is really documented.

Panel D, lettering of 'CRE' is a bit odd.

Legend last paragraph: "For -H, data were obtained ..." means what? "For ??-H, data were obtained ...".

7) Fig. 8: Maybe a small cartoon could be added to this figure indicating the role of the analyzed players in the UFMylation process. This could ease the understanding of the assay.

8) Fig. 10: The final statement in the result section on Trazodone action seems a bit strong. What are the significance values? There is a clear increase in Synapsin puncta (Fig. 10D), but no significant difference in postsynaptic marker PSD95 (Fig. 10C) nor in the overlap of pre- and postsynaptic markers (Fig. 10E). What are the actual p-values? Explain the use of different statistical tests.

9) Fig. S8: Panel assignment is wrong, panel E is described as panel D (which appears twice in the legend).

Referee #2 (Remarks for Author):

General Comment

This manuscript presents a comprehensive investigation into the cellular and molecular mechanisms underlying neurodevelopmental disorders caused by UFM1 mutations, including complete loss-of-function and the patient-derived R81C missense variant. Using both in vivo and in vitro models, the authors convincingly demonstrate that UFM1 deficiency impairs dendritic arborization, synaptogenesis, and synaptic transmission, accompanied by a marked reduction in protein synthesis and hyperactivation of the ER stress PERK-UPR pathway. Although the R81C mutant retains partial functionality, it exhibits an aberrant response to ER stress and fails to fully rescue the phenotypes observed in UFM1-deficient neurons. Notably, the authors show that the antidepressant trazodone can partially restore protein synthesis and synapse number in neurons expressing UFM1-R81C, highlighting potential therapeutic relevance.

The experimental design is thorough, employing a wide array of genetic, molecular, and electrophysiological techniques, with well-controlled and consistent results. The study offers valuable mechanistic insights into UFMylation-dependent neuronal development and function, and significantly contributes to our understanding of UFM1-linked encephalopathies.

Below, I provide major and minor comments aimed at improving clarity, interpretative depth, and contextual framing of the findings.

Major Comments

1. Unclear mechanistic classification of the UFM1-R81C variant

The authors describe UFM1-R81C as "not merely a loss-of-function variant," but its precise functional classification remains ambiguous. Given the observed phenotype - partial rescue under basal conditions and exaggerated ER stress responses under challenge - alternative mechanisms such as a dominant-negative or gain-of-toxic-function effect should be considered and discussed more explicitly.

2. Unresolved identity of UFM1-conjugated target proteins

The Western blot data reveal a reduction in high-molecular-weight UFM1-conjugates in UFM1-deficient neurons, suggesting the existence of multiple substrates. However, the identities of these putative UFM1 targets remain undetermined. Further proteomic profiling or immunoprecipitation experiments (even preliminary) would be valuable, or at least the authors should more clearly state how they plan to address this important question in future work.

3. Unclear mechanistic basis for trazodone's selective effects

The study demonstrates that trazodone rescues translation and synapse number in UFM1-R81C-expressing neurons but not in UFM1-null neurons. However, the molecular basis of this differential response is not well explained. A deeper discussion of possible mechanisms - for example, differences in residual UFMylation, PERK-UPR signaling thresholds, or alternative pathways - would enhance the interpretive strength of the study.

4. Potential confusion due to antibody limitations in detecting UFM1-R81C

Since the C-terminal epitope of UFM1-R81C is altered, it is not detected by the standard anti-UFM1 antibody, leading to an apparent lack of signal in immunoblots. While this issue is briefly addressed in supplementary figures, the main text and figure legends would benefit from a clearer and more prominent explanation to avoid misinterpretation by readers.

5. Lack of long-term functional or behavioral assessment of trazodone treatment

Although trazodone shows partial rescue of molecular and cellular phenotypes, the study does not evaluate whether these changes translate into long-term improvements in neuronal function or behavior. Future work should consider examining electrophysiological network activity, calcium dynamics, or behavioral outputs to assess therapeutic relevance more thoroughly.

Minor Comments

1. Statistical reporting clarity

The manuscript employs a range of statistical tests (e.g., Mann-Whitney, Kruskal-Wallis, ANOVA), but the criteria for selecting each test (e.g., normality checks) are not consistently stated in the Methods section or figure legends. Including the rationale for test selection alongside the sample sizes (n) would improve transparency and reproducibility.

2. Inconsistent terminology

Terms such as "UFM1-R81C," "R81C," and "UFM1-R81C-expressing neurons" are used somewhat interchangeably throughout the manuscript. Standardizing terminology across the text would enhance readability and avoid ambiguity, particularly for non-specialist readers.

3. Insufficient citation of original literature

The Introduction and Discussion sections rely heavily on review articles, with limited citation of key original studies. Important foundational and mechanistic references are notably absent, including:

- Komatsu et al., 2004 (*EMBO J*), which first identified and characterized the UFM1 conjugation system;
- Walczak et al., 2019 and DaRosa et al., 2024, which demonstrated UFMylation of RPL26 and its role in ER-associated ribosome quality control and translational regulation.

Inclusion of these primary sources would significantly strengthen the contextual framework of the study and better acknowledge the existing body of work in this field.

Referee #3 (Comments on Novelty/Model System for Author):

They claim using mice to study neuronal development but they show cell culture. Neuronal development needs to be studied in mice.

The medical potential is low because it is unrealistic to imagine a treatment for a developmental disease due to a genetic mutation ie no drug will rescue the brain defects.

Referee #3 (Remarks for Author):

In this manuscript the authors claim using a Ufm1 knock-out mice, to investigate two types of UFMylation pathologies, UFM1 loss and expression of a pathogenic UFM1-R81C variant. They claim that UFM1-deficiency confounds neuron development and synapse function. They conclude "Our study unveils a pivotal role for UFMylation in neuronal development, provides a molecular understanding of the signaling mechanisms altered in UFM1- associated encephalopathies, and offers important insights into potential treatments for these disorders". There is a disconnect between the claims and the content of the manuscript and several methodological issues.

Major issues:

1. The abstract infers that the work is done in mice, whilst in fact most of the work is done in cultured neurons.
2. The conclusions of the study are not novel. It was known before that UFM1 deficiency impairs neuronal development and function. They claim new mechanistic insights. This is not the case. There may be opportunities to explore new aspects of UFMylation in this study but they are not explored. (see below).
3. They need to show the efficacy of their gene targeting approach before the phenotypic validation. They don't show the efficacy of the CRISPR/Cas9 in utero electroporation to deplete UFM1 in E14.5 mouse cortical progenitors and their neural progeny at P23. This is a major problem with this study. One imagines that the in utero targeting efficacy of the Crisp targeting is not 100% and that the animals they generated are mosaic. There ought to be much variation. How do the authors assess whether what they report is due to inherent variability of the system or differences resulting from the genetic modification. The study is inconclusive as it stands.
 1. What are the UFM1-positive protein species in Figure 2. Identifying them could reveal mechanisms of neuronal UFM1.
 2. There are methodological issues with the translation experiments; They find that CHX reduces translation signal to 50% which establishes that they have background staining (as often in such assays). This invalidates these experiments and their conclusions.
 3. The gel Fig 4C is not equally loaded. Decrease in puro label parallels that of the total proteins.
 4. The measurements of UPR activation needs to be done with robust and validated readouts, assessing both protein and mRNA changes and including a control such as Tg or Tm to ensure that they are within the range of detection of UPR activation in their system. The blots showing ATF6, PERK and IRE1a, P-elf2 are not indicative of UPR activation.

Point-by-Point Response to the Reviewer Comments (Responses in Red)**Reviewer #1 (Comments on Novelty/Model System for Author):**

The authors have used a very interesting approach to address potential cellular mechanisms of encephalopathies caused by mutations in the UFM1 gene by deletion of the gene in murine neurons and replacing it by expression of constructs carrying either wildtype sequence or the sequence of a pathological human mutation. The work is technically sound and includes a state-of-art strategies. The medical significance is twofold: firstly, a potential disease mechanism in humans is identified, and secondly, a drug approved for the treatment of humans is used to correct the cell physiological deficits of neurons expressing the mutant human protein. Altogether I consider the mouse model as very reasonable for the addressed topic.

Reviewer #1 (Remarks for Author):

Using suitable mouse models, the authors studied potential molecular and cellular pathomechanisms underlying encephalopathies caused by absence or pathological mutations of the UFM1 gene and consequently the disturbance of the process of UFMylation. In a first in vivo experiment, the Ufm1 gene was knocked out in the murine cortex after day 14.5 in utero and analyzed at postnatal day P23. This resulted in simplification of neuronal architecture and significant reduction in excitatory synapse number indicating clear deficits in neuronal development. All further experiments were

performed in primary cultures of hippocampal neurons from cKO mice with a floxed Ufm1 gene upon infection with viruses {plus minus}Cre recombinase expression. For electrophysiological experiments, specifically autaptic micro-island cultures were used. These experiments basically confirm the in vivo findings. Individual synapses, however, that are formed seemed relatively normal. As major consequence of impaired UFMylation, a strong reduction of protein translation was detected in UFM1-KO primary neurons. In a clever set of experiments, re-expression of either wildtype UFM1 or UFM1 carrying a pathological human mutation (UFM1-R81C) was used to assess for rescue of observed phenotypes. Thereby expression levels of UFM1-WT and UFM1-R81C were carefully controlled. Under these conditions wildtype proteins favorably compensated UFM1 deficiency whereas UFM1-R81C only partially rescued the defects. Main deficits, as compared to wildtype rescue, were observed in response to ER stress. The data suggest that the human mutation in UFM1 enhances the unfolded protein stress response via the Protein kinase R-like ER kinase (PERK) pathway. Interestingly, trazodone, an inhibitor of the PERK pathway that is already in use as an antidepressant, can counteract UFM1-R81C deficits in normalizing unfolded protein response and therefore is a candidate to be repurposed for patients carrying the UFM1 gene mutation. This is a thoroughly designed and well-performed study that contributes significantly to the understanding of cellular mechanisms underlying UFMylation-related neurological disorders and identifies a candidate drug to be assessed for future treatment. I have only a couple of minor comments and questions.

We are very grateful to this reviewer for their very thorough appreciation of our work.

1) General: The authors convey the impression that UFM1 dysfunction leading to encephalopathy only affects excitatory neurons and glutamatergic synapses. The only experiment concerning inhibitory neurotransmission documents that GABAergic input on primary excitatory neurons seems normal. I hardly can believe that UFMylation does not affect interneurons and suggest that this point should be addressed in the discussion.

This reviewer is correct in assuming that inhibitory neurons should be similarly affected by UFM1 dysfunction. We performed additional targeted experiments to address this issue by investigating synaptic transmission in autaptic GABAergic striatal neurons. We observed similar defects as seen in excitatory hippocampal neurons, with effects being more modest than the ones observed in excitatory neurons. The new data are now shown as Figure S4 and described in the text (page 6, lines 200-211).

2) For clarity and understandability for non-specialists, all essential abbreviations should be defined also in the abstract (e.g. PERK-UPR). In general, a list of abbreviations may help to increase the readability.

All acronyms have now been properly defined throughout the revised manuscript, including the abstract. A list of abbreviations has been included in the revised manuscript.

3) Cartoon in synopsis: If I understand correctly, the phenotype mainly affects neuronal architecture, dendritic branching and reduced number of synapses, while available

synapses are relatively normal. The cartoon gives the impression that individual synapses are severely impaired in their function. Is this what is meant?

We are grateful to this reviewer for pointing out this inaccuracy. Indeed, the remaining synapses are functionally and structurally normal. We added Figure S5 (corresponding text description page 6, lines 212-227), which demonstrates that the remaining synapses in UFM1-KO cells have similar ultrastructure as WT synapses. We also revised the graphical abstract cartoon to avoid giving the impression that the function of individual synapses is severely altered.

4) page 4: "... we combined CRISPR/Cas9 and in utero electroporation (IUE) to deplete UFM1 in E14.5 mouse cortical progenitors and their neural progeny at P23." This statement needs clarification. I assume it means that UFM1 is deleted from E14.5 onwards in cortical progenitors and their neural progeny and that P23 was the day analysis. Please, clarify.

This reviewer is correct. IUE was performed at E14.5, and effects on neurons were assessed at P23. The corresponding text has been amended in the revised manuscript to clarify this issue (page 5, lines 130-132).

5) Fig. 1 (legend): "Data from B and C were obtained...", I assume that this must read "C and "D"

We are grateful to this reviewer for pointing out this error, which has been corrected in the revised manuscript.

6) Fig. 5: Legends for panels A and B are basically identical. However, panel A does not show any immunostaining. Please, adapt the text to what is really documented. Panel D, lettering of 'CRE' is a bit odd. Legend last paragraph: "For -H, data were obtained ..." means what? "For ??-H, data were obtained ...".

We thank this reviewer for pointing out these mistakes. The legend of Figure 5A shows binary images, while panel B shows typical immunostaining images. We have clarified this point, adjusted the CRE lettering in panel D, and finally, we have corrected the last paragraph of the legend of Figure 5.

7) Fig. 8: Maybe a small cartoon could be added to this figure indicating the role of the analyzed players in the UFMylation process. This could ease the understanding of the assay.

As requested, we have added UFMylation schemes in Figures 8 of the revised manuscript.

8) Fig. 10: The final statement in the result section on Trazodone action seems a bit strong. What are the significance values? There is a clear increase in Synapsin puncta

(Fig. 10D), but no significant difference in postsynaptic marker PSD95 (Fig. 10C) nor in the overlap of pre-and postsynaptic markers (Fig. 10E). What are the actual p-values? Explain the use of different statistical tests.

We state in all legends the types of statistical tests used and the levels of significance (p-values). The choice of statistical test depends on the type of distribution of the data and the type of comparison needed. We added a paragraph in the methods section of the revised manuscript that explains our choices of statistical tests (page 19, lines 690-701). Finally, we have revised our statement about the effect of trazodone on synapse puncta (page 12, lines 439-443, page 15, lines 550-552).

9) Fig. S8: Panel assignment is wrong, panel E is described as panel D (which appears twice in the legend).

We apologize for this lack of clarity and corrected the corresponding passage.

Reviewer #2 (Remarks for Author):

General Comment

This manuscript presents a comprehensive investigation into the cellular and molecular mechanisms underlying neurodevelopmental disorders caused by UFM1 mutations, including complete loss-of-function and the patient-derived R81C missense variant. Using both in vivo and in vitro models, the authors convincingly demonstrate that UFM1 deficiency impairs dendritic arborization, synaptogenesis, and synaptic transmission, accompanied by a marked reduction in protein synthesis and hyperactivation of the ER stress PERK-UPR pathway. Although the R81C mutant retains partial functionality, it exhibits an aberrant response to ER stress and fails to fully rescue the phenotypes observed in UFM1-deficient neurons. Notably, the authors show that the antidepressant trazodone can partially restore protein synthesis and synapse number in neurons expressing UFM1-R81C, highlighting potential therapeutic relevance. The experimental design is thorough, employing a wide array of genetic, molecular, and electrophysiological techniques, with well-controlled and consistent results. The study offers valuable mechanistic insights into UFMylation-dependent neuronal development and function, and significantly contributes to our understanding of UFM1-linked encephalopathies.

Below, I provide major and minor comments aimed at improving clarity, interpretative depth, and contextual framing of the findings.

Major Comments

1. Unclear mechanistic classification of the UFM1-R81C variant

The authors describe UFM1-R81C as "not merely a loss-of-function variant," but its precise functional classification remains ambiguous. Given the observed phenotype - partial rescue under basal conditions and exaggerated ER stress responses under challenge - alternative mechanisms such as a dominant-negative or gain-of-toxic-function effect should be considered and discussed more explicitly.

The reviewer is correct. UFM1-R81C only partially rescues the UFM1-KO phenotypes, indicating a certain degree of loss of function variant under physiological conditions. However, upon ER stress, UFM1-R81C-expressing neurons exhibit an aggravated ER stress response, which supports the notion that this variant causes a gain-of-toxic-function. We now considered this notion in our discussion, as requested (page 14, lines 524-537). We did so carefully because we do not fully understand the entire extent by which the R81C mutation affects UFMylation and its targets, and we believe that a deep analysis of the WT and R81C UFMylome is required to fully understand the pathogenic mechanisms of the R81C mutation – which is beyond the scope of the present study.

2. Unresolved identity of UFM1-conjugated target proteins

The Western blot data reveal a reduction in high-molecular-weight UFM1-conjugates in UFM1-deficient neurons, suggesting the existence of multiple substrates. However, the identities of these putative UFM1 targets remain undetermined. Further proteomic profiling or immunoprecipitation experiments (even preliminary) would be valuable, or at least the authors should more clearly state how they plan to address this important question in future work.

We thank this reviewer for this important comment. We included a preliminary dataset of neuronal UFM1 substrates (Table 2, Figure S17, corresponding text page 13, lines 476-490). We emphasize the preliminary aspect of the corresponding candidate list, which is based on a single affinity purification experiment. Biological replicates experiments will have to confirm the validity of the identified candidates, along with solid biochemistry to confirm their UFMylation in mouse brain. This notwithstanding, we decided to add the corresponding data as first insight, and now briefly discuss the results in the revised manuscript (page 13, lines 476-490). The candidates identified are inspiring and as they may extend the functional relevance of protein UFMylation beyond the ER and protein translation. Most interesting is the identification of DOCK4, which is linked to neurite growth and synapse formation (PMID: 30078728, 32009906) and whose heterozygous loss has been associated with neurodevelopmental delay and microcephaly (PMID: 38526744). Altogether, defects in DOCK4 function cause phenotypes that are akin to those observed in UFM1-KO neurons and in diseases caused by UFMylation defects.

3. Unclear mechanistic basis for trazodone's selective effects

The study demonstrates that trazodone rescues translation and synapse number in UFM1-R81C-expressing neurons but not in UFM1-null neurons. However, the molecular basis of this differential response is not well explained. A deeper discussion of possible mechanisms - for example, differences in residual UFMylation, PERK-UPR signaling

thresholds, or alternative pathways - would enhance the interpretive strength of the study.

We agree with this comment. It is striking that Trazodone rescues protein translation in UFM1-R81C expressing neurons but not in UFM1-KO neurons. One possible explanation is residual UFMylation activity in UFM1-R81C expressing neurons. This residual activity may sufficiently maintain ER homeostasis and thus tame the UPR pathway, but we do not know at which level Trazodone and UFMylation intersect. Therefore, alternative pathways may explain the differential rescue effect of Trazodone in the two different genetic scenarios. We have clarified this point in the revised discussion (page 15, lines 553-566).

4. Potential confusion due to antibody limitations in detecting UFM1-R81C

Since the C-terminal epitope of UFM1-R81C is altered, it is not detected by the standard anti-UFM1 antibody, leading to an apparent lack of signal in immunoblots. While this issue is briefly addressed in supplementary figures, the main text and figure legends would benefit from a clearer and more prominent explanation to avoid misinterpretation by readers.

We dedicated three supplementary figures and a longer passage in the main text to explain these limitations and to describe how we have overcome them (page 8, lines 261 to page 9, lines 313). We have now further improved the text to avoid misinterpretation.

5. Lack of long-term functional or behavioral assessment of trazodone treatment

Although trazodone shows partial rescue of molecular and cellular phenotypes, the study does not evaluate whether these changes translate into long-term improvements in neuronal function or behavior. Future work should consider examining electrophysiological network activity, calcium dynamics, or behavioral outputs to assess therapeutic relevance more thoroughly.

We thank the reviewer for this very pertinent comment. Our primary focus with the present study has been to understand the cell-autonomous role of UFM1 in neurons and to characterize corresponding UFM1-associated pathological perturbations in UFM1-associated genetic variants. We found that Trazodone treatment can restore normal protein synthesis in cultured neurons expressing mutant UFM1-R81C, which is exciting, but we agree that it will be important to also determine the long-term implications on neuronal function of Trazodone treatment upon pathological UFMylation. A possible strategy to address this issue would be to generate and analyze a UFM1-R81C knock-in mouse model for *in vivo* studies of Trazodone treatment. However, such an approach cannot be envisioned within the time frame of the present study. Furthermore, uncertainty arises from the lack of clear knowledge about the intersection between the Trazodone effects and UFMylation. Therefore, while clearly important, we believe that this future avenue of investigation is beyond the scope of the current submission. This notwithstanding, we have clarified our discussion to address this point (page 15, lines 560-566).

Minor Comments

1. Statistical reporting clarity

The manuscript employs a range of statistical tests (e.g., Mann-Whitney, Kruskal-Wallis, ANOVA), but the criteria for selecting each test (e.g., normality checks) are not consistently stated in the Methods section or figure legends. Including the rationale for test selection alongside the sample sizes (n) would improve transparency and reproducibility.

The choice of statistical test depends on the type of distribution of the data and the type of comparison needed. We have now added a paragraph in the methods section of the revised manuscript that explains our choices of statistical tests (page 19, lines 690-701).

2. Inconsistent terminology

Terms such as "UFM1-R81C," "R81C," and "UFM1-R81C-expressing neurons" are used somewhat interchangeably throughout the manuscript. Standardizing terminology across the text would enhance readability and avoid ambiguity, particularly for non-specialist readers.

We apologize for this lack of consistency. We will carefully clarify this terminology in the revised version of the manuscript.

3. Insufficient citation of original literature

The Introduction and Discussion sections rely heavily on review articles, with limited citation of key original studies. Important foundational and mechanistic references are notably absent, including:

- Komatsu et al., 2004 (*EMBO J*), which first identified and characterized the UFM1 conjugation system;

- Walczak et al., 2019 and DaRosa et al., 2024, which demonstrated UFMylation of RPL26 and its role in ER-associated ribosome quality control and translational regulation. Inclusion of these primary sources would significantly strengthen the contextual framework of the study and better acknowledge the existing body of work in this field.

We thank the reviewer for pointing out this shortcoming. We included these original publications in the revised version.

Reviewer #3 (Comments on Novelty/Model System for Author):

They claim using mice to study neuronal development but they show cell culture. Neuronal development needs to be studied in mice.

We believe that many key aspects of neuronal development can be studied reliably with cultured primary neurons, especially cell-autonomous aspects of neuronal development. Numerous accounts in the literature support this notion. We chose this path in our first study on neuronal UFMylation because it has many advantages. For instance, genetic replacement strategies allowed us to obtain a larger sample sizes of the different genotypes. Further, autaptic neurons are particularly well-suited for studying synaptic transmission in detail (e.g. PMID 28231043, 21040848). Clearly, additional studies in the future are indeed needed to fully understand the role of UFMylation in neuronal development, particularly *in vivo*, but our present work constitutes the first important insight in the field.

The medical potential is low because it is unrealistic to imagine a treatment for a developmental disease due to a genetic mutation ie no drug will rescue the brain defects.

We believe that the reviewer's assessment of this issue is disputable. An extremely important case-in-point is an earlier study of the Bird lab that demonstrated the reversibility of Rett syndrome phenotypes in a genetic KO-first mouse model (PMID 17289941). Similar evidence for the - partial - reversibility of disease-related 'neurodevelopmental' phenotypes was subsequently acquired for Neurexin KOs (PMID 24995986). These studies show that genetically caused 'neurodevelopmental' defects can indeed be reversible and are not always 'hardwired' and hence irreversible - a notion that continues to fuel attempts to develop corresponding therapies. Clearly, the development of a pharmacological strategy to reverse UFMylation defects is a major task for the future. But we would like to stress that such attempts cannot simply be dismissed as impossible. Rather, any positive lead towards a possible pharmacological treatment, especially with an approved drug, can be immensely important.

Reviewer #3 (Remarks for Author):

In this manuscript the authors claim using a Ufm1 knock-out mice, to investigate two types of UFMylation pathologies, UFM1 loss and expression of a pathogenic UFM1-R81C variant. They claim that UFM1-deficiency confounds neuron development and synapse function. They conclude "Our study unveils a pivotal role for UFMylation in neuronal development, provides a molecular understanding of the signaling mechanisms altered in UFM1- associated encephalopathies, and offers important insights into potential treatments for these disorders". There is a disconnect between the claims and the content of the manuscript and several methodological issues.

Major issues:

1. The abstract infers that the work is done in mice, whilst in fact most of the work is done in cultured neurons.

We revised our abstract to more appropriately describe our work.

2. The conclusions of the study are not novel. It was known before that UFM1 deficiency impairs neuronal development and function. They claim new mechanistic insights. This is not the case. There may be opportunities to explore new aspects of UFMylation in this study but they are not explored. (see below).

What are the UFM1-positive protein species in Figure 2. Identifying them could reveal mechanisms of neuronal UFM1.

As requested, we included a preliminary dataset of neuronal UFM1 substrates (Table 2, Figure S17, corresponding text page 13, lines 476-490). We emphasize the preliminary aspect of the corresponding candidate list, which is based on a single affinity purification experiment. Biological replicates experiments will have to confirm the validity of the identified candidates, along with solid biochemistry to confirm their UFMylation in mouse brain. This notwithstanding, we decided to add the corresponding data as first insight, and now briefly discuss the results in the revised manuscript (page 13, lines 476-490). The candidates identified are inspiring and as they may extend the functional relevance of protein UFMylation beyond the ER and protein translation. Most interesting is the identification of DOCK4, which is linked to neurite growth and synapse formation (PMID: 30078728, 32009906) and whose heterozygous loss has been associated with neurodevelopmental delay and microcephaly (PMID: 38526744). Altogether, defects in DOCK4 function cause phenotypes that are akin to those observed in UFM1-KO neurons and in diseases caused by UFMylation defects.

3. They need to show the efficacy of their gene targeting approach before the phenotypic validation. They don't show the efficacy of the CRISPR/Cas9 in utero electroporation to deplete UFM1 in E14.5 mouse cortical progenitors and their neural progeny at P23. This is a major problem with this study. One imagines that the in utero targeting efficacy of the Crisp targeting is not 100% and that the animals they generated are mosaic. There ought to be much variation. How do the authors assess whether what they report is due to inherent variability of the system or differences resulting from the genetic modification. The study is inconclusive as it stands.

The reliability of the CRISPR/Cas9 tools we used had been validated previously (PMID 31595041), and the use of the double-nicking CRISPR/Cas9 strategy was shown to considerably reduce off-target effects (PMID 23992846). Our IUE method allows for targeting cortical cell lineages with spatiotemporal precision in the brain, while leaving other tissues as well as surrounding neurons unaffected, thus permitting stringent analyses of cell-autonomous effects of genetic alterations. Indeed, the fact that the approach creates a mosaic of mutated cells in an otherwise unperturbed tissue is a key advantage, which makes it a state-of-the-art in vivo DNA delivery method to investigate cell-autonomous functions of genes of interest (PMID 17406448, 38849354, 32249816, 30392800, 38849354, 40300602 a.m.o.). By co-transfecting an EGFP expression plasmid with either control non-targeting or UFM1-targeting sgRNAs we can reliably

identify mutated cells and analyze them. In essence, the mosaic character of the tissue we analyzed is a key advantage, rather than a concern for our analysis. Of note, the CRISPR-Cas9 system allows to create frame-shift mutations in the locus of interest and, given the clonal divisions of in utero electroporated neural stem cells, efficient knock-out in the differentiated, postmitotic neurons. Such a combination of CRISPR-Cas9 and IUE has been broadly employed in developmental neuroscience for years now (e.g. PMID 26067104, 26758805, 33689678, 38849354). Further, we would like to point out that the variance in dendritic complexity and synapse number, neuromorphological readouts of UFM1-defects, is minor (Figure 1). Above statements notwithstanding, we have now re-validated the knock-out of UFM1 in vitro by performing nucleofection of N2a cells, followed by immunostaining and Western blotting (Figure S1). We believe that this experiment now resolves the issues raised by this reviewer.

2. There are methodological issues with the translation experiments; They find that CHX reduces translation signal to 50% which establishes that they have background staining (as often in such assays). This invalidates these experiments and their conclusions.

We clarified this point in the revised version of the manuscript. Firstly, background immunostaining was removed from all calculations, a key point that we now have clarified in the methods part of the revised submission (Supplementary Methods). Secondly, many studies showed similar decreases in protein translation upon CHX treatment, which confirms our observation (PMID 27764673, 4041860, 27026294, 38849354).

3. The gel Fig 4C is not equally loaded. Decrease in puro label parallels that of the total proteins.

We conducted four independent experiments to address this issue, and obtained statistically convincing data. We used the memcode stain for normalization, a sensitive and reliable method that has clear advantages over the use of marker protein levels as normalizing controls. We concede that the image we chose to document this experiment is not ideally representative. We have now replaced it with a more representative image in the revised manuscript (Figure 4C).

4. The measurements of UPR activation needs to be done with robust and validated readouts, assessing both protein and mRNA changes and including a control such as Tg or Tm to ensure that they are within the range of detection of UPR activation in their system. The blots showing ATF6, PERK and IRE1a, P-elf2 are not indicative of UPR activation.

Consistently in the past, especially when highly proliferating cells were studied, activation of the UPR has been characterized by Western blot analysis of PERK, ATF6, and/or p-elf2alpha (e.g. PMID 38849354, 30842412, 35299279, 31924446, 28921568, 39116259, 26450683, 31251916, 38177917, 25864123, a.m.o.). We believe that our analysis of these markers is adequate.

Furthermore, a link between UFMylation defects and UPR activation was described previously, and the corresponding information prompted us to study this link

further in our setting. We show that key components of the UPR - i.e. ATF6, p-eIF2 α , IRE1 α , and PERK - are altered in UFM1-KO neurons and in neurons expressing UFM1-R81C, with PERK exhibiting significant changes. These results confirm activation of the UPR under our experimental conditions.

To further address this reviewer's comments, we performed Western blot and qPCR analysis of the three main pillars of the UPR pathway, i.e. PERK, ATF6 and IRE1 α (Figure S14C-14J), in the presence of thapsigargin, a well-known ER stressor. Our data clearly validate our methodology as thapsigargin treatment leads to increased levels of PERK not only in WT samples, but in all four conditions (RFP, CRE, WT, R81C, Figure S14D). While mRNA levels for PERK, ATF6, and IRE1 α were unaltered by the thapsigargin treatment, *Xbp1* cleavage was confirmed in all conditions (Figure S14G-14M). We believe that this dataset validates our experimental approach and that this set of control experiments now resolve the issue raised by this reviewer.

18th Dec 2025

Dear Dr. Tirard,

Thank you for the submission of your revised manuscript to EMBO Molecular Medicine. I am pleased to inform you that we will be able to accept your manuscript pending the following final amendments:

- 1) Please address the referee #2 points by discussion.
- 2) Authors: We note name discrepancy Frederieke Moschref in manuscript text and Frederike Moschref in our submission system. Please correct.
- 3) Figures: Main figures should be uploaded as individual high-resolution files, with their legends in the manuscript text. Please check "Author Guidelines" for more information:
<https://www.embopress.org/page/journal/17574684/authorguide#figureformat>
- 4) In the main manuscript file, please do the following:
 - Please address all comments suggested by our data editors listed below:
 - o Data availability statement:
 1. Please note that the specific URLs for GSE299278, PXD071267 datasets are not provided in the data availability statement.
 - o Figure legends:
 1. Please define the annotated p values ****/**/*/* as well as provide the exact p-values for the same in the legend of figure 6F, G as appropriate.
 2. Please note that the exact p values are not provided in the legends of figures 1C, D, F, G, H; 2C, E, F, G, H, I, K, L, M; 3B, C, D, E, H, I, P; 4B, D; 5D, E, F, G, H; 6B, C, D, I-K; 7B, F, H; 9C, E; 10B, C, D, E.
 3. Please indicate the statistical test used for data analysis in the legends of figures 6F, G.
 4. Please note that information related to n is missing in the legends of figures 6F, G; 8C, E, G.
 5. Please note that the error bars are not defined in the legends of figures 6F, G; 8C, E, G.
 - Add up to 5 keywords.
 - Remove data not shown (line 992).
 - Add callouts for Fig 6G and Appendix Figure S16. Figures should be called out in sequential order. Currently Fig 3 & 4 are called out before Fig 2B. Please correct.
 - Remove "Abbreviations".
 - In Methods, remove "Vectors used in this study".
 - Add "Disclosure and competing interests statement". We updated our journal's competing interests policy in January 2022 and request authors to consider both actual and perceived competing interests. Please review the policy <https://www.embopress.org/competing-interests> and update your competing interests if necessary.
 - Author contributions: Please remove it from the manuscript and specify author contributions in our submission system. CRediT has replaced the traditional author contributions section because it offers a systematic machine-readable author contributions format that allows for more effective research assessment. You are encouraged to use the free text boxes beneath each contributing author's name to add specific details on the author's contribution. More information is available in our guide to authors:
<https://www.embopress.org/page/journal/17574684/authorguide#authorshipguidelines>
 - Indicate in legends exact n and exact p values, not a range, along with the statistical test used. To keep the figures "clear" some authors found providing an Appendix table Sx with all exact p-values preferable. You are welcome to do this if you want to.
 - Remove "Code availability".
 - In data availability statement leave only information about datasets deposited in public repositories. Please use the following format to report the accession number of your data:

[data type]: [full name of the resource] [accession number/identifier] ([doi or URL or identifiers.org/DATABASE:ACCESSION])

Please check "Author Guidelines" for more information.

<https://www.embopress.org/page/journal/17574684/authorguide#availabilityofpublishedmaterial>

- Move the figure legends after the References list.

5) Supplementary Methods: Please move suppl. methods to the Methods section in the main manuscript text. Move the tables with the primers and antibodies to the end of the manuscript text, add legends and callouts, and name them Table 1 and Table 2.

6) Tables: Please rename the tables "Dataset EV1" and "Dataset EV2". Add a legend with the dataset titles and a short description to each excel file, in a separate tab/worksheet.

7) Appendix: Please rename the suppl. figures file "Appendix" and upload it in PDF format. Add a table of contents with page numbers and correct the nomenclature to "Appendix Figure S1" etc. throughout. Also, update their callouts in the main manuscript text.

8) Source data: Please upload source data files as one ZIP file per figure.

9) The Paper Explained: Please provide "The Paper Explained" and add it to the main manuscript text. Please check "Author

Guidelines" for more information.

<https://www.embopress.org/page/journal/17574684/authorguide#researcharticleguide>

10) Synopsis:

- Synopsis text: Please remove it from the manuscript and upload it as a separate .doc file.

- Synopsis image: Please remove it from the manuscript and submit the image as a separate 550 px-wide x 300-600 pixels high high-resolution jpeg file.

11) As part of the EMBO Publications transparent editorial process (see our Editorial at

<http://embomolmed.embopress.org/content/2/9/329>), EMBO Molecular Medicine will publish online a Review Process File (RPF) to accompany accepted manuscripts. This file will be published in conjunction with your paper and will include the anonymous referee reports, your point-by-point response and all pertinent correspondence relating to the manuscript. Let us know if you want to remove or not any figures from it prior to publication. Please note that the Authors checklist will be published at the end of the RPF.

12) Please provide a point-by-point letter INCLUDING my comments as well as the reviewer's reports and your detailed responses (as Word file).

I look forward to reading a new revised version of your manuscript as soon as possible.

Yours sincerely,

Zeljko Durdevic

Zeljko Durdevic
Senior Editor
EMBO Molecular Medicine

*** Instructions to submit your revised manuscript ***

When preparing your revised manuscript, please refer to our guidelines: <https://link.springer.com/journal/44321/submission-guidelines#cms-Revised-submissions>. We perform an initial quality control of all revised manuscripts before re-review; failure to include requested items will delay the evaluation of your revision.

We require:

2) Individual production quality figure files as .eps, .tif, .jpg (one file per figure). For guidance, download the 'Figure Guide PDF': <https://media.springernature.com/original/springer-cms/rest/v1/content/27825798/data/v1>.

3) A .docx formatted letter INCLUDING the reviewers' reports and your detailed point-by-point responses to their comments. As part of the EMBO Press transparent editorial process, the point-by-point response is part of the Review Process File (RPF), which will be published alongside your paper.

4) A complete author checklist, which you can download from our author guidelines. Please insert information in the checklist that is also reflected in the manuscript. The completed author checklist will also be part of the RPF.

6) It is mandatory to include a 'Data Availability' section after the Materials and Methods. Before submitting your revision, primary datasets produced in this study need to be deposited in an appropriate public database, and the accession numbers and database listed under 'Data Availability'. Please remember to provide a reviewer password if the datasets are not yet public.

7) For data quantification: please specify the name of the statistical test used to generate error bars and P values, the number (n) of independent experiments (specify technical or biological replicates) underlying each data point and the test used to calculate p-values in each figure legend. The figure legends should contain a basic description of n, P and the test applied. Graphs must include a description of the bars and the error bars (s.d., s.e.m.).

9) Our journal encourages inclusion of *data citations in the reference list* to directly cite datasets that were re-used and obtained from public databases. Data citations in the article text are distinct from normal bibliographical citations and should directly link to the database records from which the data can be accessed. In the main text, data citations are formatted as follows: "Data ref: Smith et al, 2001" or "Data ref: NCBI Sequence Read Archive PRJNA342805, 2017". In the Reference list, data citations must be labeled with "[DATASET]". A data reference must provide the database name, accession number/identifiers and a resolvable link to the landing page from which the data can be accessed at the end of the reference.

12) Author contributions: You will be asked to provide CRediT (Contributor Role Taxonomy) terms in the submission system. These replace a narrative author contribution section in the manuscript.

13) A Conflict of Interest statement should be provided in the main text.

14) Every published paper includes a 'Synopsis' to further enhance discoverability. Synopses are displayed on the journal webpage and are freely accessible to all readers. They include a short stand first (maximum of 300 characters, including space) as well as 2-5 one-sentences bullet points that summarizes the paper. Please write the bullet points to summarize the key NEW findings. They should be designed to be complementary to the abstract - i.e. not repeat the same text. We encourage inclusion

of key acronyms and quantitative information (maximum of 30 words / bullet point). Please use the passive voice. Please attach these in a separate file or send them by email, we will incorporate them accordingly.

15) Include a Reagents and Tools Table as part of the Methods section, which can be downloaded from our author guidelines.

Photos 400-800 DPI

*Additional important information regarding figures and illustrations can be found at

<https://media.springernature.com/original/springer-cms/rest/v1/content/27825798/data/v1>

***** Reviewer's comments *****

Referee #1 (Comments on Novelty/Model System for Author):

The authors have added a lot of additional data in this revised version to further substantiate their findings. They have used a broad spectrum of adequate methods to elucidate the cellular mechanisms behind UFM1 gene knockout and the human pathological mutation. As said originally, there is a two-fold medical impact, i.e., contribution to i) understanding disease mechanisms, and ii) understanding the way of action of an approved medical drug. While most mechanistic analyses were done in cultured primary neurons, initial studies to describe the neuronal phenotype of UFM1-deficiency in neurons was performed upon in utero electroporation. This was necessary to circumvent lethality of global UFM1 knockout. I consider this research strategy as highly appropriate.

Referee #1 (Remarks for Author):

The authors have adequately addressed all my questions and suggestions. They have added a considerable amount of new data to the manuscript, which confirms and expands on their original findings.

Referee #2 (Remarks for Author):

Thank you for your careful responses and revisions. The reviewer has concluded that the revisions comprehensively and constructively address my comments. The manuscript has been significantly improved and is now suitable for publication in EMBO Molecular Medicine.

The following points are particularly commendable:

The clear inclusion of the possibility of a gain-of-toxic-function effect for the R81C variant in the discussion.

The inclusion of preliminary, yet valuable, data on potential UFM1-conjugated targets (e.g., DOCK4), which enhances the depth of the discussion.

The detailed improvements regarding the selective mechanism of action of trazodone and the increased transparency in statistical reporting.

These amendments have substantially increased the scientific rigor and clarity of the study.

While I recommend acceptance of the current manuscript, I wish to re-emphasize the importance of the following two areas for future research endeavors:

1. Identification and Verification of bona fide UFM1 Substrates:

The provided list of candidates is highly interesting. It will be crucial to rigorously verify, using biochemical methods, that these are genuine UFM1 substrates in neuronal cells. This verification is essential for reaching the core understanding of UFMylation's role in neurodevelopment.

2. Precise Functional Classification of the R81C Mutation:

Whether R81C acts merely as a loss-of-function (LOF) or as a gain-of-function (GOF) variant is critical for therapeutic considerations. A detailed analysis of the UFMylation profile (UFMyome analysis) comparing WT UFM1 and the R81C mutant- potentially considering the possibility of an "aberrant promotion" of UFMylation-is indispensable for understanding the pathogenesis.

These points will serve as important milestones for your future research.

Overall, the reviewer finds the current revised version suitable for acceptance.

***** Reviewer's comments *****

Referee #1 (Comments on Novelty/Model System for Author):

The authors have added a lot of additional data in this revised version to further substantiate their findings. They have used a broad spectrum of adequate methods to elucidate the cellular mechanisms behind UFM1 gene knockout and the human pathological mutation. As said originally, there is a two-fold medical impact, i.e., contribution to i) understanding disease mechanisms, and ii) understanding the way of action of an approved medical drug. While most mechanistic analyses were done in cultured primary neurons, initial studies to describe the neuronal phenotype of UFM1-deficiency in neurons was performed upon in utero electroporation. This was necessary to circumvent lethality of global UFM1 knockout. I consider this research strategy as highly appropriate.

Referee #1 (Remarks for Author):

The authors have adequately addressed all my questions and suggestions. They have added a considerable amount of new data to the manuscript, which confirms and expands on their original findings.

Referee #2 (Remarks for Author):

Thank you for your careful responses and revisions. The reviewer has concluded that the revisions comprehensively and constructively address my comments. The manuscript has been significantly improved and is now suitable for publication in EMBO Molecular Medicine.

The following points are particularly commendable:

The clear inclusion of the possibility of a gain-of-toxic-function effect for the R81C variant in the discussion.

The inclusion of preliminary, yet valuable, data on potential UFM1-conjugated targets (e.g., DOCK4), which enhances the depth of the discussion.

The detailed improvements regarding the selective mechanism of action of trazodone and the increased transparency in statistical reporting.

These amendments have substantially increased the scientific rigor and clarity of the study.

While I recommend acceptance of the current manuscript, I wish to re-emphasize the importance of the following two areas for future research endeavors:

1. Identification and Verification of bona fide UFM1 Substrates:

The provided list of candidates is highly interesting. It will be crucial to rigorously verify, using biochemical methods, that these are genuine UFM1 substrates in neuronal cells. This verification is essential for reaching the core understanding of UFMylation's role in neurodevelopment.

We fully agree with this assessment. The refinement of the pool of candidate UFM1 substrates via further biochemical studies and the validation of novel UFM1 substrates are the main foci of our current work.

2. Precise Functional Classification of the R81C Mutation:

Whether R81C acts merely as a loss-of-function (LOF) or as a gain-of-function (GOF) variant is critical for therapeutic considerations. A detailed analysis of the UFMylation profile (UFMylome analysis) comparing WT UFM1 and the R81C mutant-potentially considering the possibility of an "aberrant promotion" of UFMylation-is indispensable for understanding the pathogenesis.

These points will serve as important milestones for your future research.

Overall, the reviewer finds the current revised version suitable for acceptance.

We have now emphasized this aspect in our discussion, page 14, line 573-575.

28th Jan 2026

Dear Dr. Tirard,

We are pleased to inform you that your manuscript is accepted for publication and is now being sent to our publisher to be included in the next available issue of EMBO Molecular Medicine. Please make sure that all deposited data are freely accessible upon publication.

You may qualify for financial assistance for your publication charges - either via a Springer Nature fully open access agreement or an EMBO initiative. Check your eligibility: <https://link.springer.com/journal/44321/how-to-publish-with-us>

Zeljko Durdevic
Senior Editor
EMBO Molecular Medicine

>>> Please note that it is EMBO Molecular Medicine policy for the transcript of the editorial process (containing referee reports and your response letter) to be published as an online supplement to each paper. If you do NOT want this, you will need to inform the Editorial Office via email immediately. More information is available here: <https://link.springer.com/partners/embo-press/editorial-policies#Peer%20review>